# Customizing Reinforcement Learning Agent with Multi-Objective Preference Control

## Abstract

Practical reinforcement learning (RL) usually requires agents to be optimized for multiple potentially conflicting criteria, e.g. speed vs. safety. Although Multi-Objective RL (MORL) algorithms have been studied in previous works, their trained agents often lack precise controllability of the delicate trade-off among multiple objectives. Hence, the resulting agent is not versatile in aligning with customized requests from different users. To bridge the gap, we develop "Preference control (PC) RL", which trains a meta-policy that takes user preference as input controlling the generated trajectories within the preference region on the Pareto frontier. To this end, we train a preference-conditioned meta-policy by our proposed preference-regularized MORL algorithm. The achieved meta-policy performs as a multi-objective optimizer that can directly generate user-desired Pareto solutions. The proposed algorithm is analyzed and its convergence and controllability are theoretically justified. We evaluate PCRL on a discrete toy environment and challenging high-dimensional robotic control tasks with up to six objectives. In these experiments, PCRL-trained policies exhibit significantly better controllability than existing approaches and can generate Pareto solutions with better diversity and utilities.

## 1 Introduction

Multi-Objective Reinforcement Learning (MORL) has attracted growing interests in applications of training sequential decision-making agents that satisfy multiple objectives. In practice, optimizing for potentially multiple criteria often involves managing trade-offs between them. For example, speed vs. safety or distance vs. energy for robotic control tasks. Many previous MORL methods (Yang et al., 2019; Abels et al., 2019; Xu et al., 2020; Lu et al., 2023; Alegre et al., 2023) tried to address the trade-off issue by optimizing a linearly scalarized objective, which sums up multiple objectives with preference weights. However, Linear Scalarization (LS) approach's solutions do not always guarantee to achieve desired trade-offs. For example, even with equal weights on two objectives, LS may heavily optimize one objective over the other (as shown in Fig. 1). Another limitation is that some methods Xu et al. (2020); Alegre et al. (2023) require learning multiple models to identify the Pareto front , which is unscalable to increasing objectives or model sizes. By the design, each model only handles one specified preference, while a versatile RL agent should be adaptable to various unseen user preferences.

This inspires us to develop an innovative MORL scheme, "Preference Control (PC) RL", to train a meta-policy that takes user preference as input to control the trade-off between objectives. The scheme enables us to apply methods (Lin et al., 2019; Mahapatra and Rajan, 2020) from the Multi-Objective Optimization (MOO) domain to overcome the limitation of LS and find preference-specific solutions on the Pareto front. Moreover, inspired by how some MOO methods (Désidéri, 2009; Liu et al., 2021; Xiao et al., 2023) deal with conflicting gradients and stochastic gradients, we propose a novel MORL-specific algorithm "PreCo" and conduct a theoretical analysis of it, which proves that, with stochastic gradients, PCRL using PreCo can achieve Pareto stationary solutions precisely controlled by the input preference.

We conducted experiments in environments with conflicting objectives Felten et al. (2023) to empirically demonstrate that (1) our PCRL scheme is compatible with various MOO methods; and (2) PCRL with PreCo consistently achieves superior performance across multiple MORL environments. In particular, our method excels in cases with a large number of objectives or conflicting objectives.

## 2 PRELIMINARIES

**Multiple Objective Reinforcement Learning**    In the multi-objective RL (MORL) setting, agent needs to optimize possibly conflicting objectives with their separate reward function. MORL setting can be modeled as Multi-Objective Markov Decision Process (MOMDP). Unlike the scalar reward function in conventional MDP, the reward function in MOMDP is vector-valued. A MOMDP is defined as $\mathcal{M} = (\mathcal{S}, \mathcal{A}, P, \boldsymbol{r}, p_0, \gamma)$, with state space $\mathcal{S}$ and action space $\mathcal{A}$, dynamics $P(s_{t+1}|, s_t, a_t)$, initial state distribution $p_0(s_0)$, and discount factor $\gamma \in [0, 1)$. The vector-valued function $\boldsymbol{r} : \mathcal{S} \times \mathcal{A} \to \mathbb{R}^m$ is a multi-objective reward function with $m$ objectives. A policy $\pi : \mathcal{S} \to \mathcal{A}$ is a function mapping states to actions. The multi-objective value functions for a policy $\pi$ are:

$$\boldsymbol{q}^\pi(s, a) = \mathbb{E}_\pi \left[ \sum_{i=0}^\infty \gamma^i \boldsymbol{r}(S_{t+i}, A_{t+i}) | S_t = s, A_t = a \right] \tag{1}$$

$$\boldsymbol{v}^\pi(s) = \mathbb{E}_\pi \left[ \sum_{i=0}^\infty \gamma^i \boldsymbol{r}(S_{t+i}, A_{t+i}) | S_t = s \right] \tag{2}$$

Let $\boldsymbol{v}^\pi \in \mathbb{R}^m$ to be the multi-objective value vector of $\pi$ under the initial state distribution $p_0$:

$$\boldsymbol{v}^\pi = \mathbb{E}_{S_0 \sim p_0} \left[ \boldsymbol{q}^\pi(S_0, \pi(S_0)) \right] \tag{3}$$

Each entry of $\boldsymbol{v}^\pi$ is a value for an objective. The Pareto Front is a set of nondominated multi-objective value functions $\mathcal{F} := \{ \boldsymbol{v}^\pi \mid \nexists \pi' \text{ s.t. } \boldsymbol{v}^{\pi'} \succ \boldsymbol{v}^\pi \}$, where $\succ$ is the relation of Pareto dominance such that $\boldsymbol{v}^{\pi'} \succ \boldsymbol{v}^\pi$ means $(\forall i, \boldsymbol{v}_i^{\pi'} \geq \boldsymbol{v}_i^\pi) \wedge (\exists j, \boldsymbol{v}_j^{\pi'} > \boldsymbol{v}_j^\pi)$. Intuitively, if $\boldsymbol{v}^{\pi_1}$ is dominated by $\boldsymbol{v}^{\pi_2}$, then there is no objective where $\pi_1$ performs better so $\pi_2$ is always a better choice than $\pi_1$. An optimal MORL agent should have its value vector on the Pareto front.

**Preference control**    Preference quantifies the trade-off among the multiple objectives. We define the set of preferences $\mathcal{P} := \{ \boldsymbol{p} \in \mathbb{R}^m : \boldsymbol{p}^T \mathbf{1} = 1, \boldsymbol{p} \succ 0 \}$. The desired policy $\pi$ for preference $\boldsymbol{p}$ should have the value $\boldsymbol{v}^\pi$ optimizing a similarity metric $\Psi(\boldsymbol{p}, \boldsymbol{v}^\pi)$, which can be cosine similarity or what we define in Definition 4.2. The optimal $\boldsymbol{v}^\pi$ should be on the Pareto Front with a maximal similarity to $\boldsymbol{p}$. In other words, the ideal $\boldsymbol{v}^\pi$ for preference $\boldsymbol{p}$ should be on the Pareto front and closest to the intersection of between the Pareto front and the ray from the origin to the direction of $\boldsymbol{p}$.

Previous works (Yang et al., 2019; Xu et al., 2020; Alegre et al., 2023) consider maximizing a linear scalarization of objectives $\boldsymbol{p}^T \boldsymbol{v}^\pi$. However, the solution of $\max_\pi \boldsymbol{p}^T \boldsymbol{v}^\pi$ or $\max_\theta \boldsymbol{p}^T \boldsymbol{v}^{\pi_\theta}$ can only be in the convex part of the Pareto front (Boyd and Vandenberghe, 2004, Chapter 4.7) but not the non-convex part. Even for some MORL cases where the Pareto front can be considered convex (Lu et al., 2023), the solution is often limited to a Convex Coverage Set (CCS) that is a subset of Pareto front. Even for strictly convex Pareto front, LS is still not guaranteed to be close to the direction of $\boldsymbol{p}$ as shown by Fig. 1c.

Instead of learning a policy $\pi_{\boldsymbol{p}}$ for each possible $\boldsymbol{p} \in \mathcal{P}$, our goal is to learn an agent with a conditional policy $\pi(a|s, \boldsymbol{p})$ that achieves Pareto optimal values $\boldsymbol{v}^{\pi(\cdot|\cdot, \boldsymbol{p})} \in \arg\max_{\pi'} \Psi(\boldsymbol{p}, \boldsymbol{v}^{\pi'})$ for any $\boldsymbol{p} \in \mathcal{P}$. For conciseness, we denote $\boldsymbol{v}^{\pi(\cdot|\cdot, \boldsymbol{p})}$ as $\boldsymbol{v}^{\pi_{\boldsymbol{p}}}$ in the following text. There are two requirements for the agent. One is to explore the Pareto front as much as possible, and the other is to have a performance trade-off close to the input preference. These two requirements can be evaluated for two metrics: **Hypervolume(HV)** for exploration of Pareto front and **Similarity** $\Psi(\boldsymbol{p}, \boldsymbol{v}^\pi)$ for controllability.

**Multi-objective optimization methods**    Previous Multi-Objective Optimization (MOO) methods deal with how to manipulate gradients from multiple objectives so that updating with the manipulated gradient can reach Pareto optimality. A typical method MGDA (Désidéri, 2009) can guarantee to update in a common ascending direction and stops when the Pareto stationary points are reached. Methods such as CAgrad (Liu et al., 2021) and SDMGrad (Xiao et al., 2023) can provide Pareto optimal solutions by linear scalarization with preference as weights. However, as mentioned above, optimizing linearly scalarized objective with weight $\boldsymbol{p}$ can not guarantee a large similarity $\Psi(\boldsymbol{p}, \boldsymbol{v}^\pi)$.

Methods such as PMTL (Lin et al., 2019) and EPO (Mahapatra and Rajan, 2020) apply similarity constraints to reach the Pareto front with the desired preference, so they can be used for our purpose. In the next section, we show how these methods can be used for learning $\pi(a|s, \boldsymbol{p})$ and how we can make novel improvements to them for the MORL setting.

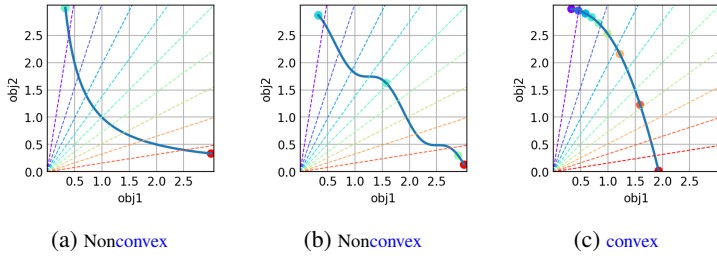

(a) Nonconvex      (b) Nonconvex      (c) convex

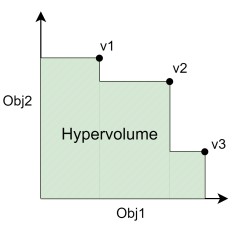

Figure 1: The threes plots show results of optimizing the two objectives using linear scalarization $\max_{\pi_p} \boldsymbol{p}^T \boldsymbol{v}^{\pi_p}$ and its lack of controllability. The blue solid curve is the Pareto front, the colored dotted rays are the preference directions of $\boldsymbol{p}$, and the same colored points are the resulted values $\boldsymbol{v}^{\pi_p}$. The Pareto front in the left two plots are non-convex, while the Pareto front in the right plot is convex. We observe an obvious gap between the preferences and the achieved values in both situations. Linear scalarization can not always achieve an ideal solution in the intersection between the preference rays and the Pareto front. This explains why optimizing a similarity $\Psi(\boldsymbol{p}, \boldsymbol{v}^\pi)$ is necessary for preference control.

Figure 2: Illustration of hypervolume of three value vectors $\boldsymbol{v}_1, \boldsymbol{v}_2, \boldsymbol{v}_3$ for a two objective optimization. Their hypervolume is the volume of the union set of their dominated regions (the green shaded area), reflecting their diversity and coverage.

## 3 LEARNING PREFERENCE CONTROLLABLE AGENT

We propose "Preference control (PC) RL" scheme to address the trade-off between multiple conflicting objectives by training a single agent that can be conditioned on different performance preferences. Conditional preference $\boldsymbol{p}$ controls the agent's emphasis on different objectives and corresponds to a desired point on the Pareto front. We denote the policy conditioned on a preference $\pi(\cdot|\cdot, \boldsymbol{p})$ as $\pi_{\boldsymbol{p}}$. During training, we sample $\boldsymbol{p} \in \mathcal{P}$ uniformly and collect rollout data to estimate $\boldsymbol{v}^{\pi_p}$ and evaluate similarity $\Psi(\boldsymbol{p}, \boldsymbol{v}^{\pi_p})$. Based on the evaluation, we obtain an update direction for $\pi_{\boldsymbol{p}}$. In PCRL scheme, the update direction can be obtained using any methods that can incorporate preference on the objectives, such as linear scalarization (optimizing $\max_{\pi_p} \boldsymbol{p}^T \boldsymbol{v}^{\pi_p}$), or other MOO methods with extra optimization or regularization of the similarity (implementations in Appendix B). We also propose a novel update method while these existing MOO methods will be tested as baselines. In the following section, we first introduce how to estimate the $\boldsymbol{v}^{\pi_p}$ values then explain our proposed update method. We provide theoretical guarantee of our proposed update method in the next section.

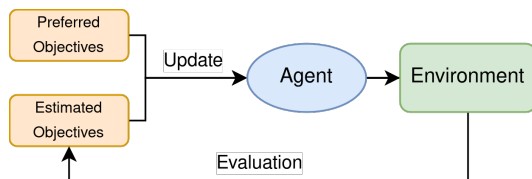

Figure 3: PCRL updates the agent based on its performance and the user preference of objectives.

### 3.1 OBJECTIVE ESTIMATION

Preference control aims to achieve the desired trade-off on conflicting objectives. In the previous RL experiments of MOO methods like Yu et al. (2020); Liu et al. (2021); Xiao et al. (2023), the loss of the value function is used as the objective for MOO, and equal weight is given to all value losses to balance the multi-objectives. While this may be appropriate for RL tasks with minimal conflict, in our setting for preference control, it is essential to align the objective with the preference, so the objective to be aligned with the preference should be $\boldsymbol{v}^{\pi_p}$ itself rather than its approximation loss. Here, we show how to estimate $\boldsymbol{v}^{\pi_p}$ for mainstream RL algorithms.

When learning $\pi_{\boldsymbol{p}}$ with value-based methods like DDPG (Lillicrap et al., 2016), TD3 (Fujimoto et al., 2018) and SAC (Haarnoja et al., 2018), we can estimate $\boldsymbol{v}^{\pi_p}$ by

$$\hat{\boldsymbol{v}}^{\pi_p} = \mathbb{E}_{S_0 \sim p_0} \left[ \boldsymbol{q}_\theta(S_0, \pi_{\boldsymbol{p}}(S_0), \boldsymbol{p}) \right] \quad (4)$$

where $\boldsymbol{q}_\theta$ is multi-objective critic network that outputs a vector of Q-values, it is also conditioned on the prefernece $\boldsymbol{p}$, because $\boldsymbol{p}$ controls the policy $\pi$ thus controlling the value $\boldsymbol{q}^\pi$.

For policy-based methods such as A3C Mnih et al. (2016), PPO Schulman et al. (2017), they update with a whole episode so $\boldsymbol{v}^{\pi_p}$ can be estimated by episodic returns.

$$\hat{\boldsymbol{v}}^{\pi_p} = \mathbb{E}_{S_0 \sim p_0} \left[ \sum_{t=0}^{T} \gamma^t \boldsymbol{r}(S_t, A_t) \right] \tag{5}$$

As a result, our scheme is applicable to both discrete action space and continuous action space. With the estimated value vector $\hat{\boldsymbol{v}}^{\pi_p}$, we can evaluate the similarity $\Psi(\boldsymbol{p}, \hat{\boldsymbol{v}}^{\pi_p})$.

## 3.2 UPDATING PROCEDURE

After estimating objective values and similarity for preference control, we need to manipulate the gradients from different objectives and update the agent using the manipulated gradient. Our scheme has the following updating procedure:

1. Get the Jacobian matrix $\nabla_{\pi_p} \hat{\boldsymbol{v}}^{\pi_p}$:

    Each row of the Jacobian matrix $\nabla_{\pi_p} \hat{\boldsymbol{v}}^{\pi_p}$ is a gradient for one objective. The gradient can be obtained by conventional RL methods, such as the policy gradient and the deterministic policy gradient. An illustrative diagram (Fig. 9 in Appendix D) provides intuition and shows how to estimate $\nabla_{\pi_p} \hat{\boldsymbol{v}}^{\pi_p}$ for different RL algorithms.

2. Get similarity gradient $\nabla_{\pi_p} \Psi(\boldsymbol{p}, \hat{\boldsymbol{v}}^{\pi_p})$:

    $\Psi$ measures the similarity between preference $\boldsymbol{p}$ and the multi-objective vector $\mathbf{J}$. For evaluation, cosine similarity is good enough to measure how close the value vector is to the preference. However, when gradient $\nabla_{\pi_p} \Psi(\boldsymbol{p}, \hat{\boldsymbol{v}}^{\pi_p})$ is used for our updates, the resulting manipulated gradient update is expected to not only keep the similarity large but also converge to the Pareto front. To this end, we propose a novel design of the similarity function with theoretical analysis in the next section.

3. Manipulate the gradients and find the optimal update direction $\boldsymbol{d}^*$ by solving:

    $$w^* \in \arg\min_{\boldsymbol{w}} \|\boldsymbol{d}\|, \;\; \boldsymbol{d} \triangleq \nabla_{\pi_p}^T \hat{\boldsymbol{v}}^{\pi_p} \boldsymbol{w} + \lambda \nabla_{\pi_p} \Psi(\boldsymbol{p}, \hat{\boldsymbol{v}}^{\pi_p}) \tag{6}$$

    $$\boldsymbol{d}^* = \nabla_{\pi_p}^T \hat{\boldsymbol{v}}^{\pi_p} \boldsymbol{w}^* + \lambda \nabla_{\pi_p} \Psi(\boldsymbol{p}, \hat{\boldsymbol{v}}^{\pi_p}) \tag{7}$$

    This is a min-norm problem similar to MGDA (Désidéri, 2009) and SMGrad (Xiao et al., 2023), but it adds a similarity gradient to every objective gradient, making the update not only ascent in a common improving direction but also closing the value $\boldsymbol{v}^{\pi_p}$ to the preference $\boldsymbol{p}$. We call this update *PREference COntrol(PreCo)* update. Updating with $\boldsymbol{d}^*$ converges to where $\|\boldsymbol{d}^*\| = 0$, indicating no common improving direction exists thus satisfying Pareto stationary. We will prove the convergence of this gradient under our proposed similarity function.

This is a general update procedure that can employ any RL algorithm for the calculation of the objective gradients $\nabla_{\pi_p} \hat{\boldsymbol{v}}^{\pi_p}$. In the third step, the gradient manipulation can also be performed by various MOO methods. In the experiment, we examine our scheme with PreCo update against the baselines with existing gradient manipulation methods such as EPO (Mahapatra and Rajan, 2020) CAGrad (Liu et al., 2021). Computationally, the min-norm problem in the third step is solved at the policy level with $\nabla_{\pi_p} \hat{\boldsymbol{v}}^{\pi_p}$ instead of the parameter level with $\nabla_\theta \hat{\boldsymbol{v}}^{\pi_p}$. The size for a sample of $\nabla_{\pi_p} \hat{\boldsymbol{v}}^{\pi_p}$ is only $m \times B$ for a batch of $B$ transitions, while $\nabla_\theta \hat{\boldsymbol{v}}^{\pi_p}$ of size $m \times M$ could have a parameter size $M \gg m$. $M$ can even be billions for large models. For those cases, solving the min-norm problem at the parameter level could be memory-inefficient and computationally intractable. A pseudo-code for the PCRL scheme with PreCo update and more details on the definitions of policy level gradients and parameter level gradients can be found in Appendix D.

## 4 THEORETICAL ANALYSIS

In this section, we provide the formal definition of our proposed similarity function $\Psi(\cdot, \cdot)$ and the theoretical analysis for the PreCo update. We will prove that it converges to Pareto stationary points, and the resulting similarity $\Psi(\boldsymbol{p}, \boldsymbol{v}^{\pi_p})$ will also converge to stationary points.

**Definition 4.1.** *We define our similarity function as follows*

$$\Psi(\boldsymbol{p}, \boldsymbol{v}) = -\frac{1}{2}\| \max_i \frac{\boldsymbol{v}_i}{\boldsymbol{p}_i}\boldsymbol{p} - \boldsymbol{v}\|^2 \tag{8}$$

Intuitively, the similarity gradient $\nabla_{\boldsymbol{v}}\Psi(\boldsymbol{p}, \boldsymbol{v})$ encourages to focus on the less optimal objectives to reach the preference $\boldsymbol{p}$. A visualization for $\Phi(\boldsymbol{p}, \cdot)$ can be found in Appendix E.

Deep reinforcement learning is inherently stochastic and sensitive to sample complexity. Therefore, we analyze the convergence rate of the proposed PreCo update in the stochastic gradient setting. The PreCo algorithm that we analyze in this case is Algorithm 1.

---

**Algorithm 1** PreCo in the theoretical analysis setting

---

**Initialize:** Preference $\boldsymbol{p}$, preference-conditioned policy $\pi_{\boldsymbol{p}}$, and weights $\boldsymbol{w}_0$
**for** $t = 0, 1, ..., T-1$ **do**
   Rollout and estimate the value to get data $\xi, \xi', \zeta$
   $\boldsymbol{w}_t = \Pi_{\mathcal{W}}\left(\boldsymbol{w}_{t-1} - \beta_t[G(\pi_{\boldsymbol{p},t}; \xi)^T(G(\pi_{\boldsymbol{p},t}; \xi')\boldsymbol{w}_{t-1} + \lambda_t g_s(\pi_{\boldsymbol{p}}; \xi'))]\right)$
   $\pi_{\boldsymbol{p},t+1} = \pi_{\boldsymbol{p},t} + \alpha_t\left(G(\pi_{\boldsymbol{p},t}; \zeta)\boldsymbol{w}_{t-1} + \lambda_t g_s(\pi_{\boldsymbol{p},t}; \zeta)\right)$
**end for**

---

$\boldsymbol{w}$ is the one defined in Equation (6) and $\Pi_{\mathcal{W}}$ means the projection to the set $\mathcal{W} \coloneqq \{\boldsymbol{w} \in \mathbb{R}^m : \boldsymbol{w}^T\mathbf{1} = 1, \boldsymbol{w} \succ 0\}$. Data $\xi, \xi', \zeta$ are different transition samples used to estimate gradient $\nabla_{\pi_{\boldsymbol{p}}}\hat{\boldsymbol{v}}^{\pi_{\boldsymbol{p}}}$. For conciseness, we denote

$$G(\pi_{\boldsymbol{p}}) = \mathbb{E}[G(\pi_{\boldsymbol{p}}; \xi)] = \nabla_{\pi_{\boldsymbol{p}}}^T \boldsymbol{v}^{\pi_{\boldsymbol{p}}} = \mathbb{E}[\nabla_{\pi_{\boldsymbol{p}}}^T \hat{\boldsymbol{v}}^{\pi_{\boldsymbol{p}}}], \tag{9}$$

where the expectation is taken w.r.t. data $\xi$, the $i$th column of $G(\pi_{\boldsymbol{p}}; \xi)$ is the gradient of $i$th objective and $g_s(\pi_{\boldsymbol{p}})$ is the similarity gradient

$$g_s(\pi_{\boldsymbol{p}}) = \mathbb{E}[g_s(\pi_{\boldsymbol{p}}, \xi)] = G(\pi_{\boldsymbol{p}})\nabla_{\boldsymbol{v}}\Psi(\boldsymbol{p}, \boldsymbol{v}^{\pi_{\boldsymbol{p}}}) = \mathbb{E}\left[G(\pi_{\boldsymbol{p}}; \xi)\nabla_{\boldsymbol{v}}\Psi(\boldsymbol{p}, \hat{\boldsymbol{v}}^{\pi_{\boldsymbol{p}}})\right]. \tag{10}$$

Algorithm 1 is only for theoretical analysis; In practice, the weight $\boldsymbol{w}$ does not need to be updated only once every iteration but can be fully optimized for the min-norm problem (6) and a more practical Algorithm 2 is provided in Appendix D.

### 4.1 CONVERGENCE ANALYSIS

First, we define what Pareto stationary is:

**Definition 4.2.** *We define $\pi$ is an $\epsilon$-accurate Pareto stationary policy if $\mathbb{E}[\min_{\boldsymbol{w}} \|G(\pi_{\boldsymbol{p}})\boldsymbol{w}\|] \leq \epsilon$, where $\boldsymbol{w}$ is a convex coefficient.*

We assume the continuity and smoothness of the objectives.

**Assumption 4.1.** *For every objective $i \in [m]$, $\boldsymbol{v}_i(\pi_{\boldsymbol{p}})$ is $l_i$-Lipschitz continuous and $\nabla\boldsymbol{v}_i(\pi_{\boldsymbol{p}})$ is $l_{i,1}$-Lipschitz continuous for any preference conditioned policy $\pi_{\boldsymbol{p}}$.*

This assumption is quite common in RL setting. By the "branched returns bound" in Janner et al. (2019),

$$|\boldsymbol{v}_i(\pi_1) - \boldsymbol{v}_i(\pi_2)| \leq 2r_{\max,i}\left(\frac{\gamma\epsilon_{\pi}}{(1-\gamma)^2} + \frac{\epsilon_{\pi}}{1-\lambda}\right), \tag{11}$$

where $r_{\max,i} = \max_{s,a} r_i(s, a)$ and $\epsilon_{\pi}$ can be any scalar satisfying $\epsilon_{\pi} \geq \max_s D_{TV}(\pi_1(\cdot|s), \pi_2(\cdot|s))$. Because

$$\max_s D_{TV}(\pi_1(\cdot|s), \pi_2(\cdot|s)) \leq D_{TV}(\pi_1, \pi_2) = \frac{1}{2}|\pi_1 - \pi_2|, \tag{12}$$

we can derive

$$|\boldsymbol{v}_i(\pi_1) - \boldsymbol{v}_i(\pi_2)| \leq r_{\max,i}\left(\frac{\gamma}{(1-\gamma)^2} + \frac{1}{1-\lambda}\right)|\pi_1 - \pi_2|, \tag{13}$$

and $L_i$ can be $r_{\max,i}\left(\frac{\gamma}{(1-\gamma)^2} + \frac{1}{1-\lambda}\right)$. Therefore, the Lipschitz continuity of objectives is naturally satisfied for conventional RL settings, and we only need to assume the gradients are also Lipschitz continuous.

Next, we make an assumption on the bias and variance of the stochastic gradient $g_i(\pi; \xi)$.

**Assumption 4.2.** *For every objective $i \in [m]$, the gradients $g_i(\pi_{\boldsymbol{p}}; \xi)$ is unbiased estimate of $g_i(\pi_{\boldsymbol{p}})$, and the variances is bounded by $\mathbb{E}_\xi[\|g_i(\pi_{\boldsymbol{p}}; \xi) - g_i(\pi_{\boldsymbol{p}})\|^2] \leq \sigma^2$.*

We also assume bounded gradient.

**Assumption 4.3.** *There exists a constant $C_g$ such that $\|G(\pi_{\boldsymbol{p}})\| \leq C_g$.*

**Lemma 4.1.** *The similarity function $\Psi(\boldsymbol{p}, \cdot)$ is $(1 + \max_i \frac{|\boldsymbol{p}|}{|\boldsymbol{p}_i|})$ -Lipschitz smooth and $g_s(\cdot)$ is Lipschitz continuous under Assumption 4.1 and Assumption 4.3.*

This lemma shows that our proposed similarity function is Lipschitz smooth. The detailed proof is in Appendix I.1. PreCo and SDMgrad (Xiao et al., 2023) both belong to MGDA-variant methods that solve a min-norm problem for gradient manipulation. Leveraging the fact that $g_s(\pi_{\boldsymbol{p}})$ is a positive linear combination of $G(\pi_{\boldsymbol{p}})$ and the Lipschitz smoothness property, we can therefore build upon their results to prove that PreCo converges to Pareto stationary points.

**Theorem 4.1.** *Under the Assumptions 4.1-4.3, setting $\alpha_t = \Theta(m^{-\frac{1}{2}} T^{-\frac{1}{2}})$, $\beta_t = \Theta(m^{-1} T^{-\frac{1}{2}})$, with a constant $\lambda$ and Lipschitz smooth similarity function $\Psi(\boldsymbol{p}, \cdot)$, we have $\frac{1}{T} \sum_{t=0}^{T-1} \mathbb{E}[\min_{\boldsymbol{w}_t} \|G(\pi_{\boldsymbol{p},t}) \boldsymbol{w}_t\|] = \mathcal{O}(m T^{-\frac{1}{2}})$. To achieve an $\epsilon$-accurate Pareto stationary point, it requires $T = \mathcal{O}(m^2 \epsilon^{-2})$ updates.*

Theorem 4.1 shows PreCo converges to Pareto stationary points when $\lambda$ is a constant. This theorem applies to our proposed similarity function $\Psi(\boldsymbol{p}, \cdot)$.

**Theorem 4.2.** *Under the Assumptions 4.1-4.3, setting $\alpha_t = \Theta(m^{-\frac{1}{2}} T^{-\frac{1}{2}})$, $\beta_t = \Theta(m^{-1} T^{-\frac{1}{2}})$, with a Lipshitz smooth similarity function with $g'_s(\pi_{\boldsymbol{p},t})$ being convex combination of $g_i(\pi_{\boldsymbol{p},t})$ for all $t$, there can be an increasing $\lambda = \Theta(\log T)$ and we have $\frac{1}{T} \sum_{t=0}^{T-1} \mathbb{E}[\min_{\boldsymbol{w}_t} \|G(\pi_{\boldsymbol{p},t}) \boldsymbol{w}_t\|] = \mathcal{O}(m T^{-\frac{1}{2}} \log T)$.*

Theorem 4.2 consider a case requiring similarity gradient to be a convex combination of objective gradients, of which its design is discussed in Appendix E.2. In this case $\lambda$ can increase without an upper limit and eventually $g_s$ will dominate the min-norm solution of (6). Proofs are in Appendix I.2.

**Remark 4.1.** *In practice, Theorem 4.1 still applies to cases where $\lambda$ increases but with an upper limit. Because after $\lambda$ gets close to the limit, it can be considered constant. This offers theoretical justification for implementing PreCo with $\Psi(\boldsymbol{p}, \cdot)$ and an increasing $\lambda$.*

**Remark 4.2.** *The convergence rate for Theorem 4.2 seems slower than results from Xiao et al. (2023) because we rigorously considered the changes in the Lipschitz constant of the $(G(\pi_{\boldsymbol{p},t}) \boldsymbol{w}_t + \lambda g_s(\pi_{\boldsymbol{p},t}))$ caused by increasing $\lambda$*

### 4.2 CONTROLLABILITY ANALYSIS

Controllability in our setting is the similarity between the desired preference $\boldsymbol{p}$ and the value $\boldsymbol{v}^{\pi_{\boldsymbol{p}}}$ of the preference-conditioned policy $\pi_{\boldsymbol{p}}$. It is measured by $\Psi(\boldsymbol{p}, \boldsymbol{v}^{\pi_{\boldsymbol{p}}})$. We provide the following results to show how $\boldsymbol{v}^{\pi_{\boldsymbol{p}}}$ will converge to the point close to the $\boldsymbol{p}$ direction.

**Theorem 4.3.** *Under the Assumptions 4.1-4.3, setting $\alpha_t = \Theta(m^{-\frac{1}{2}} T^{-\frac{1}{2}})$, $\beta_t = \Theta(m^{-1} T^{-\frac{1}{2}})$, with a constant $\lambda$ and Lipschitz smooth similarity function like $\Psi(\boldsymbol{p}, \cdot)$, we have $\frac{1}{T} \sum_{t=0}^{T-1} \mathbb{E}[\|g_s(\pi_{\boldsymbol{p}})\|] - \frac{2 C_g^2}{\lambda^2} = \mathcal{O}(m T^{-\frac{1}{2}})$.*

Theorem 4.3 provides an intuitive result, that with constant $\lambda$, the norms of the similarity gradient $\frac{1}{T} \sum_{t=0}^{T-1} \mathbb{E}[\|g_s(\pi_{\boldsymbol{p}})\|]$ will converge and be bounded. The larger $\lambda$, the lower the bound $\frac{2 C_g^2}{\lambda^2}$, and the closer the solution will reach the stationary points for maximizing similairty.

**Theorem 4.4.** *Under the Assumptions 4.1-4.3, setting $\alpha_t = \Theta(m^{-\frac{1}{2}} T^{-\frac{1}{2}})$, $\beta_t = \Theta(m^{-1} T^{-\frac{1}{2}})$, with a constant $\lambda$ and Lipschitz smooth similarity function like $\Psi(\boldsymbol{p}, \cdot)$, there can be an increasing $\lambda = \Theta(T^{\frac{1}{2}})$ and we have $\frac{1}{T} \sum_{t=0}^{T-1} \mathbb{E}[\|g_s(\pi_{\boldsymbol{p}})\|] = \mathcal{O}(m T^{-\frac{1}{2}} \log T)$.*

Theorem 4.4 shows PreCO with increasing $\lambda$ will converge to the stationary points for the similarity objective. The proofs of the theorems are in Appendix I.3.

**Remark 4.3.** *Similar to Theorem 4.1, Theorem 4.3 applies to practical implementations where $\lambda$ increases but with an upper limit.*

**Remark 4.4.** *The converged stationary points do not guarantee to have always high similarity metrics. For example, when using $\Psi(\boldsymbol{p}, \cdot)$, our results show $g_s(\pi_{\boldsymbol{p}}) = G(\pi_{\boldsymbol{p}})\nabla_{\boldsymbol{v}}\Psi(\boldsymbol{p}, \boldsymbol{v}^{\pi_{\boldsymbol{p}}})$ converges to $\boldsymbol{0}$. However, the value $\boldsymbol{v}^{\pi_{\boldsymbol{p}}}$ coincide with the preference $\boldsymbol{p}$ only when $\|\nabla_{\boldsymbol{v}}\Psi(\boldsymbol{p}, \boldsymbol{v}^{\pi_{\boldsymbol{p}}})\| = 0$. $\|g_s(\pi_{\boldsymbol{p}})\|$ can also be 0 when $\|\nabla_{\boldsymbol{v}}\Psi(\boldsymbol{p}, \boldsymbol{v}^{\pi_{\boldsymbol{p}}})\| > 0$, with $G(\pi_{\boldsymbol{p}})$ and $\nabla_{\boldsymbol{v}}\Psi(\boldsymbol{p}, \boldsymbol{v}^{\pi_{\boldsymbol{p}}})$ being orthogonal or $G(\pi_{\boldsymbol{p}}) = \boldsymbol{0}$. These situations means the points desired by the preference might not exist on the Pareto front. We discuss in practice how to deal with unreachable regions of Pareto front in Appendix H.*

The theoretical results show that PreCo can discover not only Pareto stationary solutions but also preference-specific solutions. We used a 2-D MOO example in Appendix F to demonstrate that PreCo can find preference-specific solutions for general stochastic MOO.

## 5 EXPERIMENTS

In the experiment, we want to empirically answer the following questions:

1. Can the agent have non-dominated performance for different preferences?
2. How controllable is it for unseen preference trade-offs?
3. Is our method scalable for larger number of objectives?
4. Can our method be used for both discrete action and continuous action environments?

**Benchmarking Environments** To answer the questions, we need challenging environments with conflicting objectives, so that there is a trade-off on the emphasis of different objectives. Also, we need environments with more than just two objectives. In addition, we need environments with both discrete action space and continuous action space. We use TD3 for continuous and PPO-clip for discrete actions. We chose to conduct experiments on the following four MORL environments.

- **Fruit-Tree:** A discrete environment. Every leaf contains a fruit with a 6-D reward for the nutrients Protein, Carbs, Fats, Vitamins, Minerals, and Water.
- **MO-Ant:** A higher dimensional continuous robotic control environment with 2-D reward of x-velocity and y-velocity.
- **MO-Hopper:** A continuous robotic control environment with 2-D reward. The first dimension is for going forward on the x-axis and the other for jumping high on the z-axis.
- **MO-Reacher:** A robotic control environment with continuous state space and discrete action space. The reward is 4-D and is defined based on the distance of the tip of the arm and the four target locations.

**Evaluation metrics** We evaluated the results using two metrics: **hyperVolume(HV)** for Pareto front exploration and **Cosine Similarity(CS)** for controllability evaluation. They are measured in test time with preference samples unseen in training (Appendix C). We report the mean and standard deviation results of 5 seeds.

**Baselines** We compare PreCo with existing MOO gradient manipulation methods in PCRL scheme:

- **Linear Scalarization (LS)** Optimize the linearly scalarized objective and the manipulated gradient will be simply the gradient linearly weighted by the preference $(\nabla_{\pi_{\boldsymbol{p}}} \hat{\boldsymbol{v}}^{\pi_{\boldsymbol{p}}})^T \boldsymbol{p}$. This is the most common update approach in previous MORL methods. Works like Yang et al. (2019); Xu et al. (2020); Alegre et al. (2023) can all be categorized as LS variations.
- **Exact Pareto Optimal (EPO) Search** MOO methods such as PMTL (Lin et al., 2019) and EPO (Mahapatra and Rajan, 2020) apply similarity constraints and have two modes for situations of low and high similarity. We implement it as updating with similarity gradient for low similarity mode, MGDA (Désidéri, 2009) gradient for high similarity mode.
- **Conflict-Averse Gradient (CAGrad)** CAGrad tries to find a common ascent direction that is not too far from the average gradient. In our setting, we modify it to be a common ascent direction not is not too far from the similarity gradient.

- **Stochastic Direction-oriented MO-Gradient (SDMGrad)** Similar to PreCo, SDMGrad solves a variant of the min-norm problem (6) for gradient manipulation. Instead of adding objective gradients with similarity gradients like PreCO, they add a convex combination of objective gradients. Therefore, SDMGrad can serve as a baseline to show whether our proposed similarity function $\Psi$ contributes to empirical performance for ablation study.

These baselines can be categorized into the linear scalarization approach, such as LS and SDMGrad, and the similarity approach, such as EPO and CAgrad. PreCo belongs to the similarity approach. More details of the baseline implementations can be found in the Appendix B.

## 5.1 FRUIT TREE

Fruit tree is a discrete environment, we used PPO-clip in the MORL scheme. We test our method and the baselines with rewards from 3-6 dimensions. The results of HV and CS are shown in Table 1.

| Method | 3D | 4D | 5D | 6D |
|---|---|---|---|---|
| LS | $0.12 \pm 0.01 \mid 0.78 \pm 0.03$ | $0.33 \pm 0.13 \mid 0.76 \pm 0.05$ | $1.59 \pm 0.29 \mid 0.74 \pm 0.01$ | $5.74 \pm 0.88 \mid 0.78 \pm 0.01$ |
| SDMgrad | $0.14 \pm 0.01 \mid 0.78 \pm 0.03$ | $0.66 \pm 0.02 \mid 0.72 \pm 0.00$ | $2.74 \pm 0.09 \mid 0.66 \pm 0.01$ | $13.30 \pm 0.15 \mid 0.72 \pm 0.01$ |
| EPO | $\mathbf{0.15 \pm 0.01} \mid \mathbf{0.84 \pm 0.02}$ | $1.04 \pm 0.05 \mid 0.89 \pm 0.02$ | $3.98 \pm 0.48 \mid 0.86 \pm 0.03$ | $14.97 \pm 2.29 \mid 0.77 \pm 0.03$ |
| CAGrad | $0.14 \pm 0.02 \mid 0.78 \pm 0.02$ | $0.30 \pm 0.06 \mid 0.87 \pm 0.01$ | $1.23 \pm 0.14 \mid 0.69 \pm 0.01$ | $4.93 \pm 0.81 \mid 0.60 \pm 0.09$ |
| PreCo(Ours) | $\mathbf{0.15 \pm 0.01} \mid \mathbf{0.84 \pm 0.02}$ | $\mathbf{1.09 \pm 0.02} \mid \mathbf{0.91 \pm 0.01}$ | $\mathbf{4.33 \pm 0.21} \mid \mathbf{0.87 \pm 0.01}$ | $\mathbf{15.61 \pm 0.75} \mid \mathbf{0.78 \pm 0.03}$ |

Table 1: "HV|CS" (higher is better for both) in fruit-tree environment with HV in the scale of $10^3$. Our method consistently achieves the best optimality (HV) and controllability (CS) from 3-6 objectives.

Results show our proposed PreCo perform better especially for higher reward dimensions. Also, similarity approach methods perform better than linear scalarization methods. One major reason for this can be illustrated by the following example 3-dimensional reward case.

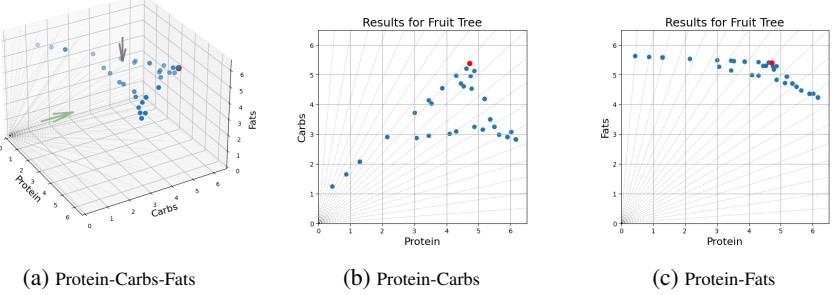

(a) Protein-Carbs-Fats  (b) Protein-Carbs  (c) Protein-Fats

Figure 4: (a) shows the 3-D values $v^{\pi_p}$ achieved under difference preference input $p$. Blue points are $v^{\pi_p}$ of PreCo while LS only learns the red point for most random seeds(even with different $p$ input). **All shown values are non-dominated by others**; (b) shows the Protein-Carbs view from the grey arrow's perspective in (a); and (c) shows the Protein-Fats view from the green arrow in (a).

In the case shown in Fig. 4, LS agent has only one constant $v^{\pi_p}$ at the red point, regardless of the preference input, which indicates that the LS agent policy is not uncontrollable by $p$ despite being conditioned by $p$. Although this red point dominates many the blue points ($v^{\pi_p}$ of different $p$ for PreCo agent) for Proteins and Carbs (Fig. 4b), there are blue points with better values for Fats (Fig. 4c).

## 5.2 MO-ANT

The MO-Ant is a challenging environment with an 8-dimensional continuous action space and a 27-dimensional continuous observation space. The reward is 2-dimensional, with one for x-velocity and one for y-velocity. Although the robotic agent has more complex dynamics, the objectives appear to be very similar since both involve the movement of the agent, making it a relatively easy for preference control. As shown in Fig. 5, the Pareto front has an intuitive convex shape. The preferences

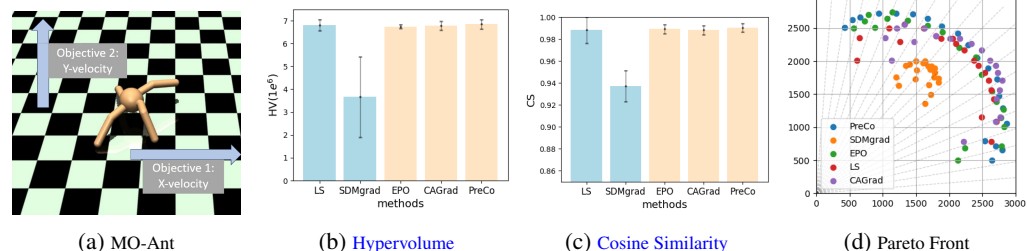

(a) MO-Ant     (b) Hypervolume     (c) Cosine Similarity     (d) Pareto Front

Figure 5: Optimality (HV) and Controllability (CS) of MO-Ant. PreCo (ours) achieves the best CS. Though being the second best on HV, it achieves the widest spread on the Pareto front in (d). In (b)-(c), methods of blue bars are based on linear scalarization, while methods of orange bars optimize the similarity.

$p$ are 20 directions: $[0.05, 0.95], [0.1, 0.9], ..., [1, 0]$. The similarity approach methods have high CS metrics over 0.98. This indicates that our proposed PCRL scheme with similarity optimization works for Pareto front discovery and preference control.

## 5.3 MO-HOPPER

The MO-Hopper is a classic continuous robotic control environment, with one objective rewards for going forward in the x-axis, and the other rewards for jumping high in the z-axis as shown in Fig. 6a. The two objectives are less symmetric than MO-Ant and there is a clear trade-off in directions.

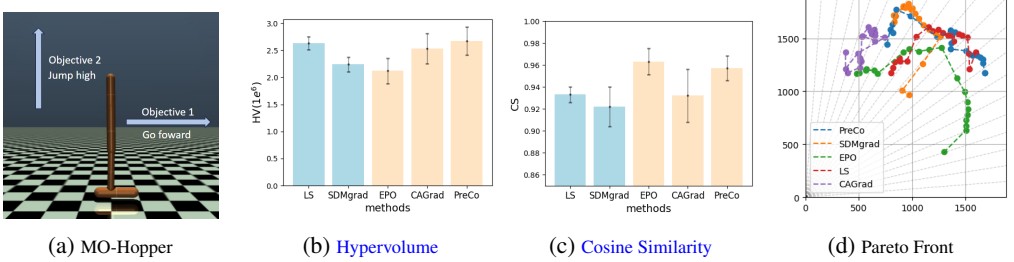

(a) MO-Hopper     (b) Hypervolume     (c) Cosine Similarity     (d) Pareto Front

Figure 6: Optimality (HV) and Controllability (CS) of MO-Hopper.

The HV and CS results are shown in Fig. 6. Our method demonstrated superior performance in HV, while its CS was only slightly lower than that of EPO. Unlike EPO, which employs hard constraints on similarity, our proposed PreCo utilizes soft constraints. This could be the reason why PreCo can sacrifice a small degree of controllability for a significant enhancement in optimality. The hopper has to be able to jump before jumping forward, this is why objective 2 is higher than objective 1 for most methods. As a result, the asymmetric objectives make the discovered Pareto front not as symmetric as that of MO-Ant. We have a calibration approach to further improve controllability in Appendix H.

## 5.4 MO-REACHER

The MO-Reacher is a challenging environment with 4-dimensional rewards for reaching four targets shown in Fig. 7a, and 9-dimensional discrete actions. The reward is depend on the L2 distance from the tip of the robotic agent's arm to 4 target locations at $[0.14, 0], [0, 0.14], [-0.14, 0], [0, 0.14]$, higher rewards for lower distances. PPO is used for PCRL.

Fig. 7 shows the quantitative results of HV and CS and Fig. 8 shows the state coverage of robotic arm tip positions. The horizontal blue dotted line in Fig. 7 is the performance metric of the randomly initialized agent. Its HV and CS serve as a reference for comparison. We can see that EPO and our PreCo higher CS than random, indicating that their preference controllability improved after training. Fig. 8b shows their state coverage controlled by 4 different $p$. From left to right they are $[0, 1, 0, 0]$, $[0, 0.66, 0.33, 0], [0, 0.33, 0.66, 0], [0, 0, 1, 0]$. The preference $[0, 1, 0, 0]$ means full focus on closing to top target $[0, 0.14]$, while $[0, 0, 1, 0]$ fully focus for left target $[-0.14, 0]$. The first row is EPO and the second is PreCo. Their state coverage can be smoothly controlled from more density to the top to

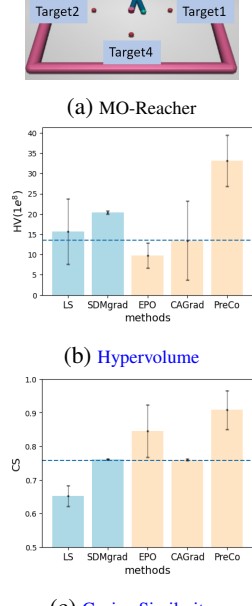

(a) MO-Reacher

(b) Hypervolume

(c) Cosine Similarity

Figure 7: HV and CS of MO-Reacher. The dotted line is the performance of a randomly initialized agent as a reference.



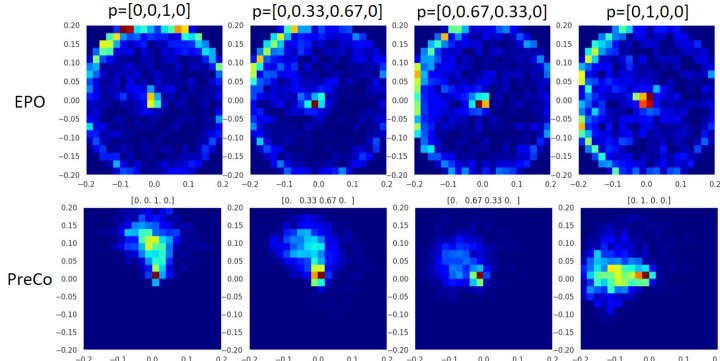

(a) State coverage of Random, LS, SDMgrad, and CAGrad (left to right). Each state coverage heatmap is the same for different preference $p$.

(b) State coverage heatmaps of EPO (top row) and PreCo (bottom row) under four different preferences $p$ (columns). It shows that they produce preference-specific policies given different $p$.

Figure 8: State coverage heatmaps for the positional states of the tip of the robotic arm. Red means higher density of coverage and blue means lower density. EPO and PreCo exhibit different state coverage controlled by different $p$, while random, LS, SDMgrad, and CAGrad show the same state coverage for different preferences.

more density to the left. In contrast, LS/SDMgrad/CAgrad have almost the same CS as the random agent because they only learn one universal policy for all preferences, and their state coverages are shown in Fig. 8a. We can see that their state coverages have uniform density for all directions, which is not preference specific, so they are not controllable by $p$.

Four different, conflicting directions can lead to highly contradictory objective gradients. Linear scalarization methods (like LS/SDMGrad) can only learn a local optimum for all objectives, that is, stay at the origin (as shown in the 2nd and 3rd plots in Fig. 8a). Due to conflicting objective gradients, uniformly sampling $p$ during training may introduce high variance in the gradients for model update, which could be the reason why EPO and CAGrad exhibit worse HV than random.

In summary of the empirical results, our proposed PCRL scheme is effective in searching for Pareto-optimal solutions that are controllable by preferences. The scheme is also scalable to larger numbers of objectives, where LS often fails. Overall, methods of the similarity approach exhibit superior preference controllability than linear scalarization approach. Our proposed PreCo consistently delivers strong performance across all environments, particularly in cases with a high number of objectives or conflicting objectives. Additionally, PreCo constantly outperforms SDMgrad, which serves as an ablation case of PreCo (using linear scalarization), highlighting the critical role of similarity optimization. More details of the empirical results can be found in Appendix C.

## 6 CONCLUSION

In this work, we proposed a PCRL scheme for preference control of the multi-objective trade-offs. To achieve this, we proposed a novel MOO approach, PreCo, and provided a comprehensive convergence analysis for stochastic optimization with non-convex smooth objective functions. Our experiments across multiple RL environments show that our proposed method consistently outperforms baselines, demonstrating that a meta-policy that adapts to user preferences is both feasible and promising for potential applications in human-interactive AI agents. Future work for improving PCRL could be progressing training preference distribution for curriculum learning.

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

# A    RELATED WORKS

## A.1    META-POLICIES

Existing MORL methods that learn a similar meta-policy conditioned on a weight or preference include Abels et al. (2019); Chen et al. (2019); Lu et al. (2023), of which Abels et al. (2019); Lu et al. (2023) are LS methods and they care more about discovering all Pareto optimal policies rather than the similarity between the weight input and the resulted value. Chen et al. (2019) employs a setting most similar to our PCRL since they optimize a Tchebycheff Scalarized (TS) (Ehrgott, 2005) objective for solutions aligned with the preference directions. However, Xu et al. (2020) reported that Chen et al. (2019) has suboptimal performance in practice. This might be due to the oscillation and stagnation issue inherent in the TS approach, as noted by Mahapatra and Rajan (2021). In particular, for MORL, which is sensitive to stochasticity and conflicting gradients, the convergence of TS can be problematic. Our baseline implementation of EPO with a small constraint threshold can be considered as a version of TS tailored to the MORL setting, designed to mitigate oscillation when near preference direction. More recent MORL algorithm (Basaklar et al., 2023) incorporates cosine similarity for preference alignment but it is designed specifically for off-policy value-based RL and its performance relies heavily on the HER (Andrychowicz et al., 2017) technique. In contrast, our PCRL with PreCo is a broader MORL framework compatible with both on-policy and off-policy RL, capable of learning quality policies without HER. Nonetheless, HER can still be integrated into PCRL with PreCo in off-policy settings to enhance sample efficiency. While not targeting exact preference alignment, Lu et al. (2023) addressed LS limitations by adding a concave augmentation term to the reward, transforming the original Pareto front into a strictly convex one. However, this introduces information loss, making the approach sensitive to the augmentation term's magnitude. Their implementation is limited to SAC, using policy entropy as the augmentation term. Compared to Lu et al. (2023), our method possesses the theoretical advantages of no information loss and exact preference alignment, which can be empirically demonstrated by our additional experiment in Appendix G.

Other RL paradigms employing meta-policies include Goal-Conditioned RL (GCRL) (Sekar et al., 2020; Yang et al., 2022; Liu et al., 2022) and skill-based RL (SBRL) (Nam et al., 2022; Lee et al., 2019). GCRL is controlled by an additional input of a target state that it aims to reach. SBRL is conditioned by a skill latent $z$ that often has a lower dimension than the state for a specific primitive skill. Similar to SBRL, the skill learning methods of Unsupervised Reinforcement Learning (Eysenbach et al., 2018; Hansen et al., 2020) learns skills without external task rewards by optimizing a Mutual Information Skill Learning (MISL) (Eysenbach et al., 2021; Yang et al., 2024) objective $I(s; z) = H(s) - H(s|z)$. Maximizing $I(s; z)$ encourages the state space coverage to be high and the state distribution to be certain when controlled by a skill $z$. The concept of Preference Control (PC) has a resemblance to optimizing $I(s; z) = H(s) - H(s|z)$. The purpose of PCRL can also be interpreted as optimizing $I(\boldsymbol{v}; \boldsymbol{p}) = H(\boldsymbol{v}) - H(\mathbf{v}|\mathbf{p})$ to encourage diverse values on the Pareto front and the distribution of the values needs to be controlled by preference $\boldsymbol{p}$.

## A.2    MULTI-OBJECTIVE OPTIMIZATION

We have already introduced PMTL and EPO (Lin et al., 2019; Mahapatra and Rajan, 2020) that could find preference-specific solutions and CAGrad, SDMGrad (Liu et al., 2021; Xiao et al., 2023) that optimize for the average objective but can deal with conflicting gradients. Désidéri (2009); Xiao et al. (2023) has the most similarity to our proposed PreCo because they all solve a min norm problem like 6 for gradient manipulation. The advantage of our PreCo is not only like SDMGrad, which can provably deal with stochastic gradients, but also can follow a preference like EPO. Our theoretical analysis of PreCo is based on some results from Xiao et al. (2023), but incorporating the similarity gradient from $\Psi$ makes it more complicated and novel.

## A.3    TRAINING SCHEMES

We uniformly sample $\boldsymbol{p}$ for every episode during training. Techniques from curriculum reinforcement learning Narvekar et al. (2020); Portelas et al. (2020) can also potentially improve the training of PCRL by using a progressing $\boldsymbol{p}$ preference distribution instead of uniform $\boldsymbol{p} \in \mathcal{P}$.

# B    IMPLEMENTATION OF BASELINE METHODS

We modify existing MOO algorithms EPO (Mahapatra and Rajan, 2020), CAGrad Liu et al. (2021), and SDMGrad (Xiao et al., 2023) for our proposed PCRL scheme and use them as baselines for our proposed PreCo algorithm.

## B.1    LINEAR SCALARIZATION PREFERENCE CONTROL

**Linear Scalarization (LS)**    For a preference $p$ conditioned policy $\pi_p$, it is updated by $d = \nabla \hat{v}^{\pi_p}$. Equivalently, it can be $d = p^T \nabla \hat{v}^{\pi_p}$, which means to linearly combine the objective gradients with a coefficient equal to the preference $p$.

**SDMGrad**    Similar to PreCo that needs to solve the min-norm problem (6) for update direction, Our implementation of SDMGrad solves:

$$\min_{w} \|\nabla_{\pi_p}^T \hat{v}^{\pi_p} w + \lambda p^T \nabla_{\pi_p} \hat{v}^{\pi_p}\| \tag{14}$$

$$d = \nabla_{\pi_p}^T \hat{v}^{\pi_p} w^* + \lambda \nabla_{\pi_p} \hat{v}^{\pi_p}, \tag{15}$$

where $w^*$ is the solution for problem (14). The update direction is for $\pi_p$ is $d$. We can see that this SDMGrad implementation and our proposed PreCo differ only in the term multiplied by $\lambda$. SDMGrad uses LS gradient for preference alignment while PreCo uses gradient of our proposed similarity function $\Psi$. This is why SDMGrad can be used as a case for ablation study of our method.

## B.2    SIMILARITY PREFERENCE CONTROL

**Exact Pareto Optimal (EPO)**    MOO methods such as PMTL Lin et al. (2019) and EPO Mahapatra and Rajan (2020) apply similarity constraints and have two modes for situations of low and high similarity. Based on this idea, we implement the EPO baseline as: When similarity is low, only similarity gradients $\nabla_{\pi_p} \Psi(w, \hat{v}^{\pi_p})$ will be used for update. When similarity is high enough, a common ascent direction calculated by MGDA (Désidéri, 2009) is used for update.

$$d = \nabla_{\pi_p} \Psi(w, \hat{v}^{\pi_p}), \qquad\qquad \text{if } \Psi'(w, \hat{v}^{\pi_p}) > \epsilon, \tag{16}$$

$$d = \nabla_{\pi_p}^T \hat{v}^{\pi_p} \arg\min_{w} \|\nabla_{\pi_p}^T \hat{v}^{\pi_p} w\|, \qquad\qquad \text{if } \Psi'(w, \hat{v}^{\pi_p}) \leq \epsilon, \tag{17}$$

where $\epsilon$ is a threshold of similarity and $\Psi'$ can be cosine similarity or our proposed $\Psi$. Equation (17) is the min-norm update from Désidéri (2009), which is equivalent to finding a common ascent direction that maximizes the least improvement among the objectives:

$$d = \arg\max_{d} \min_{i} \nabla_{\pi_p}^T \hat{v}_i^{\pi_p} d \tag{18}$$

Because most of the time during training, similarity is not high enough and only similarity gradients are applied in updates, this implementation of EPO can also be seen as an implementation of a relaxed Tchebycheff Scalarization, which avoids gradient oscillation as claimed in Mahapatra and Rajan (2020).

**Conflict-Averse Gradient (CAGrad)**    CAGrad tries to find a common ascent direction that is not too far from the average gradient. In our setting, we modify it to be a common ascent direction not is not too far from the similarity gradient.

$$d = \arg\max_{d} \min_{i} \nabla_{\pi_p}^T \hat{v}_i^{\pi_p} d \qquad \text{s.t. } \|d - \nabla_{\pi_p} \Psi(w, \hat{v}^{\pi_p})\| \leq c \nabla_{\pi_p} \Psi(w, \hat{v}^{\pi_p}), \tag{19}$$

where $c \in \{r \in \mathcal{R} \mid 0 < r < 1\}$ is a constraint constant to keep $d$ close to the similarity gradient $\nabla_{\pi_p} \Psi(w, \hat{v}^{\pi_p})$. This implementation might not apply to the convergence analysis in Liu et al. (2021). However, as shown by the empirical results, it works in practice for our PCRL scheme.

## C    MORE EXPERIMENTAL DETAILS

The test preferences are $p \in \mathcal{P}$ with a resolution of 0.1 for each dimension.

For instance, in 3-D cases, these preferences include

$$[0, 0, 1], [0, 0.1, 0.9], \dots, [0, 1, 0], [0.1, 0, 0.9], \dots, [0.9, 0.1, 0], [1, 0, 0],$$

with a quantity of 66. There are 286 test preferences for 4-D, 1001 for 5-D, and 3003 for 6-D.

During training, the preferences were sampled uniformly from the convex coefficient set $\mathcal{P}$, making the probability of sampling an exact test preference nearly zero. Therefore, high CS metric in test time means the ability to generalize to unseen preferences.

We run 5 seeds for each environment setting, and for each run, we select the best-performing agent as a candidate for testing. The results are presented as the mean and standard deviation of the 5 candidates.

### C.1    MO-ANT

The exact data for bar charts in Fig. 5 is shown in Table 2.

| | LS | SDMgrad | EPO | CAGrad | PreCO |
|---|---|---|---|---|---|
| HV($*1e^6$) | $6.81 \pm 0.24$ | $3.67 \pm 1.76$ | $6.75 \pm 0.08$ | $6.79 \pm 0.20$ | $\mathbf{6.85 \pm 0.21}$ |
| CS | $0.988 \pm 0.012$ | $0.937 \pm 0.014$ | $0.989 \pm 0.004$ | $0.988 \pm 0.0004$ | $\mathbf{0.990 \pm 0.004}$ |

Table 2: HV and CS performance in MO-Ant environment, the HV value has a unit of $1e^6$.

For MO-Ant, both SDMgrad and PreCo have $\lambda$ that increase linearly with each update from 1 to 5. EPO has a constraint threshold of $\epsilon = 3e - 4$ for cosine similarity $\Psi'$, which is very small, making it comparable to Tchebycheff Scalarization and also similar to PreCo with a large constant $\lambda_t = \lambda >> 1$. CAgrad has constraint $c = 0.2$. The definitions of $c$ and $\epsilon$ can be found in Appendix B.

Table 3: Hyper-parameters settings MO-Ant.

| Hyper-parameter | Value |
|---|---|
| Discount ($\gamma$) | 0.99 |
| Optimizer | Adam (Kingma and Ba, 2015) |
| Learning rate for networks | $3 \times 10^{-4}$ |
| Number of hidden layers for all networks | 3 |
| Number of hidden units per layer | 256 |
| Activation function | ReLU |
| Batch size | 256 |
| Buffer Size | $1 \times 10^6$ |
| Starting timesteps | $2.5 \times 10^3$ |
| Gradient clipping | False |
| Exploration method | Noise |
| Noise distribution | $\mathcal{N}(0, 0.1^2)$ |
| Noise clipping limit | 0.5 |
| Policy frequency (delay) | 2 |
| Target network update rate ($\tau$) | $5 \times 10^{-3}$ |
| Maximum episode timesteps | 500 |
| Peference sampling | every new episode untill max total steps is reached |
| Evaluation episodes for each test preference | 10 |

### C.2    MO-HOPPER

The exact data for bar charts in Fig. 6 is shown in Table 4.

|  | LS | SDMgrad | EPO | CAGrad | PreCO |
|---|---|---|---|---|---|
| HV($*1e^6$) | $2.63 \pm 0.12$ | $2.24 \pm 0.13$ | $2.53 \pm 0.28$ | $0.94$ | $\mathbf{2.67 \pm 0.26}$ |
| CS | $0.933 \pm 0.007$ | $0.922 \pm 0.018$ | $\mathbf{0.963 \pm 0.023}$ | $0.932 \pm 0.024$ | $0.957 \pm 0.011$ |

Table 4: HV and CS performance in MO-hopper environment, the HV value has a unit of $1e^6$.

For MO-Hopper, both SDMgrad and PreCo have $\lambda$ increasing linearly with every update from 3 to 11. EPO has a constraint threshold of $\epsilon = 3e - 4$ for cosine similarity $\Psi'$. CAgrad has constraint $c = 0.1$.

Table 5: Hyper-parameters settings MO-Hopper.

| Hyper-parameter | Value |
|---|---|
| Discount ($\gamma$) | 0.99 |
| Optimizer | Adam (Kingma and Ba, 2015) |
| Learning rate for networks | $3 \times 10^{-4}$ |
| Number of hidden layers for all networks | 3 |
| Number of hidden units per layer | 256 |
| Activation function | ReLU |
| Batch size | 256 |
| Buffer Size | $1 \times 10^6$ |
| Starting timesteps | $2.5 \times 10^3$ |
| Gradient clipping | False |
| Exploration method | Noise |
| Noise distribution | $\mathcal{N}(0, 0.1^2)$ |
| Noise clipping limit | 0.5 |
| Policy frequency (delay) | 2 |
| Target network update rate ($\tau$) | $5 \times 10^{-3}$ |
| Maximum episode timesteps | 500 |
| Peference sampling | every new episode untill max total steps is reached |
| Evaluation episodes for each test preference | 10 |

## C.3 MO-REACHER

The exact data for Fig. 7 is shown in Table 6.

|  | random | LS | SDMgrad | EPO | CAGrad | PreCO |
|---|---|---|---|---|---|---|
| HV($*1e^8$) | $13.47$ | $15.66 \pm 8.03$ | $20.37 \pm 0.37$ | $9.87 \pm 3.11$ | $13.46 \pm 9.71$ | $33.11 \pm 6.29$ |
| CS | $0.758$ | $0.652 \pm 0.030$ | $0.761 \pm 0.001$ | $0.845 \pm 0.078$ | $0.760 \pm 0.002$ | $\mathbf{0.906 \pm 0.002}$ |

Table 6: HV and CS performance in MO-reacher environment, the HV value has a unit of $1e^6$.

For MO-Reacher, both SDMgrad and PreCo have $\lambda_t$ increasing linearly with every update from 10 to 20. EPO has a constraint threshold of $\epsilon = 3e - 4$ for cosine similarity $\Psi'$, which is very small, making it comparable to Tchebycheff Scalarization and also similar to PreCo with a large constant $\lambda_t = \lambda >> 1$. CAgrad employs a constraint constant of $c = 0.1$.

## C.4 FRUIT TREE

For 3-D reward, most runs of LS only learn a very limited number of values like $[4.71, 5.39, 5.40]$ and the values SDMGrad for most test preferences lie at $[4.01, 7.17, 1.47]$. They, as the LS approach, discover much less Prareto optimal policies than methods of the similarity approach, which have one value for each test preference. This shows the limitation of linear scalarization methods. In theory, LS methods have the potential to discover all Pareto optimal policies for MORL (Lu et al., 2023).

Table 7: Hyper-parameters settings MO-Reacher.

| Hyper-parameter | Value |
|---|---|
| Discount ($\gamma$) | 0.99 |
| Optimizer | Adam (Kingma and Ba, 2015) |
| Learning rate for networks | $3 \times 10^{-4}$ |
| Number of hidden layers for all networks | 3 |
| Number of hidden units per layer | 256 |
| Activation function | ReLU |
| Batch size | 250 |
| Gradient clipping | False |
| Exploration method | Policy Entropy |
| Entropy Coefficient | 0.001 |
| epsilon-clip for PPO | 0.001 |
| Epochs per PPO update | 3 |
| Timesteps every update | 100 |
| Maximum episode timesteps | 250 |
| Number of episodes per preference sample | 40 |
| Number of preference samples (for 4D reward) | 600 |
| Evaluation episode for each test preference | 10 |

However, in practice, this is often not the case. Possible reasons could be the numerical instability inherent in deep RL, limitations of model capacity, and the fact that the value space is usually **not strictly convex**.

Table 8: Hyper-parameters settings Fruit-tree.

| Hyper-parameter | Value |
|---|---|
| Discount ($\gamma$) | 0.99 |
| Optimizer | Adam (Kingma and Ba, 2015) |
| Learning rate for networks | $3 \times 10^{-4}$ |
| Number of hidden layers for all networks | 3 |
| Number of hidden units per layer | 256 |
| Activation function | ReLU |
| Batch size | 100 |
| Gradient clipping | False |
| Exploration method | Policy Entropy |
| Entropy Coefficient | 0.001 |
| epsilon-clip for PPO | 0.001 |
| Epochs per PPO update | 3 |
| Timesteps every update | 100 |
| Maximum episode timesteps | 100 |
| Number of episodes per preference sample | 20 |
| Number of preference samples (for 4D reward) | 3000 |
| Evaluation episode for each test preference | 10 |

# D PRACTICAL IMPLEMENTATION OF THE UPDATE PROCEDURE

## D.1 ALGORITHM FRAMEWORK

Our goal is to train a single agent that can be conditioned on different performance preferences and 0-shot adapt to user preference at test time. During training, we need to uniformly sample preferences from $\mathcal{P}$ and let the agent learn to find Pareto optimal policies with values aligned to $p$. The procedure is shown in Algorithm 2.

---

**Algorithm 2** PCRL with PreCo update

---

**Initialize:**
    $\mathcal{B}$: Buffer.
    $N$: Number of training samples for $\boldsymbol{p}$,
    $E$: Number of training episodes for every $\boldsymbol{p}$ sample,
    $\pi_\theta$: Preference-conditioned actor model,
    $\mathbf{Q}_\phi$ for DDPG/TD3 or $\mathbf{V}_\phi$ for A3C/PPO: Preference-conditioned critic model with $m$-dimensional output, where $m$ is the objective number.
**for** $n = 0, 1, ..., N-1$ **do**
    Sample preference $\boldsymbol{p} \in \mathcal{P}$
    **for** $e = 0, 1, ..., E-1$ **do**
        Rollout with policy $\pi_\theta(\cdot|\cdot, \boldsymbol{p})$
        Store transitions $(s, a, \boldsymbol{r}, \boldsymbol{p})$ in $\mathcal{B}$
    **end for**
    Update $\mathbf{Q}_\phi$ or $\mathbf{V}_\phi$ by minimizing TD error for every objective.
    Estimate policy-level gradient $\nabla_{\pi_{\boldsymbol{p}}}^T \hat{\boldsymbol{v}}^{\pi_{\boldsymbol{p}}}$ by Eq.21/23 for TD3 or Eq.27/34 for PPO.
    Estimate similarity gradient $\nabla_{\pi_{\boldsymbol{p}}} \Psi(\boldsymbol{p}, \hat{\boldsymbol{v}}^{\pi_{\boldsymbol{p}}}) = \nabla_{\pi_{\boldsymbol{p}}}^T \hat{\boldsymbol{v}}^{\pi_{\boldsymbol{p}}} \nabla_{\boldsymbol{v}} \Psi(\boldsymbol{p}, \hat{\boldsymbol{v}}^{\pi_{\boldsymbol{p}}})$
    Get policy-level update direction $\boldsymbol{d}^*$ by solving Eq.6 with $\nabla_{\pi_{\boldsymbol{p}}}^T \hat{\boldsymbol{v}}^{\pi_{\boldsymbol{p}}}$ and $\nabla_{\pi_{\boldsymbol{p}}} \Psi(\boldsymbol{p}, \hat{\boldsymbol{v}}^{\pi_{\boldsymbol{p}}})$
    Update $\theta$ by solving Eq.22 for TD3 or Eq.29/31 for PPO with $\boldsymbol{d}^*$
**end for**

---

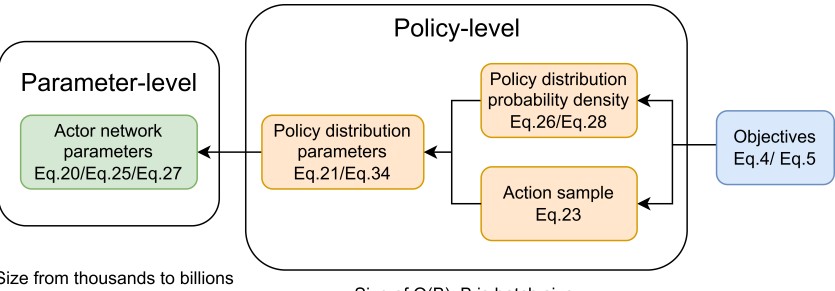

Figure 9: Backward path of policy update. we can see that the gradient from objectives to the parameters of the actor network first backpropagate through the probability density of action distribution (for policy-based methods such as A3C/PPO) or action sample (for value-based methods such as DDPG/TD3), then propagate through the distribution parameters of policies such as the logits for categorical distribution or $\mu, \Sigma$ for Gaussian distributions. We consider these the policy-level gradients. They often have a size of $\mathcal{O}(B)$, where $B$ is the batch size. Since $B$ is often limited to a few hundreds, the size of a policy-level gradient would be much smaller than the size of the neural network parameter.

## D.2 POLICY-LEVEL GRADIENT

Solving the min-norm problem (6) with parameter-level gradients $\nabla_\theta \hat{\boldsymbol{v}}^{\pi_{\boldsymbol{p}}}$ at every gradient update can be memory and computationally expensive when $|\theta|$ is large. Video game playing agents like AlphaZero Silver et al. (2018) and AlphaStar Vinyals et al. (2019) can have millions of model parameters. Besides, Large models with billions of model parameters have become very common with recent developments in Large Language Models (LLMs) Zhao et al. (2023); Minaee et al. (2024). To circumvent this issue, we suggest solving the min-norm problem (6) before the gradient propagates to the model parameter $\theta$. Therefore, ideally, we want to solve the min-norm problem with gradients at the policy-level $\nabla_{\pi_{\boldsymbol{p}}} \hat{\boldsymbol{v}}^{\pi_{\boldsymbol{p}}}$, which has only a size of batch size $B$ of hundreds at each update; In practice for deep RL implementations, as shown by Fig. 9, $\nabla_{\pi_{\boldsymbol{p}}} \hat{\boldsymbol{v}}^{\pi_{\boldsymbol{p}}}$ can also be replaced by the gradients of the value $\hat{\boldsymbol{v}}^{\pi_{\boldsymbol{p}}}$ with respect to the policy model outputs, such as $\nabla_{l_{\boldsymbol{p}}} \hat{\boldsymbol{v}}^{\pi_{\boldsymbol{p}}}$ for logits of categorical distribution policies and $\nabla_{\mu_{\boldsymbol{p}}, \sigma_{\boldsymbol{p}}} \hat{\boldsymbol{v}}^{\pi_{\boldsymbol{p}}}$ for means and standard deviations of diagonal Gaussian distribution policies.

### D.2.1 Continuous Action space

Value-based methods like TD3 Fujimoto et al. (2018) and SAC Haarnoja et al. (2018) are often used for continuous action spaces. To avoid computing min-norm with parameter gradient

$$\nabla_\theta \hat{\boldsymbol{v}}^{\pi_{\boldsymbol{p}}} = \mathbb{E}[\nabla_\theta \hat{\mathbf{Q}}(s, a, \boldsymbol{p})|_{a \sim \pi_\theta(s, \boldsymbol{p})}] \tag{20}$$

We look at their policy formulations. Their policy $\pi(a|s, \boldsymbol{p})$ is often a Gaussian or squashed Gaussian distribution with parameters mean $\mu_{\boldsymbol{p}}(s)$ and log standard deviation $\log \sigma_{\boldsymbol{p}}(s)$. We denote a distribution parameter vector $\rho_{\boldsymbol{p}}$ with $\rho_{\boldsymbol{p}}(s) = [\mu_{\boldsymbol{p}}(s), \log \sigma_{\boldsymbol{p}}(s)]^T$ and we can get

$$\nabla_{\rho_{\boldsymbol{p}}} \hat{\boldsymbol{v}}^{\pi_{\boldsymbol{p}}} = \mathbb{E}[\nabla_{\rho_{\boldsymbol{p}}} \hat{\mathbf{Q}}(s, a, \boldsymbol{p})] \tag{21}$$

For each update, the size of each objective gradient $\nabla_{\rho_{\boldsymbol{p}}} \hat{\boldsymbol{v}}_i^{\pi_{\boldsymbol{p}}}$ is $2|\mathcal{A}| \times B$, where $B$ is the batch size and $\rho$ has a size of $2|\mathcal{A}|$. This means that $\nabla_{\rho_{\boldsymbol{p}}} \hat{\boldsymbol{v}}_i(s, \boldsymbol{p})$ could have a much lower dimension than $\nabla_\theta \hat{\boldsymbol{v}}_i(s, \boldsymbol{p})$, thus increasing the memory and computational efficiency.

After getting the update direction $\boldsymbol{d}$ for $\rho_{\boldsymbol{p}}$ by solving the min norm problem (6) with $\nabla_{\rho_{\boldsymbol{p}}} \hat{\boldsymbol{v}}^{\pi_{\boldsymbol{p}}}$, we update model parameter $\theta$ by solving

$$\max_\theta \left\{ \boldsymbol{d}^\top \nabla_\theta \rho_{\boldsymbol{p}} \ \ s.t. \ \|\rho - \rho_{\text{old}}\|_2 < \delta \right\},$$

which is a trust region formulation that updates $\rho$ in the direction of $\boldsymbol{d}$ while keeping in a local region where $\boldsymbol{d}$ is valid. A simple and practical implementation for parameter update can be as follows:

$$\max_\theta \mathcal{J}(\theta) = \mathbb{E}_{s,a} \left[ \text{clip}(\rho_{\boldsymbol{p}}(s, a), \rho_{\boldsymbol{p}}(s, a) - \epsilon, \rho_{\boldsymbol{p}}(s, a) + \epsilon) \boldsymbol{d}(s, a) \right]. \tag{22}$$

The update of every entry of $\pi$ is clipped to $\epsilon$, so $\|\pi_\theta - \pi_\theta\|_2 \leq \sqrt{B * \epsilon^2} = \delta$, where $B$ is the batch size.

For more expressive models such as diffusion models Wang et al. (2023) or normalizing flows Brahmanage et al. (2023), the mean and covariance gradients would not be adequate. We can instead use the gradient of action samples as policy-level gradients and we get an update direction

$$\boldsymbol{d}(s, a) = \nabla_a \hat{\mathbf{Q}}(s, a, \boldsymbol{p}) \tag{23}$$

for every $(s, a)$ sample in the batch. Then we can perform min-norm with $\boldsymbol{d}$ and update the more expressive policy networks by reparameterization techniques.

### D.2.2 Discrete action space

Policy-based methods like A3C Mnih et al. (2016) and PPO Schulman et al. (2017) are often used for discrete action spaces. We can approximate the multi-objective value function $\hat{\boldsymbol{v}}^{\pi_{\boldsymbol{p}}}(s)$ by a function $\hat{\boldsymbol{v}}(s, \boldsymbol{p})$ that takes $s$ and $\boldsymbol{p}$ as inputs, sample the episodic returns as vector $\mathbf{R}$, and calculate the multi-objective advantage function as

$$\hat{\mathbf{A}}(s, a) = \mathbf{R} - \hat{\boldsymbol{v}}(s, \boldsymbol{p}) \tag{24}$$

Then, the policy gradient in the model parameter space is

$$\nabla_\theta \hat{\boldsymbol{v}}^{\pi_{\boldsymbol{p}}} = \mathbb{E} \left[ \frac{\nabla_\theta \pi(a|s, \boldsymbol{p})}{\pi(a|s, \boldsymbol{p})} \hat{\mathbf{A}}(s, a, \boldsymbol{p}) \right], \tag{25}$$

And for gradient at the policy space $\boldsymbol{d} = \nabla_{\pi_{\boldsymbol{p}}} \hat{\boldsymbol{v}}(s, \boldsymbol{p})$, we have

$$\boldsymbol{d}(s, a) = \frac{1}{\pi(a|s, \boldsymbol{p})} \hat{\mathbf{A}}(s, a, \boldsymbol{p}) \tag{26}$$

for very $(s, a)$ sample. When using policy optimization methods like PPO/TRPO they are

$$\nabla_\theta \hat{\boldsymbol{v}}^{\pi_{\boldsymbol{p}}} = \mathbb{E} \left[ \frac{\nabla_\theta \pi(a|s, \boldsymbol{p})}{\pi_{\text{old}}(a|s, \boldsymbol{p})} \hat{\mathbf{A}}(s, a, \boldsymbol{p}) \right], \tag{27}$$

$$\boldsymbol{d}(s, a) = \frac{1}{\pi_{\text{old}}(a|s, \boldsymbol{p})} \hat{\mathbf{A}}(s, a, \boldsymbol{p}) \tag{28}$$

In practical situations, at every update, the size of each objective gradient $\nabla_{\pi_{\boldsymbol{p}}} \hat{\boldsymbol{v}}_i(s, \boldsymbol{p})$ is the batch size $B$, and the min norm problem (6) can be performed with gradients of batch size $B$, which could be much smaller than the parameter size of deep neural networks, especially when implemented for large language models.

After getting the update direction $d = \nabla_{\pi_{\boldsymbol{p}}} \hat{\boldsymbol{v}}(s, \boldsymbol{p})$ for $\pi_{\boldsymbol{p}}$, we optimize model parameters by

$$\max_\theta \left\{ d^\top \nabla_\theta \pi_{\boldsymbol{p}} \ \ s.t. \ \|\pi_{\boldsymbol{p}} - \pi_{\boldsymbol{p},\text{old}}\|_2 < \delta \right\}, \tag{29}$$

which is a trust region formulation that updates $\pi_{\boldsymbol{p}}$ in the direction of $d$ while keeping in a local region where $\boldsymbol{d}$ is valid. This can be practically implemented by an objective as follows:

$$\mathcal{J}(\theta) = \mathbb{E}_{s,a} \left[ \text{clip}(\pi_\theta(a|s, \boldsymbol{p}), \pi_{\theta_\text{old}}(a|s, \boldsymbol{p}) - \epsilon, \pi_{\theta_\text{old}}(a|s, \boldsymbol{p}) + \epsilon) \boldsymbol{d}(s, a) \right]. \tag{30}$$

The update of every entry of $\pi$ is clipped to $\epsilon$, so $\|\pi_\theta - \pi_\theta\|_2 \leq \sqrt{B * \epsilon^2} = \delta$, where $B$ is the batch size.

Trust region formulation with KL-divergence could be more suitable for categorical distribution $\pi_{\boldsymbol{p}}$, so another formulation of parameter update could be

$$\max_\theta \left\{ d^\top \nabla_\theta \pi_{\boldsymbol{p}} \ \ s.t. \ \ D_\text{KL}(\pi_{\boldsymbol{p}} \parallel \pi_{\boldsymbol{p},\text{old}}) < \delta \right\}. \tag{31}$$

Whether a KL divergence trust region is theoretically compatible with the solution $\boldsymbol{d}$ of the min norm problem (6) will be further researched in our future work.

One potential issue of $\nabla_{\pi_{\boldsymbol{p}}} \hat{\boldsymbol{v}}(s, \boldsymbol{p}))$ is that $(\pi_{\boldsymbol{p}} + \alpha \nabla_{\pi_{\boldsymbol{p}}} \hat{\boldsymbol{v}}(s, \boldsymbol{p}))$ may not be in the probability simplex. As a result, projecting it back onto the probability simplex could cause it to deviate from the intended update direction. Since policies for discrete action spaces are often categorical distributions, one way to avoid this issue is to consider the gradient of $l_{\boldsymbol{p}}$, which denotes the logits for policy $\pi(\cdot|\cdot, \boldsymbol{p})$ conditioned on preference $\boldsymbol{p}$, and $l_{\boldsymbol{p}}(s)$ are the logits for $\pi(\cdot|s, \boldsymbol{p})$. The logits do not have the constraint to be in the probability simplex.

$$\begin{aligned}
&\nabla_{l_{\boldsymbol{p}}} \pi_{\boldsymbol{p}}(s) \\
&= -\pi(a|s, \boldsymbol{p}) \left[ \pi(a_1|s, \boldsymbol{p}), \pi(a_2|s, \boldsymbol{p}), ..., \pi(a|s, \boldsymbol{p}) - 1, ..., \pi(a_{|\mathcal{A}|}|s, \boldsymbol{p}) \right]^T,
\end{aligned} \tag{32}$$

where $\nabla_{l_{\boldsymbol{p}}} \pi_{\boldsymbol{p}}(s)$ is the $s$-th entry of the jacobian $\nabla_{l_{\boldsymbol{p}}} \pi_{\boldsymbol{p}}$, and

$$\left[ \pi(a_1|s, \boldsymbol{p}), \pi(a_2|s, \boldsymbol{p}), ..., \pi(a|s, \boldsymbol{p}) - 1, ... \right]^T$$

is a vector of action space size $|\mathcal{A}|$.

Then, we can get

$$\begin{aligned}
&\nabla_{l_{\boldsymbol{p}}} \hat{\boldsymbol{v}}^{\pi_{\boldsymbol{p}}}(s) \\
&= \mathbb{E} \left[ -\frac{\pi(a|s, \boldsymbol{p})}{\pi_\text{old}(a|s, \boldsymbol{p})} \hat{\mathbf{A}}(s, a, \boldsymbol{p}) [\pi(a_1|s, \boldsymbol{p}), \pi(a_2|s, \boldsymbol{p}), ..., \pi(a|s, \boldsymbol{p}) - 1, ..., \pi(a_{|\mathcal{A}|}|s, \boldsymbol{p})] \right],
\end{aligned} \tag{33}$$

where $\nabla_{l_{\boldsymbol{p}}} \hat{\boldsymbol{v}}^{\pi_{\boldsymbol{p}}}(s)$, of size $m \times \mathcal{A}$, is the index $[:, s, :]$ for $\nabla_{l_{\boldsymbol{p}}} \hat{\boldsymbol{v}}^{\pi_{\boldsymbol{p}}}$ tensor, of size $m \times |\mathcal{S}| \times |\mathcal{A}|$.

In every update, the size of the objective gradient $\nabla_{l_{\boldsymbol{p}}} \hat{\boldsymbol{v}}^{\pi_{\boldsymbol{p}}}(i, s)$ for the objective $i$ has a size of $B \times |\mathcal{A}|$. For large language models, the action space could be the vocabulary size of tens of thousands, so $B \times |\mathcal{A}|$ could be in millions, but is still much smaller than the parameter size that is often in billions. Moreover, for large action spaces, $\pi(a|s, \boldsymbol{p}) \cdot \pi(a'|s, \boldsymbol{p})$ could be much smaller than $\pi(a|s, \boldsymbol{p})$, so $\nabla_{l_{\boldsymbol{p}}} \hat{\boldsymbol{v}}^{\pi_{\boldsymbol{p}}}(s, a)$ can be approximated by

$$\nabla_{l_{\boldsymbol{p}}} \hat{\boldsymbol{v}}^{\pi_{\boldsymbol{p}}}(s, a) \approx \frac{\pi(a|s, \boldsymbol{p})}{\pi(a|s, \boldsymbol{p})} \hat{\mathbf{A}}(s, a, \boldsymbol{p}) = \hat{\mathbf{A}}(s, a, \boldsymbol{p}). \tag{34}$$

This approximation of $\nabla_{l_{\boldsymbol{p}}} \hat{\boldsymbol{v}}^{\pi_{\boldsymbol{p}}}$ is what we implemented to replace $\nabla_{\pi_{\boldsymbol{p}}} \hat{\boldsymbol{v}}^{\pi_{\boldsymbol{p}}}$ in the min norm problem (6) to avoid solving min norm with large parameter gradient $\nabla_\theta \hat{\boldsymbol{v}}^{\pi_{\boldsymbol{p}}}$. The model parameters are updated by solving (29) using $\nabla_{l_{\boldsymbol{p}}} \hat{\boldsymbol{v}}^{\pi_{\boldsymbol{p}}}$ as $\boldsymbol{d}$.

# E MORE DETAILS ABOUT THE SIMILARITY OBJECTIVE

## E.1 ABOUT PROPOSED SIMILARITY FUNCTION $\Psi$

For two objective cases, when $\boldsymbol{p} = [0.5, 0.5]$, $\Psi(\boldsymbol{p}, \boldsymbol{v})$ is shown in Fig. 10. The x-axis is the first element of $\boldsymbol{v}$ and y-axis is the second element of $\boldsymbol{v}$. The z-axis is the value of $\Psi(\boldsymbol{p}, \boldsymbol{v})$.

We can see that the similarity is maximized to 0 only when $\frac{\boldsymbol{v}_0}{\boldsymbol{p}_0} = \frac{\boldsymbol{v}_1}{\boldsymbol{p}_1}$. It is also smooth as proved by Lemma 4.1.

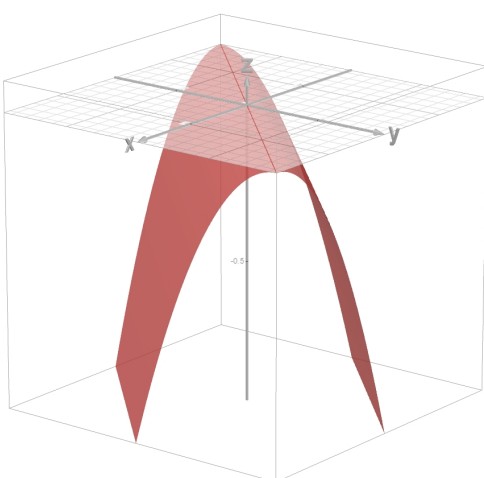

Figure 10: $\Psi(\boldsymbol{p}, \cdot)$ when $\boldsymbol{p} = [0.5, 0.5]^T$

## E.2 SIMILARITY OBJECTIVE DESIGN FOR BETTER THEORETICAL PROPERTIES

Theorem 4.2 requires the similarity gradient to be both Lipschitz continuous and convex combinations of the objective gradients. Which formally means that for an similarity objective $\Psi'(\boldsymbol{p}, \cdot)$,

$$\nabla_{\boldsymbol{v}} \Psi'(\boldsymbol{p}, \boldsymbol{v}) \in \mathcal{W},$$

which can not satisfied by $\nabla_{\boldsymbol{v}} \Psi(\boldsymbol{p}, \boldsymbol{v}) = \max_i \frac{\boldsymbol{v}_i}{\boldsymbol{p}_i} \boldsymbol{p} - \boldsymbol{v}$, because $\nabla_{\boldsymbol{v}} \Psi(\boldsymbol{p}, \boldsymbol{v})$ will be $\boldsymbol{0}$ when $\boldsymbol{p}$ and $\boldsymbol{v}$ are perfectly aligned. Moreover, we can not directly normalize $\nabla_{\boldsymbol{v}} \Psi(\boldsymbol{p}, \boldsymbol{v})$ by dividing $\|\nabla_{\boldsymbol{v}} \Psi(\boldsymbol{p}, \boldsymbol{v})\|_1$, because this will make the normalized gradient not Lipschitz continuous (the gradient changes drastically when $\boldsymbol{v}$ passes the direction of $\boldsymbol{p}$).

Our hints to design such a similarity function $\Psi'(\boldsymbol{p}, \cdot)$ are as follows: Its similarity gradient $\nabla_{\boldsymbol{v}} \Psi'(\boldsymbol{p}, \boldsymbol{v})$ could get close to $\boldsymbol{p}$, when $\boldsymbol{v}$ has a high similarity to the preference $\boldsymbol{p}$; And $\nabla_{\boldsymbol{v}} \Psi'(\boldsymbol{p}, \boldsymbol{v})$ should be close to the normalized gradient $\nabla_{\boldsymbol{v}} \Psi(\boldsymbol{p}, \boldsymbol{v}) / \|\nabla_{\boldsymbol{v}} \Psi(\boldsymbol{p}, \boldsymbol{v})\|_1$ when the similarity is low.

# F MOO TOY EXAMPLE

This is an toy example used in SDMGrad (Xiao et al., 2023) to show that in MOO, our proposed PreCo can achieve better or comparable performance under stochastic settings. Besides, PreCo can find the Pareto optimal point optimizing the similarity function $\Psi(\boldsymbol{p}, \cdot)$.

The two objectives $L_1(x)$ and $L_2(x)$ shown in Fig. 11 are defined on $x = (x_1, x_2)^\top \in \mathbb{R}^2$,

$$L_1(x) = f_1(x)g_1(x) + f_2(x)h_1(x) \text{ and } L_2(x) = f_1(x)g_2(x) + f_2(x)h_2(x),$$

where the functions are given by

$$f_1(x) = \max\big(\tanh(0.5x_2), 0\big)$$

$$f_2(x) = \max\big(\tanh(-0.5x_2), 0\big)$$

$$g_1(x) = \log\Big(\max\big(|0.5(-x_1 - 7) - \tanh(-x_2)|, 0.000005\big)\Big) + 6$$

$$g_2(x) = \log\Big(\max\big(|0.5(-x_1 + 3) - \tanh(-x_2) + 2|, 0.000005\big)\Big) + 6$$

$$h_1(x) = \big((-x_1 + 7)^2 + 0.1(-x_1 - 8)^2\big)/10 - 20$$

$$h_2(x) = \big((-x_1 - 7)^2 + 0.1(-x_1 - 8)^2\big)/10 - 20.$$

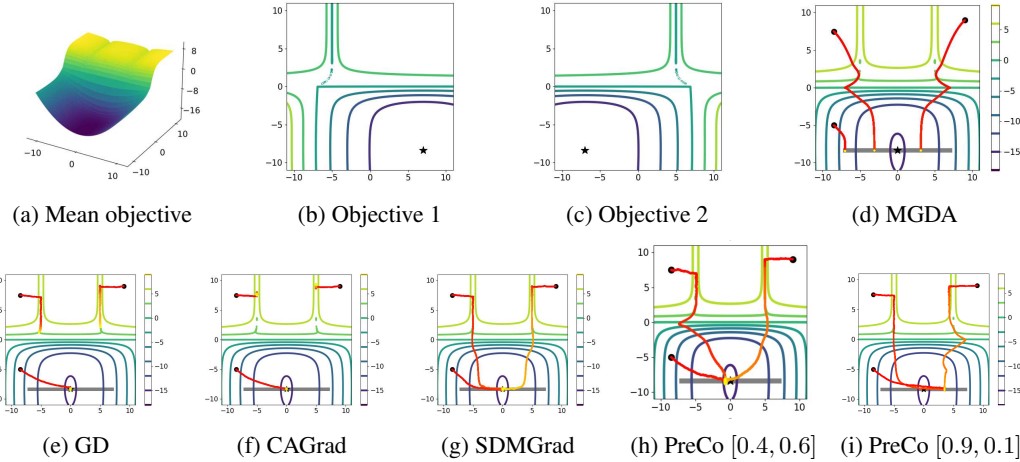

(a) Mean objective    (b) Objective 1    (c) Objective 2    (d) MGDA

(e) GD    (f) CAGrad    (g) SDMGrad    (h) PreCo $[0.4, 0.6]$    (i) PreCo $[0.9, 0.1]$

Figure 11: A two-objective toy example.

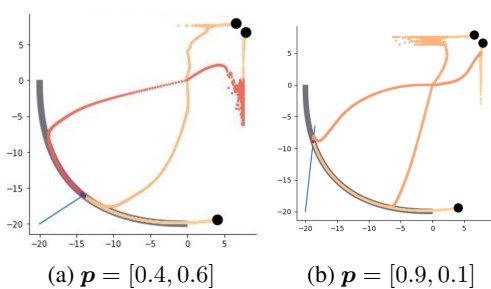

(a) $\boldsymbol{p} = [0.4, 0.6]$      (b) $\boldsymbol{p} = [0.9, 0.1]$

Figure 12: Plots showing the Pareto front: The x-axis is $L_1$ and the y-axis is $L_2$. The blue line is the direction of the preference, for (a) it is $\boldsymbol{p} = [0.4, 0.6]$, and for (b) it is $\boldsymbol{p} = [0.9, 0.1]$. All three initial points converged to the Pareto optimal point that intersects with the line of $\boldsymbol{p}$ direction.

Initializations points are from $\{(-8.5, 7.5), (-8.5, 5), (9, 9)\}$. The optimization trajectories are visualized in Fig. 11. The starting point of every trajectory in Fig. 11d-Fig. 11g is given by the ● symbol, and the color of every trajectory changes gradually from red to yellow. The gray horizontal line illustrates the Pareto front, and the ⋆ symbol denotes the global optimum for the mean objective $L_0 = 0.5L_1 + 0.5L_2$. The setting is the same as in Xiao et al. (2023) and all other methods except PreCo optimize for $L_0$. Zero-mean Gaussian noise is added to the gradient of each objective for all the methods except MGDA. Adam optimizer is adopted with learning rate of 0.002 and 70000 iterations for each run. We can see that GD and CAGrad can fail to converge to the Pareto front in certain circumstances. Only SDMGrad and our proposed PCGrad converge to the Pareto front in all cases. Notice that, the preference PreCo in Fig. 11h is $\boldsymbol{p} = [0.4, 0.6]$, as shown in Fig. 12, we can see

that it converges to a point optimizing $\Psi(\boldsymbol{p}, [L_1, L_2]^T)$. In addition, Figs. 11i and 12b show for the case where $\boldsymbol{p} = [0.9, 0.1]$, PreCo also updates to the preference specific Pareto optimal point.

Below is a figure comparing PreCo with existing preference following MOO algorithms such as Tchebycheff scalarization (Lin et al., 2024)(TS) Ehrgott (2005) and smooth Tchebycheff scalarization

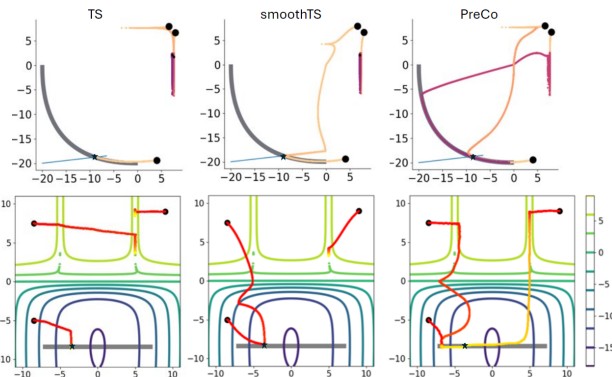

Figure 13: Plots showing the results of TS, SmoothTS and PreCo for preference $\boldsymbol{p} = [0.1, 0.9]$. Our proposed PreCo converges to the preference-aligned Pareto front for all initialization.

In Fig. 13, PreCo first converges to the Pareto front, then as $\lambda$ goes up, it converges to the preference desired solution.

## G   ADDITIONAL RESULTS COMPARING SIMILARITY-BASED METHODS AND CONCAVE AUGUMENTATION

Lu et al. (2023) has tried to address LS limitations by adding a concave augmentation term to the reward, transforming the original Pareto front into a strictly convex one. Then for this "more convex" new problem, LS can find more optimal solutions. Their implementation only included SAC Haarnoja et al. (2018), as the entropy maximization in SAC serves as the reward augmentation. To ensure a fair comparison independent of settings, code-level implementations, and algorithmic techniques (such as HER (Andrychowicz et al., 2017)), we modified our original LS with PPO into a "maximum entropy PPO". The modified multi-objective advantages are:

$$\hat{\mathbf{A}}(s, a) = \mathbf{R} + E - \hat{\boldsymbol{v}}(s, \boldsymbol{p}) \tag{35}$$

where $\mathbf{R}$ represents the vector of multi-objective episodic returns, $E$ denotes the sum of future policy entropies in the sampled episode, and $\hat{\boldsymbol{v}}(s, \boldsymbol{p})$ is the multi-objective vector value conditioned on preference $\boldsymbol{p}$, approximating both expected returns and entropies:

$$\min_{\hat{\boldsymbol{v}}} \mathbb{E}\left[||\mathbf{R} + E - \hat{\boldsymbol{v}}(s, \boldsymbol{p})||^2\right] \tag{36}$$

With these modifications, our modified "maximum entropy multi-objective PPO" with Linear Scalarization(LS) is optimizing the concave augmented objective in Eq.(10) from Lu et al. (2023).

To showcase the advantage of our PCRL (Ours) framework with similarity-based methods EPO and PreCo (ours), we test in the 'simple but hard' fruit-tree environment. It is simple for RL due to small discrete state and action spaces but challenging for MORL with 6 objectives and a non-strictly convex Pareto front. This comparison isolates MORL performance from lower-level RL factors, directly highlighting our method's strengths.

The results in Table 9 show when the augmentation strength $\alpha = 0.01$, the performance of CAPQL-modified PPO is marginally better than the original LS ($\alpha = 0$), but still significantly worse than similarity-based methods (EPO, PreCo(ours)). Larger $\alpha$ values lead to performance drops. This result aligns with Remark 5 and Figure 9 in Lu et al. (2023), which highlights that such augmentation can cause information loss in the original problem, and excessive augmentation results in performance degradation. In contrast, our method has the theoretical advantage of overcoming the LS limitation without any reward augmentation, thus avoiding information loss from the original problem.

| $\alpha$/Method | Hyper volume($1e3$) | Cosine Similarity |
|---|---|---|
| 0 (LS) | $5.74 \pm 0.88$ | $0.718 \pm 0.040$ |
| 0.01 | $5.95 \pm 1.12$ | $0.722 \pm 0.006$ |
| 0.05 | $5.18 \pm 0.36$ | $0.718 \pm 0.040$ |
| 0.10 | $1.75 \pm 1.35$ | $0.633 \pm 0.141$ |
| EPO | $14.97 \pm 2.29$ | $0.77 \pm 0.03$ |
| PreCo(ours) | $15.61 \pm 0.75$ | $0.78 \pm 0.03$ |

Table 9: HV and CS performance in 6D Fruit-tree environment, the HV value has a unit of $1e^3$. The comparison is between PCRL (Ours) framework with similarity-based methods such as EPO and PreCo (Ours) and LS with different strength of concave augmentation from Lu et al. (2023)

## H CALIBRATION

The reachable Pareto front for PCRL is often not the entire $\mathbb{R}^m$ value space, and there are often gaps between the desired preferences and the values reached. To calibrate the possible misalignment between the input preference and the reached value in a sample-efficient way, we employ a Gaussian process (GP) based method to model the relationship between the input of the desired preference and the actual values reached by the agent.

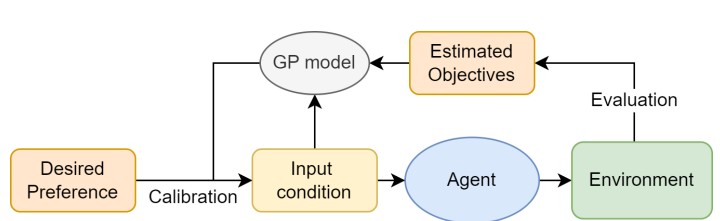
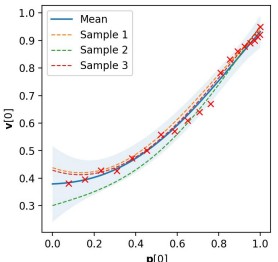

Figure 14: After training, due to general errors in deep learning or unreachable regions of the Pareto front, there could still be gaps between the actually reached objectives ratios and the desired preference. The calibration procedure obtains the reachable reachable Pareto front and modifies the desired preference into an input that results in performance more aligned with the desired preference.

Figure 15: Example of GP regression for the $(\boldsymbol{p}, \boldsymbol{v}^{\pi_p})$ samples, this case has two objectives and shows the relation between the first element of the value and the first element of the preference. Each red cross point in this plot is a $(\boldsymbol{p}_0, \boldsymbol{v}_0^{\pi_p})$, where $\boldsymbol{p}_0, \boldsymbol{v}_0^{\pi_p}$ are the first elements of $\boldsymbol{p}, \boldsymbol{v}^{\pi_p}$.

After training, there might still be an input $\boldsymbol{p}'$ with better $\Psi(\boldsymbol{p}, \boldsymbol{v}^{\pi_{p'}})$ than $\boldsymbol{v}^{\pi_p}$. We want to find $\boldsymbol{p}'$ that solves $\max_{\boldsymbol{p}'} E[\Psi(\boldsymbol{p}, \boldsymbol{v}^{\pi_{p'}})]$ for any $\boldsymbol{p}$. we first uniformly sample the values $\boldsymbol{v}^{\pi_p}$ reached by giving the agent preference input from $\{\boldsymbol{p} \in \mathbb{R}^m : \boldsymbol{p}^T \mathbf{1} = 1\}$, then perform a GP regression for the $(\boldsymbol{p}, \boldsymbol{v}^{\pi_p})$ samples. As shown in Fig. 15, some samples can provide a Gaussian distribution of the mapping $\phi(\boldsymbol{p})$ from $\boldsymbol{p}$ to $\boldsymbol{v}^{\pi_p}$. Based on the distribution of $\phi$, for a desired preference $\boldsymbol{p}$, we can find a good input $\boldsymbol{p}'$ to solve

$$\max_{\boldsymbol{p}'} \mathbb{E}[\Psi(\boldsymbol{p}, \phi(\boldsymbol{p}'))] \qquad (37)$$

This procedure learns the reachable regions of the agent and calibrates the desired preference into the best input for reaching the preference. Also, it is general and can be applied to any preference control approach. Here is an empirical example for calibration:

The colored rays are preferences $\boldsymbol{p}$. The points with the same color as the preference vector $\boldsymbol{p}$ are value $\boldsymbol{v}^{\hat{\pi}_p}$ of preference conditioned policy $\pi_p$. The left plot is before calibration, $\boldsymbol{p}$ is directly used as the input for $\pi_p$. The right plot is after calibration, we know which regions can be reached, so preferences $\boldsymbol{p}$ are not all directions but reachable directions. Also, the input for $\pi_p$ is $\boldsymbol{p}'$ by solving (37), resulting in higher similairty and the CS metric improved from 0.991 to 0.997.

Figure 16: Pareto front before and after calibration.

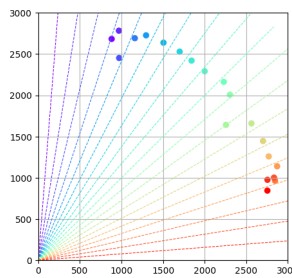
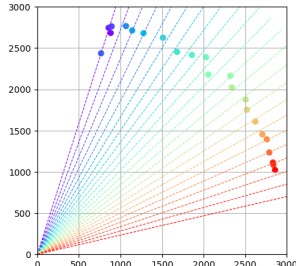

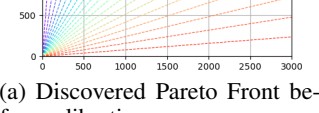

(a) Discovered Pareto Front before calibration

(b) Resampled Pareto Front after calibration

# I   THEORETICAL PROOFS

## I.1   PROOF FOR LEMMA 4.1

**Lemma 4.1.** *The similarity function $\Psi(\boldsymbol{p}, \cdot)$ is $(1 + \max_i \frac{|\boldsymbol{p}|}{|\boldsymbol{p}_i|})$ -Lipschitz smooth and $g_s(\cdot)$ is Lipschitz continuous under Assumption 4.1 and Assumption 4.3.*

*Proof.* By definition:

$$\Psi(\boldsymbol{p}, \boldsymbol{v}) = -\frac{1}{2} \| \max_i \frac{\boldsymbol{v}_i}{\boldsymbol{p}_i} \boldsymbol{p} - \boldsymbol{v} \|^2 \tag{38}$$

and

$$\nabla_{\boldsymbol{v}} \Psi(\boldsymbol{p}, \boldsymbol{v}) = \max_i \frac{\boldsymbol{v}_i}{\boldsymbol{p}_i} \boldsymbol{p} - \boldsymbol{v} = d(\boldsymbol{v}, \boldsymbol{p}) \boldsymbol{p} - \boldsymbol{v} \tag{39}$$

. where $d(\boldsymbol{v}, \boldsymbol{p})$ denotes $\max_i \frac{\boldsymbol{v}_i}{\boldsymbol{p}_i}$, we have

$$
\begin{aligned}
\|\nabla_{\boldsymbol{v}} \Psi(\boldsymbol{p}, \boldsymbol{v}) - \nabla_{\boldsymbol{v}'} \Psi(\boldsymbol{p}, \boldsymbol{v}')\| &= \|d(\boldsymbol{v}, \boldsymbol{p})\boldsymbol{p} - \boldsymbol{v} - d(\boldsymbol{v}', \boldsymbol{p})\boldsymbol{p} + \boldsymbol{v}'\| \\
&= \|d(\boldsymbol{v}, \boldsymbol{p})\boldsymbol{p} - d(\boldsymbol{v}', \boldsymbol{p})\boldsymbol{p} - (\boldsymbol{v} - \boldsymbol{v}')\| \\
&\leq |d(\boldsymbol{v}, \boldsymbol{p})\boldsymbol{p} - d(\boldsymbol{v}', \boldsymbol{p})| \|\boldsymbol{p}\| + \|\boldsymbol{v} - \boldsymbol{v}'\|
\end{aligned} \tag{40}
$$

Without loss of generality, we first consider the case where $d(\boldsymbol{v}, \boldsymbol{p}) - d(\boldsymbol{v}', \boldsymbol{p}) \geq 0$. We denote $i_{\boldsymbol{v}} = \arg\max_j \frac{\boldsymbol{v}_i}{p_i}$

$$
\begin{aligned}
\|d(\boldsymbol{v}, \boldsymbol{p})\boldsymbol{p} - d(\boldsymbol{v}', \boldsymbol{p})\boldsymbol{p}\| + \|\boldsymbol{v} - \boldsymbol{v}'\| &= (d(\boldsymbol{v}, \boldsymbol{p}) - d(\boldsymbol{v}', \boldsymbol{p})) \|\boldsymbol{p}\| + \|\boldsymbol{v} - \boldsymbol{v}'\| \\
&\leq (\frac{\boldsymbol{v}_{i_{\boldsymbol{v}}}}{|\boldsymbol{p}_{i_{\boldsymbol{v}}}|} - \frac{\boldsymbol{v}'_{i_{\boldsymbol{v}}}}{|\boldsymbol{p}_{i_{\boldsymbol{v}}}|}) \|\boldsymbol{p}\| + \|\boldsymbol{v} - \boldsymbol{v}'\| \\
&\leq \frac{\|\boldsymbol{p}\|}{|\boldsymbol{p}_{i_{\boldsymbol{v}}}|} \|\boldsymbol{v} - \boldsymbol{v}'\| + \|\boldsymbol{v} - \boldsymbol{v}'\| \\
&\leq \max_i \frac{\|\boldsymbol{p}\|}{|\boldsymbol{p}_i|} \|\boldsymbol{v} - \boldsymbol{v}'\| + \|\boldsymbol{v} - \boldsymbol{v}'\| \\
&\leq (1 + \max_i \frac{\|\boldsymbol{p}\|}{|\boldsymbol{p}_i|}) \|\boldsymbol{v} - \boldsymbol{v}'\|
\end{aligned} \tag{41}
$$

The first inequality is because $i_{\boldsymbol{v}}$ is optimal for $\boldsymbol{v}$ but not necessarily for $\boldsymbol{v}'$. The case where $d(\boldsymbol{v}, \boldsymbol{p}) - d(\boldsymbol{v}', \boldsymbol{p}) < 0$ can be proved by the same procedure by denote $i'_{\boldsymbol{v}} = \arg\max_j \frac{\boldsymbol{v}'_i}{p_i}$, then

$$
\begin{aligned}
\|d(\boldsymbol{v}, \boldsymbol{p})\boldsymbol{p} - d(\boldsymbol{v}', \boldsymbol{p})\boldsymbol{p}\| + \|\boldsymbol{v} - \boldsymbol{v}'\| &= (d(\boldsymbol{v}', \boldsymbol{p}) - d(\boldsymbol{v}, \boldsymbol{p}))\|\boldsymbol{p}\| + \|\boldsymbol{v} - \boldsymbol{v}'\| \\
&\leq (\frac{\boldsymbol{v}'_{i'_{\boldsymbol{v}}}}{|\boldsymbol{p}_{i'_{\boldsymbol{v}}}|} - \frac{\boldsymbol{v}_{i'_{\boldsymbol{v}}}}{|\boldsymbol{p}_{i'_{\boldsymbol{v}}}|})\|\boldsymbol{p}\| + \|\boldsymbol{v} - \boldsymbol{v}'\| \\
&\leq \frac{\|\boldsymbol{p}\|}{|\boldsymbol{p}_{i'_{\boldsymbol{v}}}|}\|\boldsymbol{v} - \boldsymbol{v}'\| + \|\boldsymbol{v} - \boldsymbol{v}'\| \\
&\leq \max_i \frac{\|\boldsymbol{p}\|}{|\boldsymbol{p}_i|}\|\boldsymbol{v} - \boldsymbol{v}'\| + \|\boldsymbol{v} - \boldsymbol{v}'\| \\
&\leq (1 + \max_i \frac{\|\boldsymbol{p}\|}{|\boldsymbol{p}_i|})\|\boldsymbol{v} - \boldsymbol{v}'\|.
\end{aligned}
\tag{42}
$$

Therefore, we have proven

$$
\|\nabla_{\boldsymbol{v}} \Psi(\boldsymbol{p}, \boldsymbol{v}) - \nabla_{\boldsymbol{v}'} \Psi(\boldsymbol{p}, \boldsymbol{v}')\| \leq (1 + \max_i \frac{\|\boldsymbol{p}\|}{|\boldsymbol{p}_i|})\|\boldsymbol{v} - \boldsymbol{v}'\|
\tag{43}
$$

and the similarity function for a preference $\boldsymbol{p}$ is $1 + \max_i \frac{\|\boldsymbol{p}\|}{|\boldsymbol{p}_i|}$ Lipschitz smooth.

Next, we prove that $g_s(\pi_{\boldsymbol{p}}) = G(\pi_{\boldsymbol{p}})\nabla_{\boldsymbol{v}}\Psi(\boldsymbol{p}, \boldsymbol{v}^{\pi_{\boldsymbol{p}}})$ is Lipschitz continuous.

We have:

$$
\begin{aligned}
\|g_s(x) - g_s(y)\| &= \|G(x)\nabla_{\boldsymbol{v}}\Psi(\boldsymbol{p}, \boldsymbol{v}^x) - G(y)\nabla_{\boldsymbol{v}}\Psi(\boldsymbol{p}, \boldsymbol{v}^y)\| \\
&= \|(G(x) - G(y))\nabla_{\boldsymbol{v}}\Psi(\boldsymbol{p}, \boldsymbol{v}^x) + G(y)(\nabla_{\boldsymbol{v}}\Psi(\boldsymbol{p}, \boldsymbol{v}^x) - \nabla_{\boldsymbol{v}}\Psi(\boldsymbol{p}, \boldsymbol{v}^y))\| \\
&\leq \sum_{i=1}^m \|g_i(x) - g_i(y)\|\|\nabla_{\boldsymbol{v}}\Psi(\boldsymbol{p}, \boldsymbol{v}^x)\| + \|G(y)\|\|\nabla_{\boldsymbol{v}}\Psi(\boldsymbol{p}, \boldsymbol{v}^x) - \nabla_{\boldsymbol{v}}\Psi(\boldsymbol{p}, \boldsymbol{v}^y)\|
\end{aligned}
\tag{44}
$$

where the inequality is by Cauchy-Schwartz. Since under Assumption 4.1 and Assumption 4.3, $\|G(y)\| \leq C_g$ and $\|g_i(x) - g_i(y)\| \leq l_{i,1}\|x - y\|$, and

$$
\|\nabla_{\boldsymbol{v}}\Psi(\boldsymbol{p}, \boldsymbol{v}^x) - \nabla_{\boldsymbol{v}}\Psi(\boldsymbol{p}, \boldsymbol{v}^y)\| \leq (1 + \max_i \frac{\|\boldsymbol{p}\|}{|\boldsymbol{p}_i|})\|\boldsymbol{v}^x - \boldsymbol{v}^y\| \leq (1 + \max_i \frac{\|\boldsymbol{p}\|}{|\boldsymbol{p}_i|})\|l\|\|x - y\|,
\tag{45}
$$

where $l = [l_1, l_2, ..., l_m]^T$ is the vector of Lipschitz constants of all objectives. Denoting $L_m = (1 + \max_i \frac{\|\boldsymbol{p}\|}{|\boldsymbol{p}_i|})\|l\|$, we have

$$
\|g_s(x) - g_s(y)\| \leq \left(\|\nabla_{\boldsymbol{v}}\Psi(\boldsymbol{p}, \boldsymbol{v}^x)\|\sum_{i=1}^m l_{i,1} + C_g L_m\right)\|x - y\|
\tag{46}
$$

By definition in Equation (39):

$$
\|\nabla_{\boldsymbol{v}}\Psi(\boldsymbol{p}, \boldsymbol{v}^x)\| = \|\max_i \frac{\boldsymbol{v}_i^x}{\boldsymbol{p}_i}\boldsymbol{p} - \boldsymbol{v}^x\| \leq \|\max_{\boldsymbol{v}}\max_i \frac{\boldsymbol{v}_i}{\boldsymbol{p}_i}\boldsymbol{p}\|,
\tag{47}
$$

where the inequality is because the values of $x$ should be no larger than the maximum values for the objectives. Denoting

$$
L_{\boldsymbol{p}} = \|\max_{\boldsymbol{v}}\max_i \frac{\boldsymbol{v}_i}{\boldsymbol{p}_i}\boldsymbol{p}\|,
\tag{48}
$$

we have

$$
\|g_s(x) - g_s(y)\| \leq \left(L_{\boldsymbol{p}}\sum_{i=1}^m l_{i,1} + C_g L_m\right)\|x - y\|,
\tag{49}
$$

and we define $L_s = (L_{\boldsymbol{p}}\sum_{i=1}^m l_{i,1} + C_g L_m)$. We have proven $g_s(\cdot)$ is to be $L_s$-Lipschitz continuous. Therefore, both claims of this lemma have been proven. $\qquad\square$

### I.2 PROOF FOR THEOREM 4.1 AND 4.2

Before proving Theorem 4.2, we prove Lemma I.1 for the requirements to use the proof idea in Xiao et al. (2023) for their theorem.3 and obtain Lemma I.2. To be consistent with their proof, we consider minimizing the negative value and similarity with gradient descent.

**Lemma I.1.** *Under the Assumptions 4.1-4.3, we have*

$$\|g_s(\pi_{\boldsymbol{p},t})\| \leq L_{\boldsymbol{p}}C_g, \quad \mathbb{E}\left[\|g_s(\pi_{\boldsymbol{p},t};\xi) - g_s(\pi_{\boldsymbol{p},t})\|^2\right] \leq L_{\boldsymbol{p}}^2 m\sigma^2 \tag{50}$$

$$\mathbb{E}[\|G(\pi_{\boldsymbol{p},t};\xi)^T(G(\pi_{\boldsymbol{p},t};\xi')w_t + \lambda g_s(\pi_{\boldsymbol{p},t};\xi'))\|^2] \leq \underbrace{8(m\sigma^2 + C_g^2)^2 + 8L_{\boldsymbol{p}}^2\lambda^2(m\sigma^2 + C_g^2)^2}_{C_1} \tag{51}$$

$$\mathbb{E}[\|G(\pi_{\boldsymbol{p},t};\zeta)w_t + \lambda g_s(\pi_{\boldsymbol{p},t};\zeta)\|^2] \leq \underbrace{4m\sigma^2 + 4C_g^2 + 4\lambda^2 L_{\boldsymbol{p}} m\sigma^2 + 4\lambda^2 L_{\boldsymbol{p}}^2 C_g^2}_{C_2} \tag{52}$$

$$\mathbb{E}[\|(G(\pi_{\boldsymbol{p},t})w + \lambda\pi_{\boldsymbol{p},t}(\pi_{\boldsymbol{p},t}))^T G(\pi_{\boldsymbol{p},t})\|\|w_t - w_{t+1}\|] \leq \underbrace{2(1 + L_{\boldsymbol{p}}\lambda)^2 C_g^2(m\sigma + C_g)^2}_{C_2} \tag{53}$$

*Proof.* Under the Assumptions 4.1-4.3, by (47) and (48), we have

$$\|g_s(\pi_{\boldsymbol{p},t})\| \leq \|G(\pi_{\boldsymbol{p},t})\nabla_{\boldsymbol{v}}\Psi(\boldsymbol{p},\boldsymbol{v}^{\pi_{\boldsymbol{p},t}})\| \leq L_{\boldsymbol{p}}C_g, \tag{54}$$

and

$$\mathbb{E}\left[\|g_s(\pi_{\boldsymbol{p},t};\xi) - g_s(\pi_{\boldsymbol{p},t})\|^2\right] \leq \mathbb{E}\left[\|G(\pi_{\boldsymbol{p},t};\xi) - G(\pi_{\boldsymbol{p},t})\|^2\|\nabla_{\boldsymbol{v}}\Psi(\boldsymbol{p},\boldsymbol{v}^{\pi_{\boldsymbol{p},t}})\|^2 \leq L_{\boldsymbol{p}}^2 m\delta^2\right] \tag{55}$$

The first two claims in (50) are proven.

Next, we have

$$\mathbb{E}[\|G(\pi_{\boldsymbol{p},t};\xi)^T(G(\pi_{\boldsymbol{p},t};\xi')\boldsymbol{w}_t + \lambda g_s(\pi_{\boldsymbol{p},t};\xi'))\|^2]$$
$$\overset{(i)}{\leq} 2\mathbb{E}[\underbrace{\|G(\pi_{\boldsymbol{p},t};\xi)^T G(\pi_{\boldsymbol{p},t};\xi')\boldsymbol{w}_t\|^2}_{N_1} + 2\lambda^2 \underbrace{\|G(\pi_{\boldsymbol{p},t};\xi)^T g_s(\pi_{\boldsymbol{p},t};\xi'))\|^2}_{N_2}], \tag{56}$$

where $(i)$ is by the Young's inequality. Next, we provide bounds for $\mathbb{E}[N_1]$ and $\mathbb{E}[N_2]$, separately:

$$\mathbb{E}[N_1] \overset{(i)}{\leq} \mathbb{E}[\|(G(\pi_{\boldsymbol{p},t};\xi)^T - G(\pi_{\boldsymbol{p},t})^T + G(\pi_{\boldsymbol{p},t})^T)(G(\pi_{\boldsymbol{p},t};\xi') - G(\pi_{\boldsymbol{p},t}) + G(\pi_{\boldsymbol{p},t}))\|^2]$$
$$= \mathbb{E}[\|(G(\pi_{\boldsymbol{p},t};\xi)^T - G(\pi_{\boldsymbol{p},t})^T)(G(\pi_{\boldsymbol{p},t};\xi') - G(\pi_{\boldsymbol{p},t})) + (G(\pi_{\boldsymbol{p},t};\xi)^T - G(\pi_{\boldsymbol{p},t})^T)G(\pi_{\boldsymbol{p},t})$$
$$+ G(\pi_{\boldsymbol{p},t})^T(G(\pi_{\boldsymbol{p},t};\xi') - G(\pi_{\boldsymbol{p},t})) + G(\pi_{\boldsymbol{p},t})^T G(\pi_{\boldsymbol{p},t})\|^2]$$
$$\overset{(ii)}{\leq} 4\mathbb{E}[\|G(\pi_{\boldsymbol{p},t};\xi)^T - G(\pi_{\boldsymbol{p},t})^T\|^2\|G(\pi_{\boldsymbol{p},t};\xi') - G(\pi_{\boldsymbol{p},t})\|^2 + \|G(\pi_{\boldsymbol{p},t};\xi)^T - G(\pi_{\boldsymbol{p},t})^T\|^2\|G(\pi_{\boldsymbol{p},t})\|^2$$
$$+ \|G(\pi_{\boldsymbol{p},t})^T\|^2\|(G(\pi_{\boldsymbol{p},t};\xi') - G(\pi_{\boldsymbol{p},t})\|^2 + \|G(\pi_{\boldsymbol{p},t})^T G(\pi_{\boldsymbol{p},t})\|^2]$$
$$\overset{(iii)}{\leq} 4m^2\sigma^4 + 8m\sigma^2 C_g^2 + 4C_g^4 = 4(m\sigma^2 + C_g^2)^2, \tag{57}$$

where $(i)$ follows from Cauchy–Schwarz inequality and $\boldsymbol{w}_t$ is a convex coeffiecient, $(ii)$ follows from Young's inequality and $(iii)$ follows from Assumption 4.2 and Assumption 4.3. For another term,

$$\mathbb{E}[N_2] = \mathbb{E}[\|(G(\pi_{\boldsymbol{p},t};\xi)^T - G(\pi_{\boldsymbol{p},t})^T + G(\pi_{\boldsymbol{p},t})^T)(g_s(\pi_{\boldsymbol{p},t};\xi') - g_s(\pi_{\boldsymbol{p},t}) + g_s(\pi_{\boldsymbol{p},t}))\|^2]$$
$$\overset{(i)}{\leq} 4\mathbb{E}[\|(G(\pi_{\boldsymbol{p},t};\xi)^T - G(\pi_{\boldsymbol{p},t})^T)(g_s(\pi_{\boldsymbol{p},t};\xi') - g_s(\pi_{\boldsymbol{p},t}))\|^2 + \|(G(\pi_{\boldsymbol{p},t};\xi)^T - G(\pi_{\boldsymbol{p},t})^T)g_s(\pi_{\boldsymbol{p},t})\|^2$$
$$+ \|G(\pi_{\boldsymbol{p},t})^T(g_s(\pi_{\boldsymbol{p},t};\xi') - g_s(\pi_{\boldsymbol{p},t}))\|^2 + \|G(\pi_{\boldsymbol{p},t}^T)g_s(\pi_{\boldsymbol{p},t})\|^2]$$
$$\overset{(ii)}{\leq} 4L_{\boldsymbol{p}}^2 m^2\sigma^4 + 8L_{\boldsymbol{p}}^2 m\sigma^2 C_g^2 + 4L_{\boldsymbol{p}}^2 C_g^4 = 4L_{\boldsymbol{p}}^2(m\sigma^2 + C_g^2)^2, \tag{58}$$

where $(i)$ follows from Young's inequality, $(ii)$ follows from (54) and (55). Then substituting (57) and (58) into (56), we can obtain,

$$\mathbb{E}[\|G(\pi_{\boldsymbol{p},t};\xi)^T(G(\pi_{\boldsymbol{p},t};\xi')\boldsymbol{w}_t + \lambda g_s(\pi_{\boldsymbol{p},t};\xi'))\|^2] \leq 8(m\sigma^2 + C_g^2)^2 + 8L_{\boldsymbol{p}}^2\lambda^2(m\sigma^2 + C_g^2)^2 = C_1.$$

We have proved (51). Then, we look at (52) :

$$\mathbb{E}[\|G(\pi_{\boldsymbol{p},t};\zeta)\boldsymbol{w}_t + \lambda g_s(\pi_{\boldsymbol{p},t};\zeta)\|^2]$$
$$=\mathbb{E}[\|G(\pi_{\boldsymbol{p},t};\zeta)\boldsymbol{w}_t - G(\pi_{\boldsymbol{p},t})\boldsymbol{w}_t + G(\pi_{\boldsymbol{p},t})\boldsymbol{w}_t + \lambda g_s(\pi_{\boldsymbol{p},t};\zeta) - \lambda g_s(\pi_{\boldsymbol{p},t}) + \lambda g_s(\pi_{\boldsymbol{p},t})\|^2]$$
$$\overset{(i)}{\leq}4\mathbb{E}[\|G(\pi_{\boldsymbol{p},t};\zeta) - G(\pi_{\boldsymbol{p},t})\|^2] + 4\mathbb{E}[\|G(\pi_{\boldsymbol{p},t})\|^2] + 4\lambda^2\mathbb{E}[\|g_s(\pi_{\boldsymbol{p},t};\zeta) - g_s(\pi_{\boldsymbol{p},t})\|^2]$$
$$+ 4\lambda^2\mathbb{E}[\|g_s(\pi_{\boldsymbol{p},t})\|^2]$$
$$\overset{(ii)}{\leq}\underbrace{4m\sigma^2 + 4C_g^2 + 4\lambda^2L_{\boldsymbol{p}}^2m\sigma^2 + 4\lambda^2L_{\boldsymbol{p}}^2C_g^2}_{C_2} \tag{59}$$

where $(i)$ follows from Young's inequality, and $(ii)$ follows from 54 and 55.

Finally,

$$\mathbb{E}[\|(G(\pi_{\boldsymbol{p},t})\boldsymbol{w} + \lambda\pi_{\boldsymbol{p},t}(\pi_{\boldsymbol{p},t}))^T G(\pi_{\boldsymbol{p},t})\|\|\boldsymbol{w}_t - \boldsymbol{w}_{t+1}\|]$$
$$=\beta_t\mathbb{E}[\|(G(\pi_{\boldsymbol{p},t})\boldsymbol{w} + \lambda\pi_{\boldsymbol{p},t}(\pi_{\boldsymbol{p},t}))^T G(\pi_{\boldsymbol{p},t})\|\|G(\pi_{\boldsymbol{p},t};\xi)^T(G(\pi_{\boldsymbol{p},t};\xi')\boldsymbol{w}_t + \lambda\pi_{\boldsymbol{p},t}(\theta;\xi'))\|]$$
$$\leq\beta_t\mathbb{E}[\|(G(\pi_{\boldsymbol{p},t})\boldsymbol{w} + \lambda\pi_{\boldsymbol{p},t}(\pi_{\boldsymbol{p},t}))^T G(\pi_{\boldsymbol{p},t})\|(\|G(\pi_{\boldsymbol{p},t};\xi)^T(G(\pi_{\boldsymbol{p},t};\xi')\boldsymbol{w}_t\| + \lambda\|G(\pi_{\boldsymbol{p},t};\xi)\pi_{\boldsymbol{p},t}(\theta;\xi')\|)]$$
$$=\beta_t\mathbb{E}[\|(G(\pi_{\boldsymbol{p},t})\boldsymbol{w} + \lambda\pi_{\boldsymbol{p},t}(\pi_{\boldsymbol{p},t}))^T G(\pi_{\boldsymbol{p},t})\|(\sqrt{N_1} + \sqrt{N_2})]$$
$$\leq\beta_t 2(1 + L_{\boldsymbol{p}}\lambda)^2 C_g^2(m\sigma + C_g)^2 = \beta_t C_3, \tag{60}$$

$\square$

**Lemma I.2.** *Under the Assumptions 4.1-4.3, setting $\alpha_t = \Theta(m^{-\frac{1}{2}}T^{-\frac{1}{2}})$, $\beta_t = \Theta(m^{-1}T^{-\frac{1}{2}})$, the updates by our method satisfy*

$$E[\|G(\pi_{\boldsymbol{p},t})\boldsymbol{w}_{t,\lambda} + \lambda_t g_s(\pi_{\boldsymbol{p},t})\|^2 \leq \frac{1}{\alpha_t}\mathbb{E}[l'(\pi_{\boldsymbol{p},t}) - l'(\pi_{\boldsymbol{p},t+1})] + \frac{1}{2\beta_t}\mathbb{E}[\|\boldsymbol{w}_t - \boldsymbol{w}\|^2 - \|\boldsymbol{w}_{t+1} - \boldsymbol{w}\|^2]$$
$$+ \frac{\beta_t}{2}C_1(\lambda_t) + \frac{l_1'\alpha_t}{2}C_2(\lambda_t) + \beta_t C_3(\lambda_t) \tag{61}$$

*where $w$ is a fixed convex coefficient, and*

$$l'(\pi_{\boldsymbol{p},t}) = -\boldsymbol{w}^T\boldsymbol{v}(\pi_{\boldsymbol{p},t}) - \lambda_t\Psi(\boldsymbol{p}, \pi_{\boldsymbol{p},t}), \tag{62}$$
$$l_1' = \max_i l_{i,1} + \lambda L_s \tag{63}$$
$$C_1 = 8(m\sigma^2 + C_g^2)^2 + 8L_{\boldsymbol{p}}^2\lambda^2(m\sigma^2 + C_g^2)^2, \tag{64}$$
$$C_2 = 4m\sigma^2 + 4C_g^2 + 4\lambda^2L_{\boldsymbol{p}}m\sigma^2 + 4\lambda^2L_{\boldsymbol{p}}^2C_g^2, \tag{65}$$
$$C_3 = 2(1 + L_{\boldsymbol{p}}\lambda)^2C_g^2(m\sigma + C_g)^2, \tag{66}$$

*where $L_s$ is the Lipschitz constant for $g_s(\cdot)$, defined in (49).*

Under Assumptions.(4.1-4.3), previous results show Lemma 4.1 and Lemma I.1 hold. Therefore, we can replace $g_0$ in their analysis with $g_s$ and apply their (33) in our case and it becomes Equation (61). Stochastic gradient samples like $G(\pi_{\boldsymbol{p},t},\xi)$ have been taken expectations and become $G(\pi_{\boldsymbol{p},t})$ or $\sigma$.

This is an intuitive result of convergence analysis for smooth non-convex objective functions using conventional techniques. Next we prove our main theoretical contributions based on Lemma I.2.

**Theorem 4.1.** *Under the Assumptions 4.1-4.3, setting $\alpha_t = \Theta(m^{-\frac{1}{2}}T^{-\frac{1}{2}})$, $\beta_t = \Theta(m^{-1}T^{-\frac{1}{2}})$, with a constant $\lambda$ and Lipschitz smooth similarity function $\Psi(\boldsymbol{p}, \cdot)$, we have $\frac{1}{T}\sum_{t=0}^{T-1}\mathbb{E}[\min_{\boldsymbol{w}_t}\|G(\pi_{\boldsymbol{p},t})\boldsymbol{w}_t\|] = \mathcal{O}(mT^{-\frac{1}{2}})$. To achieve an $\epsilon$-accurate Pareto stationary point, it requires $T = \mathcal{O}(m^2\epsilon^{-2})$ updates.*

*Proof.* By definition,

$$\nabla_{\boldsymbol{v}}\Psi(\boldsymbol{p}, \boldsymbol{v}) = \max_i \frac{\boldsymbol{v}_i}{\boldsymbol{p}_i}\boldsymbol{p} - \boldsymbol{v} > 0. \tag{67}$$

So $g_s(\pi_{\boldsymbol{p}}) = G(\pi_{\boldsymbol{p}})\nabla_{\boldsymbol{v}}\Psi(\boldsymbol{p}, \boldsymbol{v}^{\pi_{\boldsymbol{p}}})$ can be considered as a positive linear combination of objective gradients. We have

$$\mathbb{E}[\|G(\pi_{\boldsymbol{p},t})\boldsymbol{w}_{t,\lambda} + \lambda g_s(\pi_{\boldsymbol{p},t})\|^2 = E[\|(G(\pi_{\boldsymbol{p},t})\boldsymbol{w}_{t,\lambda} + \lambda G(\pi_{\boldsymbol{p},t})\tilde{\boldsymbol{w}}_t)\|^2]$$
$$\geq E[\min_{\boldsymbol{w}_t}\|G(\pi_{\boldsymbol{p},t})\boldsymbol{w}_t\|^2]. \tag{68}$$

For every time step $t$, by Equation (61) from Lemma I.2 and constant $\lambda$,

$$E[\min_{\boldsymbol{w}_t}\|G(\pi_{\boldsymbol{p},t})\boldsymbol{w}_t\|^2] \leq \frac{1}{\alpha_t}\mathbb{E}[l'(\pi_{\boldsymbol{p},t}) - l'(\pi_{\boldsymbol{p},t+1})] + \frac{1}{2\beta_t}\mathbb{E}[\|\boldsymbol{w}_t - \boldsymbol{w}\|^2 - \|\boldsymbol{w}_{t+1} - \boldsymbol{w}\|^2]$$
$$+ \frac{\beta_t}{2}C_1(\lambda) + \frac{l_1'\alpha_t}{2}C_2(\lambda) + \beta_t C_3(\lambda) \tag{69}$$

We take $\alpha_t = \alpha$ and $\beta_t = \beta$ as constants and telescope (69),

$$\frac{1}{T}\sum_{t=0}^{T-1}E[\min_{\boldsymbol{w}_t}\|G(\pi_{\boldsymbol{p},t})\boldsymbol{w}_t\|^2] \leq \frac{1}{\alpha T}\mathbb{E}[l'(\pi_{\boldsymbol{p},0}) - l'(\pi_{\boldsymbol{p},T})] + \frac{1}{2\beta T}\mathbb{E}[\|\boldsymbol{w}_0 - \boldsymbol{w}\|^2 - \|\boldsymbol{w}_T - \boldsymbol{w}\|^2]$$
$$+ \frac{1}{T}\sum_{t=0}^{T-1}\frac{\beta}{2}C_1(\lambda) + \frac{1}{T}\sum_{t=0}^{T-1}\frac{l_1'\alpha}{2}C_2(\lambda) + \frac{1}{T}\sum_{t=0}^{T-1}\beta C_3(\lambda)$$
$$\leq \mathcal{O}(\frac{1}{\alpha T} + \frac{1}{\beta T} + \beta m^2 + \alpha m) \tag{70}$$

By setting $\alpha = \Theta(m^{-\frac{1}{2}}T^{-\frac{1}{2}})$, $\beta = \Theta(m^{-1}T^{-\frac{1}{2}})$, we can get

$$\frac{1}{T}\sum_{t=0}^{T-1}E[\min_{\boldsymbol{w}_t}\|G(\pi_{\boldsymbol{p},t})\boldsymbol{w}_t\|^2] = \mathcal{O}(mT^{-\frac{1}{2}}).$$

To achieve an $\epsilon$-accurate Pareto stationary point, it requires $T = \mathcal{O}(m^2\epsilon^{-2})$ updates. $\qquad\square$

After proving for cases with constant $\lambda$, we need to prove further for cases with increasing $\lambda = \Theta(T^{\frac{1}{2}})$.

**Theorem 4.2.** *Under the Assumptions 4.1-4.3, setting $\alpha_t = \Theta(m^{-\frac{1}{2}}T^{-\frac{1}{2}})$, $\beta_t = \Theta(m^{-1}T^{-\frac{1}{2}})$, with a Lipshitz smooth similarity function with $g_s'(\pi_{\boldsymbol{p},t})$ being convex combination of $g_i(\pi_{\boldsymbol{p},t})$ for all $t$, there can be an increasing $\lambda = \Theta(\log T)$ and we have $\frac{1}{T}\sum_{t=0}^{T-1}\mathbb{E}[\min_{\boldsymbol{w}_t}\|G(\pi_{\boldsymbol{p},t})\boldsymbol{w}_t\|] = \mathcal{O}(mT^{-\frac{1}{2}}\log T)$.*

*Proof.* Because the similarity gradients $g_s'(\pi_{\boldsymbol{p},t})$ are convex combinations of $G(\pi_{\boldsymbol{p},t})$, let $g_s'(\pi_{\boldsymbol{p},t}) = G(\pi_{\boldsymbol{p},t})\tilde{\boldsymbol{w}}_t$ where $\tilde{\boldsymbol{w}}_t$ is a convex coefficient, then

$$\mathbb{E}[\|G(\pi_{\boldsymbol{p},t})\boldsymbol{w}_{t,\lambda} + \lambda_t g_s'(\pi_{\boldsymbol{p},t})\|^2 = E[\|(G(\pi_{\boldsymbol{p},t})\boldsymbol{w}_{t,\lambda} + \lambda_t G(\pi_{\boldsymbol{p},t})\tilde{\boldsymbol{w}}_t)\|^2]$$
$$\geq E[(1 + \lambda_t)^2\min_{\boldsymbol{w}_t}\|G(\pi_{\boldsymbol{p},t})\boldsymbol{w}_t\|^2] \tag{71}$$

holds because $(\boldsymbol{w}_{t,\lambda} + \lambda \tilde{\boldsymbol{w}}_t)$ is also a convex coefficient which can not be more optimal than $\arg\min_{\boldsymbol{w}_t} \|G(\pi_{\boldsymbol{p},t})\boldsymbol{w}_t\|^2$. For every time step $t$, by Equation (61) from Lemma I.2,

$$E[(1+\lambda_t)^2 \min_{\boldsymbol{w}_t} \|G(\pi_{\boldsymbol{p},t})\boldsymbol{w}_t\|^2] \leq \frac{1}{\alpha_t} \mathbb{E}[l'(\pi_{\boldsymbol{p},t}) - l'(\pi_{\boldsymbol{p},t+1})] + \frac{1}{2\beta_t}\mathbb{E}[\|\boldsymbol{w}_t - \boldsymbol{w}\|^2 - \|\boldsymbol{w}_{t+1} - \boldsymbol{w}\|^2]$$

$$+ \frac{\beta_t}{2}C_1(\lambda_t) + \frac{l'_1 \alpha_t}{2}C_2(\lambda_t) + \beta_t C_3(\lambda_t)$$

$$E[\min_{\boldsymbol{w}_t} \|G(\pi_{\boldsymbol{p},t})\boldsymbol{w}_t\|^2] \leq \frac{1}{\alpha_t} \mathbb{E}[l'(\pi_{\boldsymbol{p},t}) - l'(\pi_{\boldsymbol{p},t+1})] + \frac{1}{2\beta_t}\mathbb{E}[\|\boldsymbol{w}_t - \boldsymbol{w}\|^2 - \|\boldsymbol{w}_{t+1} - \boldsymbol{w}\|^2]$$

$$+ \frac{\beta_t}{2(1+\lambda_t)^2}C_1(\lambda_t) + \frac{l'_1 \alpha_t}{2(1+\lambda_t)^2}C_2(\lambda_t) + \frac{\beta_t}{(1+\lambda_t)^2}C_3(\lambda_t)$$

$$\leq \frac{1}{\alpha_t} \mathbb{E}[l'(\pi_{\boldsymbol{p},t}) - l'(\pi_{\boldsymbol{p},t+1})] + \frac{1}{2\beta_t}\mathbb{E}[\|\boldsymbol{w}_t - \boldsymbol{w}\|^2 - \|\boldsymbol{w}_{t+1} - \boldsymbol{w}\|^2]$$

$$+ \frac{\beta_t}{2\lambda_t^2}C_1(\lambda_t) + \frac{l'_1 \alpha_t}{2\lambda_t^2}C_2(\lambda_t) + \frac{\beta_t}{(1+\lambda_t)^2}C_3(\lambda_t)$$

$$\overset{(i)}{\leq} \frac{1}{\alpha_t} \mathbb{E}[l'(\pi_{\boldsymbol{p},t}) - l'(\pi_{\boldsymbol{p},t+1})] + \frac{1}{2\beta_t}\mathbb{E}[\|\boldsymbol{w}_t - \boldsymbol{w}\|^2 - \|\boldsymbol{w}_{t+1} - \boldsymbol{w}\|^2]$$

$$+ \frac{\beta_t}{2\lambda_t^2}C_1(\lambda_t) + \frac{(\max_i l_{i,1} + \lambda_T L_s)\alpha_t}{2\lambda_t^2}C_2(\lambda_t) + \frac{\beta_t}{(1+\lambda_t)^2}C_3(\lambda_t)$$

$$\tag{72}$$

where (i) is by the definition of $l'_1$ in (63). In the proofs of Theroem 1 and 3 of Xiao et al. (2023), $l'_1$ was considered constant as constant. For more rigor, we upper bound it with $\lambda_T \leq \mathcal{O}(\log T)$. We take $\alpha_t = \alpha$ and $\beta_t = \beta$ as constants and telescope (72), and by $\lambda_t = \Theta(\log t)$, we have

$$\frac{1}{T}\sum_{t=0}^{T-1} E[\min_{\boldsymbol{w}_t}\|G(\pi_{\boldsymbol{p},t})\boldsymbol{w}_t\|^2] \leq \frac{1}{\alpha T}\mathbb{E}[l'(\pi_{\boldsymbol{p},0}) - l'(\pi_{\boldsymbol{p},T})] + \frac{1}{2\beta T}\mathbb{E}[\|\boldsymbol{w}_0 - \boldsymbol{w}\|^2 - \|\boldsymbol{w}_T - \boldsymbol{w}\|^2]$$

$$+ \frac{1}{T}\sum_{t=0}^{T-1}\frac{\beta}{2\lambda_t^2}C_1(\lambda_t) + \frac{1}{T}\sum_{t=0}^{T-1}\frac{(\max_i l_{i,1} + \lambda_T L_s)\alpha}{2\lambda_t^2}C_2(\lambda_t) + \frac{1}{T}\sum_{t=0}^{T-1}\frac{\beta}{(1+\lambda_t)^2}C_3(\lambda_t)$$

$$= \mathcal{O}(\frac{1}{\alpha T} + \frac{1}{\beta T} + \frac{\beta m^2}{\log T} + \beta m^2 + \alpha m \log T + \alpha m)$$

$$\tag{73}$$

By setting $\alpha = \Theta(m^{-\frac{1}{2}}T^{-\frac{1}{2}})$, $\beta = \Theta(m^{-1}T^{-\frac{1}{2}})$, we can get

$$\frac{1}{T}\sum_{t=0}^{T-1} E[\min_{\boldsymbol{w}_t}\|G(\pi_{\boldsymbol{p},t})\boldsymbol{w}_t\|^2] = \mathcal{O}(mT^{-\frac{1}{2}}\log T),$$

and proof is done. $\qquad\square$

### I.3 Proof for Theorem 4.3 and 4.4

To be consistent with previous results in MOO literature, we consider minimizing the negative objectives and similarity with gradient descent.

**Theorem 4.3.** *Under the Assumptions 4.1-4.3, setting $\alpha_t = \Theta(m^{-\frac{1}{2}}T^{-\frac{1}{2}})$, $\beta_t = \Theta(m^{-1}T^{-\frac{1}{2}})$, with a constant $\lambda$ and Lipschitz smooth similarity function like $\Psi(\boldsymbol{p}, \cdot)$, we have $\frac{1}{T}\sum_{t=0}^{T-1}\mathbb{E}[\|g_s(\pi_{\boldsymbol{p}})\|] - \frac{2C_g^2}{\lambda^2} = \mathcal{O}(mT^{-\frac{1}{2}}).$*

*Proof.* By Equation (61) from Lemma I.2 and constant $\lambda$, we have

$$
\begin{aligned}
\mathbb{E}[\|g_s(\pi_{\boldsymbol{p},t})\|^2] \leq & \frac{2}{\lambda^2}\mathbb{E}[\|G(\pi_{\boldsymbol{p},t})\boldsymbol{w}_{t,\lambda} + \lambda g_s(\pi_{\boldsymbol{p},t})\|^2 + \frac{2}{\lambda^2}\mathbb{E}[\|G(\pi_{\boldsymbol{p},t})\boldsymbol{w}_{t,\lambda}\|^2] \\
\leq & \frac{2}{\lambda^2\alpha_t}\mathbb{E}[l'(\pi_{\boldsymbol{p},t}) - l'(\pi_{\boldsymbol{p},t+1})] + \frac{1}{\lambda^2\beta_t}\mathbb{E}[\|\boldsymbol{w}_t - \boldsymbol{w}\|^2 - \|\boldsymbol{w}_{t+1} - \boldsymbol{w}\|^2] \\
& + \frac{\beta_t}{\lambda^2}C_1(\lambda) + \frac{l'_1\alpha}{\lambda^2}C_2(\lambda) + \frac{2\beta_t}{\lambda^2}C_3(\lambda) + \frac{2C_g^2}{\lambda^2}.
\end{aligned}
\tag{74}
$$

By the definition of $l'_1$ in (63), it is a constant when $\lambda$ is constant. Take $\alpha_t = \alpha$ and $\beta_t = \beta$ as constants telescope (74), we get

$$
\begin{aligned}
\frac{1}{T}\sum_{t=t_0}^{T-1}\mathbb{E}[\|g_s(\pi_{\boldsymbol{p},t})\|^2] - \frac{2C_g^2}{\lambda^2} \leq & \frac{2}{\lambda^2\alpha T}\mathbb{E}[l'(\pi_{\boldsymbol{p},0}) - l'(\pi_{\boldsymbol{p},T})] + \frac{1}{\lambda^2\beta T}\mathbb{E}[\|\boldsymbol{w}_0 - \boldsymbol{w}\|^2 - \|\boldsymbol{w}_T - \boldsymbol{w}\|^2] \\
& + \frac{1}{T}\sum_{t=t_0}^{T-1}\frac{\beta}{\lambda^2}C_1(\lambda) + \frac{1}{T}\sum_{t=t_0}^{T-1}\frac{l'_1\alpha}{\lambda^2}C_2(\lambda) + \frac{1}{T}\sum_{t=t_0}^{T-1}\frac{2\beta}{\lambda^2}C_3(\lambda) + \frac{2C_g^2}{\lambda^2} \\
& \leq \mathcal{O}(\frac{1}{\alpha T} + \frac{1}{\beta T} + \beta m^2 + \alpha m)
\end{aligned}
\tag{75}
$$

By setting $\alpha = \Theta(m^{-\frac{1}{2}}T^{-\frac{1}{2}})$, $\beta = \Theta(m^{-1}T^{-\frac{1}{2}})$, we can get

$$
\frac{1}{T}\sum_{t=0}^{T-1}E[\min_{\boldsymbol{w}_t}\|G(\pi_{\boldsymbol{p},t})\boldsymbol{w}_t\|^2] - \frac{2C_g^2}{\lambda^2} = \mathcal{O}(mT^{-\frac{1}{2}}).
$$

To achieve an $\epsilon$-accurate stationary point, it requires $T = \mathcal{O}(m^2\epsilon^{-2})$ updates. $\qquad\square$

**Theorem 4.4.** *Under the Assumptions 4.1-4.3, setting $\alpha_t = \Theta(m^{-\frac{1}{2}}T^{-\frac{1}{2}})$, $\beta_t = \Theta(m^{-1}T^{-\frac{1}{2}})$, with a constant $\lambda$ and Lipschitz smooth similarity function like $\Psi(\boldsymbol{p}, \cdot)$, there can be an increasing $\lambda = \Theta(T^{\frac{1}{2}})$ and we have $\frac{1}{T}\sum_{t=0}^{T-1}\mathbb{E}[\|g_s(\pi_{\boldsymbol{p}})\|] = \mathcal{O}(mT^{-\frac{1}{2}}\log T)$.*

*Proof.* Suppose for all time steps $t > t_0$, $\lambda_t \geq 1$, by Equation (74) we have

$$
\begin{aligned}
\mathbb{E}[\|g_s(\pi_{\boldsymbol{p},t})\|^2] \leq & \frac{2}{\lambda_t^2}\mathbb{E}[\|G(\pi_{\boldsymbol{p},t})\boldsymbol{w}_{t,\lambda} + \lambda_t g_s(\pi_{\boldsymbol{p},t})\|^2 + \frac{2}{\lambda_t}\mathbb{E}[\|G(\pi_{\boldsymbol{p},t})\boldsymbol{w}_{t,\lambda}\|^2] \\
\leq & \frac{2}{\alpha_t}\mathbb{E}[l'(\pi_{\boldsymbol{p},t}) - l'(\pi_{\boldsymbol{p},t+1})] + \frac{1}{\beta_t}\mathbb{E}[\|\boldsymbol{w}_t - \boldsymbol{w}\|^2 - \|\boldsymbol{w}_{t+1} - \boldsymbol{w}\|^2] \\
& + \frac{\beta_t}{\lambda_t^2}C_1(\lambda_t) + \frac{l'_1\alpha}{\lambda_t^2}C_2(\lambda_t) + \frac{2\beta_t}{\lambda_t^2}C_3(\lambda_t) + \frac{2C_g^2}{\lambda_t^2}.
\end{aligned}
\tag{76}
$$

Take $\alpha_t = \alpha$ and $\beta_t = \beta$ as constants telescope (76) and by the definition of $l'_1$ in (63) we get

$$
\begin{aligned}
\sum_{t=t_0}^{T-1}\mathbb{E}[\|g_s(\pi_{\boldsymbol{p},t})\|^2] \leq & \sum_{t=t_0}^{T-1}\frac{2}{\alpha}\mathbb{E}[l'(\pi_{\boldsymbol{p},0}) - l'(\pi_{\boldsymbol{p},T})] + \sum_{t=t_0}^{T-1}\frac{1}{\beta}\mathbb{E}[\|\boldsymbol{w}_0 - \boldsymbol{w}\|^2 - \|\boldsymbol{w}_T - \boldsymbol{w}\|^2] \\
& + \sum_{t=t_0}^{T-1}\frac{\beta}{\lambda_t^2}C_1(\lambda_t) + \sum_{t=t_0}^{T-1}\frac{l'_1\alpha}{\lambda_t^2}C_2(\lambda_t) + \sum_{t=t_0}^{T-1}\frac{2\beta}{\lambda_t^2}C_3(\lambda_t) + \sum_{t=t_0}^{T-1}\frac{2C_g^2}{\lambda_t^2} \\
\leq & \sum_{t=t_0}^{T-1}\frac{2}{\alpha}\mathbb{E}[l'(\pi_{\boldsymbol{p},0}) - l'(\pi_{\boldsymbol{p},T})] + \sum_{t=t_0}^{T-1}\frac{1}{\beta}\mathbb{E}[\|\boldsymbol{w}_0 - \boldsymbol{w}\|^2 - \|\boldsymbol{w}_T - \boldsymbol{w}\|^2] \\
& + \sum_{t=t_0}^{T-1}\frac{\beta}{\lambda_t^2}C_1(\lambda_t) + \sum_{t=t_0}^{T-1}\frac{(\max_i l_{i,1} + \lambda_T L_s)\alpha}{\lambda_t^2}C_2(\lambda_t) + \sum_{t=t_0}^{T-1}\frac{2\beta}{\lambda_t^2}C_3(\lambda_t) + \sum_{t=t_0}^{T-1}\frac{2C_g^2}{\lambda_t^2},
\end{aligned}
\tag{77}
$$

then by $\lambda_t = \Theta(\log t)$ we have:

$$
\frac{1}{T} \sum_{t=t_0}^{T-1} \mathbb{E}[\|g_s(\pi_{\boldsymbol{p},t})\|^2] \leq \frac{1}{T} \sum_{t=t_0}^{T-1} \frac{2}{\alpha} \mathbb{E}[l'(\pi_{\boldsymbol{p},0}) - l'(\pi_{\boldsymbol{p},T})] + \frac{1}{T} \sum_{t=t_0}^{T-1} \frac{1}{\beta} \mathbb{E}[\|\boldsymbol{w}_0 - \boldsymbol{w}\|^2 - \|\boldsymbol{w}_T - \boldsymbol{w}\|^2]
$$

$$
+ \frac{1}{T} \sum_{t=t_0}^{T-1} \frac{\beta}{\lambda_t^2} C_1(\lambda_t) + \frac{1}{T} \sum_{t=t_0}^{T-1} \frac{(\max_i l_{i,1} + \lambda_T L_s)\alpha}{\lambda_t^2} C_2(\lambda_t) + \frac{1}{T} \sum_{t=t_0}^{T-1} \frac{2\beta}{\lambda_t^2} C_3(\lambda_t) + \frac{1}{T} \sum_{t=t_0}^{T-1} \frac{2C_g^2}{\lambda_t^2}
$$

$$
= \mathcal{O}\left(\frac{1}{\alpha T} + \frac{1}{\beta T} + \frac{\beta m^2}{\log T} + \beta m^2 + \alpha m \log T + \alpha m + \frac{1}{(\log T)^2}\right)
$$

$$\tag{78}$$

Adding the terms before $t_0$, we have

$$
\frac{1}{T} \sum_{t=0}^{T-1} \mathbb{E}[\|g_s(\pi_{\boldsymbol{p},t})\|^2] = \frac{1}{T} \sum_{t=0}^{t_0} \mathbb{E}[\|g_s(\pi_{\boldsymbol{p},t})\|^2] + \frac{1}{T} \sum_{t=t_0}^{T-1} \mathbb{E}[\|g_s(\pi_{\boldsymbol{p},t})\|^2]
$$

$$
\leq \mathcal{O}\left(\frac{1}{T} + \frac{1}{\alpha T} + \frac{1}{\beta T} + \frac{\beta m^2}{\log T} + \beta m^2 + \alpha m \log T + \alpha m + \frac{1}{(\log T)^2}\right).
$$

$$\tag{79}$$

By setting $\alpha = \Theta(m^{-\frac{1}{2}} T^{-\frac{1}{2}})$, $\beta = \Theta(m^{-1} T^{-\frac{1}{2}})$, we can get

$$
\frac{1}{T} \sum_{t=0}^{T-1} \mathbb{E}[\|g_s(\pi_{\boldsymbol{p},t})\|^2] = \mathcal{O}(m T^{-\frac{1}{2}} \log T),
$$

$$\tag{80}$$

and the proof is done. $\qquad\square$

