# OpenReview forum: "Customizing Reinforcement Learning Agent with Multi-Objective Preference Control"
_ICLR.cc/2025/Conference — Submitted to ICLR 2025_

### Official Review · Reviewer_tiB3 · 2024-10-23

**Soundness:** 2
**Presentation:** 2
**Contribution:** 2
**Rating:** 5
**Confidence:** 4

**Summary:**

This paper studies multi-objective reinforcement learning (MORL) algorithms that can be controlled to produce policies with different trade-offs between conflicting objectives. In particular, the paper tackles the case in which the policy is conditioned on a preference vector $p$ that controls the desired trade-offs. The paper introduces a novel method, named Preference Control RL (PreCo), which learns a policy that encourages the preference vector $p$ to have high similarity with the value vector of the corresponding policy. Experiments were performed in discrete toy domains as well as in multi-objective versions of the Mujoco benchmark to assess the properties of the method proposed.

**Strengths:**

- The problem of designing MORL algorithms with better controllability is of high relevance in the RL community, and the idea of using a similarity function to achieve that is novel to the best of my knowledge.
- The experiments consider many different MORL problems with diverse characteristics and complexities.

**Weaknesses:**

- The paper has a significant problem of clarity, to the point of being very difficult to follow what is being proposed in terms of theory, mathematical notation, and algorithms. I discuss this in my Questions below.
- The experiments do not provide confidence intervals or information regarding the number of random seeds used to compute the metrics. Hence, it is not possible to assess whether the results are not due by chance.

**Questions:**

Below, I have some questions and constructive feedback to the authors:

1) “... Alegre et al. (2023) require learning multiple models to identify the Pareto front”
This is incorrect. Alegre et al. (2023) introduce a version of their method that uses a single policy conditioned on the preferences $w$, $\pi(s|w)$. This is also done in many state-of-the-art MORL algorithms.

2) The authors claim to learn a “meta-policy”. However, I do not think “meta-policy” is the appropriate term since the policy learned outputs actions as defined in the original action space of the MDP. In RL, meta-policies are policies that control a standard RL policy by learning in the space of meta-actions that are different than the regular actions.

3) In Figure 1, the PFs are “convex” instead of “concave”. That is, linear scalarization is only able to reach points in the convex part of the PF. The paper is confusing both terms.

4) The idea of training a policy conditioned on agent preferences is well-studied and applied in the MORL literature. The authors are proposing a different representation of the preferences. However, the paper does not discuss that previous MORL algorithms also can control MORL policies via a vector representation.

5) “we sample $p\in\mathcal{P}$ uniformly”. What is the domain of $p$? How do you sample it uniformly from this space?

6) How is the value of $\lambda$ selected in Equations 6 and 7?

7) What does it mean to solve the min-norm problem in the third step at “policy level” vs. “parameter level”? This is not clear.

8) In Definition 4.1, the preferences $p$ are vectors such that their elements sum to 1. However, $v$ does not have this constraint and can have very different magnitudes in its elements. Hence, if the maximum element of $v$ has a value of $10$, this value could always be selected in the max in Eq. 8, and the preference vector would be pushed towards $v$. Is this correct?

9) “$\Pi_{W}$ means the projection to the set of convex coefficients.” It is not clear what does this mean. Please provide a more detailed explanation. What are the coefficients? How is this projection computed?

10) Algorithm 1 and its explanation are very difficult to follow. What is the data collected in line 230? What are the variables $\zeta$?

11) Equations 9 and 10 require some intuitive explanation. It is not clear how $G$ is computed during the RL learning agent training. For instance, the expected value in Eq. 9 is w.r.t what distribution?

12) Regarding Remark 4.1, how is the value of $\lambda$ increased?

13) In Table 1, how many random seeds were used to compute these metrics? The authors should also provide confidence intervals or dispersions metrics. It is not possible to infer whether the results are due to chance.

14) In Figure 4, why did linear scalarization only achieve a single point? This is very strange since previous MORL algorithms based on linear scalarization have been employed in this problem and have been able to identify many points in the PF.

15) The claim that LS agents are uncontrollable by $p$ is not true given that many previous works have proposed MORL agents that generate different Pareto-optimal solutions conditioned on a preference vector $w$.

16) Based on the results from Figures 5 and 6, and given that no confidence intervals were provided, it is not possible to infer that PreCo results in better Hypervolume than the competitors.

17) The paper has a considerable number of grammar issues. I suggest the authors to carefully review the paper.

**Details Of Ethics Concerns:**

I did not identify ethical concerns that need to be addressed in this paper.

---

> ### Author Response · Authors · 2024-11-19
> **A response (1/3)**
>
> Thanks very much for your review and constructive feedback!
> ***
> We would like to briefly clarify our theoretical contributions:
>
> * We have theoretically proven that our proposed PreCo has the following properties:
>   * It converges to Pareto stationary solutions.
>    * Its solutions align with user preferences (controllability).
>    * It converges under stochastic gradients."
>
> These properties are empirically demonstrated via a toy example of multi-objective optimization provided in Appendix F (code available in the [anonymous link 1](https://anonymous.4open.science/r/ICLR255131/)).  The properties suit MORL well, as previous methods based on linear scalarization could fail for preference alignment (Fig. 1), and RL's stochastic gradients often have high variance.
>
> The experimental results are averaged over five random seeds. We provide some easy-to-run code in this [anonymous link 2](https://anonymous.4open.science/r/PCRL-6E05), and the full code will be released upon acceptance.
>
> We hope our responses to your questions below further clarify our contributions.
> ***
> # Answers to the questions
> ***
>
>  **A1:** Thanks for pointing out this issue and we will make adjustments accordingly. But Alegre et al. (2023) [1] is still an LS method that inherently has the limitation discussed in the cases of Figure 1, so it is not the best choice for preference alignment.
>
> ***
> **A2:** "Meta-policy" has been used in [2] for MORL.
>
> ***
>
> **A3:**We consistently consider **maximization** for MORL that linear scalarization can reach the part of the Pareto front with concave curvature. "Conve" or "concave" is often used to describe the curvature of the Pareto front [3].
> ***
> **A4:** We have discussed it in the "Related works" section in the Appendix. A.
> ***
> **A5:** As defined in paragraph "Preference control" of section 2, the domain of $p$ is $\mathcal{P}:=\{\mathbf{p}\in \mathbb{R}^m: \mathbf{p}^T\mathbf{1}=1,\mathbf{p}\geq0\}$, where $m$ is the number of objectives. The uniform sampling can be efficiently implemented by "np.random.dirichlet([1 for i in range(m)])"
> ***
> **A6:** $\lambda$ is chosen as a hyperparameter.
>  ***
> **A7:** Appendix.D discussed the topic “policy level” vs. “parameter level” in detail. Consider the PPO example in Appendix.D.2, its parameter level gradient is
> $$
> \nabla_{\theta} \hat{\mathbf{v}}^{\pi_p}=E\left[\frac{\nabla_\theta \pi(a|s,p)}{\pi(a|s,p)}\hat{\mathbf{A}}(s,a,p)\right],
> $$
> and denote its policy level gradient as $\mathbf{d}=\nabla_{\pi_p} \hat{\mathbf{v}}^{\pi_p}$, we have
> $$
> \mathbf{d}(s,a)=\frac{1}{\pi(a|s,\mathbf{p})}\hat{\mathbf{A}}(s,a,\mathbf{p})
> $$
> for every $(s,a)$ sample.
> “Policy level” gradient $\mathbf{d}$ has a dimension of batch size for every PPO update while the “parameter level” gradient $\nabla_{\theta} \hat{\mathbf{v}^{\pi_p}}$ has a size of model parameter size. The parameter size of neural network models is often much larger than the batch size. We provided a diagram of the backward path for more intuition in the [anonymous link 2](https://anonymous.4open.science/r/PCRL-6E05)
> ***
> **A8:** No it is not correct. There is no constraint on the values. If $\mathbf{v}=[v_1,v_2]=[10,1]$ and $p=[0.8,0.2]$, the maximum element will be $\frac{10}{0.8}$, then $\max_i \frac{v_i}{p_i}p=[10,2.5]$, and we obtain $$\nabla_{v_1}\Psi(p,\mathbf{v})=10-10=0$$ and $$\nabla_{v_2}\Psi(p,\mathbf{v})=2.5-1=1.5.$$ Therefore, $v_2$ will be optimized to push $\mathbf{v}$ towards $[10,2.5]$.
> As mentioned right after Def.4.1, a visualization in Appendix E could help for an intuitive understanding of this similarity function.
> ***
> **A9:** $w$ is the coefficient to combine the objective gradients into a similarity gradient. After updating $w$ by adding a gradient, it might no longer be a convex coefficient(similar to the preference domain $\mathcal{P}$, its sum should equal 1 and its elements should be nonnegative). The projection just means to find the point $w'$ within the set of convex coefficients with minimum L2 distance to $w$, $$\min_{w'\in \mathcal{P}}||w'-w||^2,$$ which is a common problem that can be efficiently implemented by popular packages like "scipy.optimize".
> ***
> **A10:** In algorithm 1, we consider the stochastic setting, so policy gradient samples are random variables. Each data (a batch of transitions in the RL setting) gives us a gradient sample. $G(\pi;\zeta)$ is a sample of the gradient used for $\pi$ update and $G(\pi;\xi)$ is a sample of gradient used for $w$ update.
> Thanks for pointing out that algorithm 1 is not entirely straightforward. We provide an intuitive algorithm next page and will include in the revisions for better clarity.
> ***
> **A11:** The expectation in Eq. 9 is taken w.r.t. the distribution of data $\xi$. For the RL setting, $\xi$ is a random variable of the batch of transitions that calculates the gradient for a policy update. The computation of $G$ for mainstream RL algorithms is discussed in more detail in Appendix.D.
> ***

---

> ### Author Response · Authors · 2024-11-19
> **A response (2/3)**
>
> ***
> **A12:** Regarding Remark 4.1, the upper limit of $\lambda$ can be predefined as a hyperparameter, then in practice, you can linearly increase $\lambda$. Once $\lambda$ reaches the upper limit it stays constant.
> ***
> **A13:** 5 random seeds were used for mean values. For better reproducibility, we rerun the experiments with seed [1,2,3,10,20] and use fixed hyperparameter for all reward dimmensions. We get the results below:
>
>
> #### Hypervolume:
> |method|3D|4D|5D|6D|
> |-|-|-|-|-|
> |LS|$122.23\pm 14.56$|	$326.20\pm 133.28$|$1589.66\pm 294.55$|$5741.79\pm 877.41$|
> |SDMgrad|$144.54\pm 9.03$|	$661.78\pm 22.21$|	$2743.73\pm 92.27$|	$13296.96\pm 145.77$|
> |CAGrad|$139.87\pm 18.43$|$301.08\pm 62.67$|$1229.03\pm 142.08$|	$4931.13\pm 813.10$|
> |EPO|$\mathbf{147.21\pm 10.79}$|$1042.87\pm 50.17$|	$3984.30\pm 484.54$|$14973.62\pm 2276.65$|
> |PreCo(Ours)|$146.04\pm 10.02$|$\mathbf{1091.17\pm 21.94}$|	$\mathbf{4329.35\pm 209.79}$|$\mathbf{15605.90\pm 752.97}$|
> #### Cosine similarity:
> |method|3D|4D|5D|6D|
> |-|-|-|-|-|
> |LS|$0.784\pm 0.025$|		$0.756\pm 0.051$|	$0.741\pm 0.014$|	$0.718\pm 0.004$|
> |SDMgrad|$0.806\pm 0.027$|	$0.743\pm 0.005$|	$0.715\pm 0.002$|	$0.664\pm 0.009$	|
> |CAGrad|$	0.776\pm 0.012$|$0.710\pm 0.030$|$0.691\pm 0.001$|$0.602\pm 0.091$|
> |EPO|$0.837\pm 0.016$|$0.894\pm 0.016$|$0.860\pm 0.025$	|$0.774\pm 0.030$|
> |PreCo(Ours)|$\mathbf{0.838\pm 0.018}$|	$\mathbf{0.907\pm 0.009}$|$\mathbf{0.873\pm 0.006}$|	$\mathbf{0.779\pm 0.026}$|
>
> CAGrad only guarantees convergence when the reference gradient is defined as the mean gradient across all objectives. However, when using the similarity gradient as the reference, its convergence is not theoretically established. Consequently, without specific tuning tailored to each setting, it tends to exhibit poor performance.
> ***
> **A14:** For 3 out of 5 seeds, Linear Scalarization (LS) learns a single point, similar to the case shown in Figure 4. In the remaining two seeds, most values also overlap at a single point. These results are reasonable because the red point is indeed optimal for $\max_v v^Tp$ for most $p$. (shown by the rays in Figure 4(a)). Especially when preferences are uniformly sampled during LS training, a value at the red point, independent of the preference condition, represents an obvious local optimum.
> Even in the ideal case, LS methods like [1] can only identify points within the "Convex Coverage Set (CCS)." However, preferences can target Pareto-optimal points that lie outside the CCS, such as most of the blue points in Figure 4.
> ***
> **A15:** We **did not** claim that LS agents are uncontrollable by $p$ **for all cases**, only in the 3D fruit-tree case it is completely uncontrollable. Additionally, by the fact that LS methods can only find the concave part for maximization, they inherently lack controllability, which was shown in Fig.1.
> ***
> **A16:** It was calculated by mean values of 5 seeds. We now finetune the hyperparameters for all methods and rerun the experiments with seeds [1,2,3,10,20]. We have obtained the results below and will include the full results in the revisions:
>
> MO-Reacher:
> |seed|Hypervolume（1e8）|CosineSimilarity|
> |-----|---|-----|
> |LS|$15.66\pm8.03$|$0.652\pm 3.02e-2$|
> |SDMgrad|$20.37\pm 0.37$|$0.761\pm 0.13e-2$|
> |EPO|$9.81\pm 3.11$|$0.845\pm 7.8e-2$|
> |CAGrad|$13.46\pm 9.71$|$0.760\pm 0.20e-2$|
> |PreCo(Ours)|$\mathbf{33.12}\pm 7.02$|$\mathbf{0.863}\pm 5.74e-2$|
>
>
> MO-Hopper:
> |seed|Hypervolume（1e6）|CosineSimilarity|
> |-----|---|-----|
> |LS|$2.63\pm 0.12$|$0.933\pm 0.007$|
> |CAGrad|$2.53\pm 0.28$|$0.932\pm 0.024$|
> |PreCo(Ours)|$2.63\pm 0.21$|$\mathbf{0.955}\pm 0.011$|
>
> The results demonstrate that our proposed PreCo outperforms the baselines in both hypervolume and cosine similarity, particularly in tasks with a larger number of objectives. While LS may achieve high hypervolume in simpler 2-objective tasks such as Hopper, its controllability remains inferior to that of PreCo.
> ***
> **A17:** Thanks for the valuable feedback and we will carefully review it for better clarity.
> ***
>
> # Reference
> > [1] Alegre et al. (2023) "Sample-Efficient Multi-Objective Learning via
> Generalized Policy Improvement Prioritization"
>
> > [2] Chen et al.(2019) "Meta-Learning for Multi-objective Reinforcement Learning"
>
>
> > [3] Zhanget al.(2023) "Hypervolume Maximization: A Geometric View of Pareto Set Learning"

---

> > ### Author Response · Authors · 2024-11-20
> > **A response(3/3): Pseudo Code**
> >
> > ```python
> > def train(N,E,agent):
> >     # N: Number of preference samples
> >     # E: Episodes per preference sample
> >     # Agent should have preference-conditioned actor and critic
> >     # The critic should output a vector of multi-objective values
> >
> >     for n in range(N):                     # Loop over preference samples
> >         Sample preference p                # Sample a preference p
> >         for e in range(E):                 # Loop over episodes for the this preference
> >             Rollout with policy conditioned on p  # Rollout with the current preference
> >             Store transitions (s,a,r_vec,p) in the buffer  # Store transitions
> >
> >         Sample (s,a,r_vec,p) from buffer       # Sample transitions from buffer
> >         Update critic by the samples           # Approximate objective values
> >         Calculate policy-level gradient        # Eq.20 for TD3, Eq.26/32 for PPO
> >         Estimated values of the agent          # Eq.4 for TD3, Eq.5 for PPO
> >         Calculate similarity gradient          # Eq.8
> >         Solve Eq.6 using policy-level gradient   # Solve the problem for PreCo update
> >         Update actor parameter by solution of Eq.6   # Eq.21/27 or Eq.29
> > ```

---

> > > ### Comment · Reviewer_tiB3 · 2024-11-20
> > >
> > > I thank the authors for their careful response. Below, I have a few more comments:
> > >
> > > A1: First, when assuming stochastic policies, the Pareto front is always convex (see (Lu et al., 2023)). Hence, the problem of concave points is not relevant since the paper also learns stochastic policies.
> > > Moreover, the fact that the solutions are not at the intersection of the rays and the Pareto front does not mean that methods using LS can not be controllable. For instance, when you increase the weight for some objective, the solution of that objective will also increase, allowing users to get different solutions with different weights. Why is the intersection of the preference rays and the PF of critical importance?
> > >
> > > A2: In [2], it is a meta-policy because their work employs a meta-learning algorithm to learn policies. It is not clear why the policy learned in this paper is a "meta-policy". If it is, then all MORL algorithms also learn meta-policies.
> > >
> > > A3: In the MORL literature, all works define that LS learns policies in the convex region of the PF (hence the solution set named convex coverage set (CCS)), and not in the concave regions. See (Lu et al. 2023) or (Abels et al. 2019), for instance.
> > >
> > > A4: This should appear in the main text since they are the main relevant competing methods.
> > >
> > > Regarding the experiments, the results for LS are still unexpected. The work that introduced the Fruit-Tree benchmark (Yang et al. 2019) is able to find most (if not all) the optimal solutions. It seems that the limitations of LS encountered in the experiments are probably due to function approximation errors or a poor underlying RL algorithm used to learn the policies.
> > >
> > > Finally, I do not feel comfortable with increasing my score since *all the experimental results* in the paper were presented without any statistical confidence levels (e.g., standard error, standard deviation), and the paper does not seem ready for publication. However, I strongly suggest the authors take the feedback of the reviewers into account and re-submit an improved version of the paper to another venue, since the idea has strong relevance potential to the community.

---

> ### Author Response · Authors · 2024-11-21
> **Second response**
>
> Thanks for the feedback and suggestions!
> ***
> First of all, there seems to be a major theoretical misunderstanding to clarify:
>
> * **Solutions in the convex coverage set (CCS) are not the only Pareto optimal solutions**
>
> For a Pareto front that is not strictly convex, like a polytope shape, the CCS that Linear Scalarization (LS) methods aims to learn are the solutions at the "vertices" rather than the "edges", while solutions at the "edges" are also Pareto-optimal and contribute to the hypervolume.
>
> Beside proving convexity for stochastic policies,   *Lu et al., (2023)* also claim that LS often fails to find all Pareto optimal solutions for these not strictly convex cases, and it is necessary to add a strongly convex augmentation for more LS solutions. (strictly convex is a necessary condition of strongly convex)
>
> The intuitive illustrations can be found in Figure 10 of *Lu et al., (2023)* and  Figure 4 of our paper.
>
> ***
> And there can be a very simple but also practical example to demonstrate this:
>
> **Example:** There is only one state, two actions $[a_1,a_2]$, and two objectives. At every step, if $a_1$ is made, the agent receives $[r_1(a_1)=1,r_2(a_1)=0]$ and if $a_2$ is made, the agent receives  $[r_1(a_2)=0,r_2(a_2)=1]$
>
> For this example, all stochastic policies are Pareto optimal and their values $\mathbf{v}=\mathbb{E}_a[[r_1(a),r_2(a)]]$ all lie on the line $v_2=1-v_1$. The Pareto front is convex but not strictly convex.
>
> LS methods only learns the two CCS solutions: $[\pi_1(a_1)=1,\pi_1(a_2)=0]$ for $w_1 \geq w_2$ and $[\pi_2(a_1)=0,\pi_2(a_2)=1]$ for $w_2 \geq w_1$.
>
> **Therefore, for this example, LS methods find only $2$ CCS policies, even though there are an infinite number of Pareto optimal policies.** And these two CCS policies are not aligned with most preferences.
> ***
> In addition, for continuous control tasks, the policies are often not "stochastic"[3].
> ***
> ### **About experimental results:**
>
> Based on the clarified point above,
>
> ***"Fruit-Tree benchmark (Yang et al. 2019) is able to find most (if not all) the optimal solutions."* is incorrect.**
>
> (Yang et al. 2019) only reported the Coverage Ratio of CCS solutions, not the Pareto-optimal solutions. As shown by our Fig.4, the Pareto front of FTN is often not strictly convex and LS can not discover all optimal solutions in practice.
>
> For strictly convex two-objective cases such as Hopper and Ant, our implementation LS does exhibit similar hypervolumes to PreCo. Therefore, our code-level implementation is not the problem.
>
> We have updated the experimental results with statistic metric standard deviation and provided easy-to-run [code](https://anonymous.4open.science/r/PCRL-6E05) and seed. We hope this could address your concerns.
> ***
> ### **About previous answers:**
> **A1&A2:** Our method is trained with varying preferences (analogous to different 'tasks' in a meta-learning setting). At test time, our learned agent aims to adapt to the user's preferences in a zero-shot manner by treating these preferences as conditional input. This makes the *"intersection of preference rays with the Pareto front"* **(controllability)** critically important. Even in strictly convex cases like Figure 1(c), where LS successfully learns all Pareto-optimal policies, there can still be a significant mismatch between the achieved values and the user's preferences, leading to high adaptation costs.
>
> By *"For instance, when you increase the weight ...  get different solutions with different weights."* you suggested that adjusting the conditional input can help identify a preferred policy at test time. *(In fact, we have already proposed a principled approach for this purpose in Appendix G. )*
>
> However, this approach still inherently favors policies with higher hypervolume and a greater number of Pareto-optimal solutions. **As mentioned before, CCS does not represent the full Pareto front.** Consequently, the user is not guaranteed to find their preferred policy simply by exploring different conditional inputs. Even in strictly convex cases like Figure 1(c), **the density of practically learned policies can vary significantly across different preferences.** As a result, there remains the possibility that users may fail to find their preferred preference in low-density regions.
>
> ***
> **A3** Thank you, we will adopt 'convex' for better consistency in terminology.
> ***
> **A4** Thank you for the suggestions and we will consider the rearrangement of sections.
> ***
> ### **References**
> *Lu et al., (2023) "Multi-objective reinforcement learning: Convexity, stationarity and pareto optimality"*
>
> *Yang et al. (2019) "A generalized algorithm for multi-objective reinforcement learning and policy adaptation"*
>
> *[2] Chen et al.(2019) "Meta-Learning for Multi-objective Reinforcement Learning"*
>
> *[3] Lillicrap 2015 "Continuous control with deep reinforcement learning"*
> ***
> Thank you very much and the discussions above will be added as a Q&A section in revisions for better clarity.

---

> ### Author Response · Authors · 2024-11-23
> **Kind Reminder and looking foward for further discussion**
>
> Dear reviewer tiB3,
>
> Thanks again for your time, effort, and valuable suggestions to improve our work! We have carefully addressed your initial concerns and revised the manuscript accordingly. We kindly ask that you consider the clarifications provided in our second response, as it directly addresses a key point that is crucial for understanding the contributions of our work. We are looking forward to hearing you and will be happy to answer any further questions or concerns you may have.
>
> Best regards,

---

> > ### Comment · Reviewer_tiB3 · 2024-11-23
> >
> > I thank the authors for their response. I, however, maintain my opinion that the paper is not yet ready for publication. Below I discuss a few of the reasons:
> >
> > * Clarity: As also pointed out by other reviewers, there were many points that required clarification, in terms of mathematical definitions and algorithmic explanations.
> >
> > * Missing comparison with state-of-the-art MORL methods: The method is not compared with methods such as Envelope Q-learning, GPI-LS, or CAPQL, which are state-of-the-art MORL algorithms. In Figure 8, for instance, the LS method did not learn to reach any of the targets, while current state-of-the-art LS MORL methods are able to learn many optimal policies in Reacher.
> >
> > * Unclear advantage of the proposed method: Based on Figures 5 and 6, the proposed technique does not outperform LS on hypervolume.

---

> ### Author Response · Authors · 2024-11-23
> **3rd response**
>
> We are pleased that some of your initial concerns have been resolved. Below, we address your newly raised points:
>
> ## Clarity:
> ***
> The main concerns raised by other reviewers [caBw, nUpd] have been clarified and addressed. They both increased their ratings Reviewer nUpd explicitly acknowledged improved clarity after the revision and increased their confidence accordingly.
> ***
> ## Missing comparison:
> ***
> Our proposed method is a more general, high-level approach applicable to both value-based and policy-based methods, unlike the three Linear Scalarization (LS) methods [1][2][3] you mentioned, which are specifically designed for value-based RL. Envelope Q-learning [1] is limited to discrete action spaces and relies on **Hindsight Experience Replay** (HER)[4], GPI-LS [2] involves a specific training **curriculum** and CAPQL[3] needs **reward augmentation**—none of which are directly applicable to policy-based methods like PPO.
>
> To ensure a **fair comparison independent of HER, reward augmentation, and training curriculum**, we implemented two LS baselines: a basic, general LS method and SDMGrad as an ablation study (replacing similarity with LS for PreCo). Further direct comparisons to LS methods with lower-level modifications are unnecessary and would be unfair at this stage. (Other reviewers [caBw] highlighted the impact of HER on fair comparisons, and [nUpd] understands the limitations of LS.)
>
> Even so, For the typical discrete 6d fruit tree environment applicable to Envelope Q-learning[1], even with HER techniques, their reported hypervolume ($8427.51$) was **significantly lower** than both our proposed PreCo ($1.56e4$) and our LS baseline SDMGrad ($1.33e4$).
>
> About the results of Reacher,  Figure 8 shows the typical local optimum they learn for most runs. This is reasonable, as even in the toy example in Appendix F, due to stochasticity, many methods fail (Figure 11). Moreover, [1][2][3] did not report the results of Reacher. **Could you please provide clear references for your claim?** Regarding the LS method that you mentioned to learn many optimal policies, are they training the LS agent with uniformly sampled preferences? What are their evaluation metrics? Are they using the HER technique? This would be important to keep the discussion constructive.
> ***
> ## Unclear advantage of the proposed method:
> ***
> The main advantage we claim is the **controllability of preferences**, measured by the cosine similarity between preferences and values. The higher cosine similarity of PreCo compared to the two LS baselines supports our contribution.
>
> Additionally, as clarified previously, in cases with strictly convex, and especially when there are only two objective numbers, LS can learn all Pareto-optimal policies. The hypervolume data demonstrates that **our implementation of LS is fair and adequate**.
>
> The lower hypervolume of LS for cases with less convex Pareto fronts and higher objective numbers, and the higher hypervolume of LS for cases with more convex Pareto fronts and lower objective numbers, align precisely with the theoretical insights we wished to clarify in our second response.
>
> ***
> [1] Yang et al. (2019) "A generalized algorithm for multi-objective reinforcement learning and policy adaptation"
>
> [2]  Alegre et al. (2023) "Sample-Efficient Multi-Objective Learning via Generalized Policy Improvement Prioritization"
>
> [3] Lu et al., (2023) "Multi-objective reinforcement learning: Convexity, stationarity and pareto optimality"
>
> [4] Andrychowicz et al.(2018) "Hindsight Experience Replay. "

---

> > ### Comment · Reviewer_tiB3 · 2024-11-25
> >
> > First of all, the fact that the state-of-the-art MORL methods mentioned employ different techniques is **not a valid justification** for not performing experiments that compare the proposed method with them. Below, I try to summarize why I believe the experiments in this paper are currently insufficient to support the claims, and why some of the claims are incorrect:
> >
> > 1) "unlike the three Linear Scalarization (LS) methods [1][2][3] you mentioned, which are specifically designed for value-based RL."
> > This is incorrect. CAPQL is built on top of SAC, an actor-critic algorithm (thus, policy-based), and GPI-LS for continuous actions is built on top of TD3 (another actor-critic algorithm).
> >
> > 2) "GPI-LS [2] involves a specific training curriculum and CAPQL[3] needs reward augmentation—none of which are directly applicable to policy-based methods like PPO"
> > This is also incorrect. Curriculum learning techniques can be employed with any algorithm, and the reward augmentation of CAQPL is also a very simple idea that can be used with any algorithm: the only modification is to add a concave term to the reward function, which they showed to be enough to address the limitation of linear scalarization since it makes the Pareto front continuous and convex. Hence, **it is expected that the authors compare their method with CAPQL, since the work of CAPQL claims to have addressed the limitations of LS**.
> >
> > 3) As also pointed out by reviewer caBw, it is also expected that the authors would compare their method with PD-MORL. PD-MORL also introduces a technique that includes a directional angle term to increase the similarity between the preference vector and the value function. Notice that **PD-MORL is addressing a very similar issue to the one tackled by this paper and is a natural baseline to compare with**.
> >
> > 4) "Moreover, [1][2][3] did not report the results of Reacher. Could you please provide clear references for your claim? Regarding the LS method that you mentioned to learn many optimal policies, are they training the LS agent with uniformly sampled preferences? What are their evaluation metrics? Are they using the HER technique? This would be important to keep the discussion constructive."
> > The performance of state-of-the-art MORL algorithms on Reacher has been reported in morl-baselines by Felten et al. (2023). See https://wandb.ai/openrlbenchmark/MORL-Baselines for their results.
> > However, independently of the evaluation metrics used, it is natural that any linear scalarization MORL algorithm, trained with preferences [1,0,0,0] for instance, would be able to reach the first target of the domain. Hence, the results reported in Fig. 8, where the LS baseline only stays fixed in the central position are very unexpected.
> >
> > 5) "The main advantage we claim is the controllability of preferences, measured by the cosine similarity between preferences and values. The higher cosine similarity of PreCo compared to the two LS baselines supports our contribution."
> > The authors imposed that "controllability = cosine similarity between preferences and values". While their method optimizes for this cosine similarity, and naturally obtains higher value for this metric, this does not mean that other state-of-the-art MORL algorithms are not "controllable". For instance, CAPQL is able to learn a continuous and convex Pareto front. Why would someone argue that their method is not controllable?
> >
> > 6) "For the typical discrete 6d fruit tree environment applicable to Envelope Q-learning[1], even with HER techniques, their reported hypervolume was significantly lower"
> > First, [1] does not evaluate w.r.t. hypervolume, so I believe the authors are referring to the results reported by the PD-MORL paper. There are many reasons why this could be the case: different reference points when computing hypervolume, different numbers of preferences used for evaluating the PF, different budgets for training, etc. Independently of that, [1] reports a near to 1 coverage ratio, which means a very close to optimal solution to the problem. Hence, it is also surprising that any method would obtain such a higher value of hypervolume.
> >
> > Finally, the authors have accused me of being "biased and misleading", which is extremely **disrespectful and not appropriate in a peer-reviewing process**. As pointed out by reviewer NkSe: "Further, I believe **reviewer tiB3 is honestly and critically analyzing the paper, and the authors' strong statements in the response are highly unwarranted**. I urge the author's to be more respectful to the reviewers and their service to the community."

---

> > > ### Author Response · Authors · 2024-11-28
> > >
> > > Thank you once again for your time and valuable feedback. We sincerely appreciate the effort you took to share your thoughts and are truly grateful about your useful suggestions. We would like to kindly ask if your concerns have been fully addressed. Should you have any remaining questions, we would be more than happy to assist further.
> > >
> > > If your main concerns have indeed been resolved, it would mean a great deal to us if you could consider raising your rating further, as it would help us significantly.

---

> ### Author Response · Authors · 2024-11-25
> **Further clarification**
>
> We apologize for any strong statement that may need to be more respectful. With all due respect, we would like to clarify some points regarding your summarized points.
> ***
> **Point 1:**
>
> This is incorrect because SAC/TD3 rely on the Q value critic, all gradient information of their policy network come from the Q-value. We use the term "value-based" to emphasize their dependency on Q-value, which are essentially different from "policy-based" methods that rely on policy gradient and don't need a Q-value critic. Their implemented HER technique does not apply to PPO.
> ***
> **Point 2:**
>
> It is incorrect that GPI-LS can be applied to PPO because GPI-LS's curriculum and policy are based on "Generalized Policy Improvement (GPI)" that also requires only Q-value, therefore not applicable to PPO.
>
> CAPQL only proposed one practical implementation, using the entropy regularization term from SAC for concave reward augmentation, which is incompatible with TD3. PPO inherently includes policy entropy regularization, meaning our LS baseline with PPO already incorporates a policy loss with CAPQL’s concave augmentation. We also experimented with an additional entropy term to values, creating a "Maximum Entropy PPO" implementation, which can be viewed as a full CAPQL modified for PPO. The results for the 6D fruit tree environment are as follows:
>
> |$\alpha$|HV|HR|
> |-|-|-|
> |$0$ |$5741.8±877.4$|$0.718±0.040$|
> |$0.01$|$5952.2±1115.3$|$0.722±0.006$|
> |$0.05$|$5175.3±357.7$|$0.718±0.040$|
> |$0.1$|$1754.7±1349.0$|$0.633±0.141$|
>
> Only when the augmentation strength $\alpha=0.01$ is the result marginally better than the original LS ($\alpha=0$), yet it remains significantly worse than PreCo ($1.56e4$). Larger $\alpha$ values lead to performance drops. This result aligns with Remark 5 and Figure 9 in the CAPQL paper, which highlight that such augmentation can cause information loss in the original problem, and excessive augmentation results in performance degradation. In contrast, our method has the theoretical advantage of overcoming the LS limitation **without any information loss** from the original problem.
>
> (The 8427.51 Hypervolume of Envelope Q relies on the advantage of HER.)
>
> ***
> **Point 3:**
>
> Reviewer caBw is aware that a direct comparison with PD-MORL might be unfair because it is specifically designed for Q-value based method heavily relying on HER. We have tried to implement PD-MORL without HER for a fair comparison but it fails due to potential underrepresented preferences as pointed out in their original paper.
>
> ***
> **Point 4:**
>
> Through this link, we found MO-reacher results for PCN and GPI-LS. The hypervolume of PCN often converges to a value similar to that of the initialization, GPI-LS can achieve marginal improvement to the initial hypervolume due to a specific curriculum design. Therefore, this still fails to provide convincing evidence supporting that LS methods in a general setting (learned by uniformly sampled preferences without HER) can outperform our baselines.
>
> Any preference-conditioned method can learn a single preference better than random initialization; the challenge lies in robustness to stochasticity from uniform preference sampling. Our method has a **key theoretical advantage** in its **convergence under stochastic gradients**, as demonstrated in the toy example in Appendix F. Furthermore, even for a single preference, our method exhibits a cosine similarity advantage, also shown in Appendix F.
> ***
> **Point 5:**
>
> We have clarified that we never claimed "LS is **always** uncontrollable": "uncontrollable" was only used in the context of the 3d Fruit tree case where a very limited number of CCS solutions are learned with uniform sampled preferences. Our results do show better controllability in terms of cosine similarity.
> ***
> ### **Point 6: Key misunderstanding persists:**
>
> We would like to kindly refer you to our second response in detail, as it clarifies the key misunderstanding: Since **Convex coverage set(CCS) is only a subset of Pareto Optimal Set**, your mentioned "coverage ratio" is the coverage ratio of only CCS solutions, not all Pareto optimal solutions. A coverage ratio close to 1 only indicates the closeness to the upper limit of LS methods rather than the optimal solution.
>
> ***
> Once again we apologize for any strong statements because a rating of 3 is unfair and unexpected to us. We do greatly appreciate your time and respect your suggestions. We sincerely expect that the above can further clarify our contributions and resolve potential misunderstandings.

---

### Official Review · Reviewer_nUpd · 2024-10-28

**Soundness:** 3
**Presentation:** 2
**Contribution:** 3
**Rating:** 6
**Confidence:** 4

**Summary:**

The paper provides a method for optimizing multi-objective RL problems such that the obtained policy lies on the Pareto frontier. Instead of relying on scalarization of the objective, like standard MORL methods do, the proposed method instead optimized for a similarity metric between the preference vector and the value function. The paper provides a theoretical convergence analysis and compares with a number of adequate baselines on multiple environments.

**Strengths:**

The paper provides a novel and reasonable approach for finding pareto-optimal policies. This is non-trivial, most existing methods rely on scalarization and fall short, or require costly and complex constrained optimization methods. This paper instead provides a simpler updating scheme applicable in both discrete and continuous action-space MDPs. Good, adequate baseline comparison.

**Weaknesses:**

Clarity: Method section is hard to follow. In order to understand fully why the method converges to pareto-optimal policies, I'd appreciate if the authors could provide additional intuition (one or two sentences) for step 3 in section 3.2.

Experiments & reproducibility: While the paper provides a number of experiments, it does not seem like results were averaged across multiple random seeds, maybe I missed it? This is crucial though, because RL methods depend strongly on the seed. I strongly feel like the authors should report statistics across multiple seeds, otherwise it is very hard to say if the results are reproducible and significant.

Meta-RL: It is not clear to me why the authors learn a preference conditioned meta policy. Is this a requirement of the learning algorithm or is this solely done to have access to multiple policies at inference time? Does the method rely on training with uniformly sampled preference vectors or can it also optimize and find the pareto-optimal policy for a single preference, without uniformly sampling from the preference space during training at random?

**Questions:**

See the points under weaknesses. I think this is a good approach but also believe that the current version of the manuscript has some limitations that need to accounted for (or the clarity needs to improved) to justify a higher score.

---

> ### Author Response · Authors · 2024-11-19
> **A response**
>
> Thanks very much for the review and detailed comments!
>
> We wish to address your concern below:
> ***
> # Clarity:
>
> **Intuition for convergence to Pareto-optimal policies:**
>
> The term $||d^*||$ in (Eq. 6) is  an upper bound for $\min_w||\nabla_{\pi_p} v^{\pi_p} w||$. According to [1], when $\min_w||\nabla_{\pi_p} v^{\pi_p} w||=0$, it implies that the origin lies within the convex hull of the gradients. In this scenario, updating the policy in any direction will decrease at least one objective. This indicates that the policy is non-dominated and, therefore, Pareto-optimal. As updates are performed using $||d^*||$ in (Eq. 6), $||d^*||$ will converge to zero. Since $\min_w||\nabla_{\pi_p} v^{\pi_p} w||$ is bounded by $||d^*||$, it will also converge zero, confirming that the policies achieve Pareto optimality.
>
> We will include this explanation in the revisions for better clarity.
> ***
> # Experiments & Reproduciblity:
>
> The experimental results are averaged over five random seeds. We provide some easy-to-run code in this [anonymous git repo 1](https://anonymous.4open.science/r/PCRL-6E05), and the full code will be released upon acceptance.
>
> We will include the following implementation details in the revisions.
> ## Fruit-Tree
>
> |Hyper-parameter |Value|
> |-----|--------|
> |RL backbone|PPO|
> |Discount ($\gamma$)|$0.99$       |
> |Optimizer  |Adam    |
> |Learning rate for networks |$3 × 10^{−4}$|
> |Number of hidden layers for all networks|$3$|
> |Number of hidden units per layer|$256$|
> |Activation function|ReLU|
> |Batch Size|$100$|
> |Gradient clipping |False|
> |Exploration method|Policy Entropy|
> |Entropy Coefficient|$0.001$|
> |epsilon-clip for PPO|$0.001$|
> |Epochs per PPO update|$3$|
> |Timesteps every update |$100$|
> |Maximum episode timesteps| $100$|
> |Number of episodes per preference sample |$20$|
> |Number of preference samples (for reward dimensions from 3D-6D)| $3000$ |
> |Evaluation episode| $10$|
>
> ## MO-Reacher
> |Hyper-parameter |Value|
> |-----|--------|
> |RL backbone|PPO|
> |Discount ($\gamma$)|$0.99$       |
> |Optimizer  |Adam    |
> |Learning rate for networks |$3 × 10^{−4}$|
> |Number of hidden layers for all networks|$3$|
> |Number of hidden units per layer|$256$|
> |Activation function|ReLU|
> |Batch Size|$250$|
> |Gradient clipping |False|
> |Exploration method|Policy Entropy|
> |Entropy Coefficient|$0.001$|
> |epsilon-clip for PPO|$0.001$|
> |Epochs per PPO update|$3$|
> |Timesteps every update |$100$|
> |Maximum episode timesteps| $250$|
> |Number of episodes per preference sample |$40$|
> |Number of preference samples (for 4D reward)| $600$ |
> |Evaluation episode| $10$|
>
>
> ## MO-Ant & MO-Hopper
> |Hyper-parameter |Value|
> |-----|--------|
> |RL backbone|TD3|
> |Discount ($\gamma$)|$0.99$       |
> |Optimizer  |Adam    |
> |Learning rate for networks |$3 × 10^{−4}$|
> |Number of hidden layers for all networks|$3$|
> |Number of hidden units per layer|$256$|
> |Activation function|ReLU|
> |Batch Size|$256$|
> |Buffer Size|$1\times 10^6$|
> |Starting timesteps|$2.5\times 10^3$|
> |Gradient clipping |False|
> |Exploration method|Noise|
> |Noise distribution|$\mathcal{N}(0,0.1^2)$|
> |Noise clipping limit|0.5|
> |Policy frequency (delay)|2|
> |Target network update rate ($\tau$) |$5\times 10^{−3}$|
> |Maximum episode timesteps | $500$|
> |Maximum total timesteps|$3\times10^6$|
> |Peference sampling|every new episode until max total steps are reached |
> |Evaluation episode| $10$|
> ***
> # Meta-RL:
> The motivation for learning a preference-conditioned meta-policy is that we need access to policies for multiple preferences at inference time. In practice, a preference-conditioned agent that can 0-shot adapt to different user's preferences.
>
> The method can also identify the Pareto-optimal policy for a specific preference without requiring uniform sampling from the preference space during training. For better intuition, refer to the multi-objective optimization toy example in Appendix F (code available in [anonymous git repo 2](https://anonymous.4open.science/r/ICLR255131)).
>
> [1] Désidéri(2009) "Multiple-Gradient Descent Algorithm"

---

> > ### Author Response · Authors · 2024-11-23
> > **Looking forward to hearing your feedback**
> >
> > Dear reviewer nUpd
> >
> > Thanks again for your time, effort, and valuable suggestions to improve our work! We have addressed your initial concerns and revised our manuscript accordingly. We are looking forward to hearing your feedback and will be happy to answer any further questions or concerns you may have.
> >
> > Best regards,

---

> > ### Comment · Reviewer_nUpd · 2024-11-23
> >
> > Thank you for your thorough reply.
> >
> > The fact that results are indeed averaged over five random seeds eliminates a big concern. In addition, the hyperparameters for the baselines methods are sensible, to the best of my knowledge, which further increases the trustworthiness of the experiments. The additional results on the fruit-tree env provide additional support of the efficacy of this method.
> >
> > The revision, especially the addition of the method's pseudocode in app D.1, greatly benefit clarity, thanks for those.
> >
> > With those changes, I feel more confident in my understanding of this method and appreciate the contribution more. The scalarization approach in standard MORL is a major limitation that prevents learning of non-scalarizable tasks.
> > The method proposed in this paper, minimizing the similarity between the pareto preference vector and the value function, is novel, sensible, and a general purpose way for solving such challenging, non-scalarizable, multi-objective tasks.
> > I raised both my rating and confidence score under consideration of the strong theoretical analysis, extensive empirical results, and increased clarity of the paper.

---

> ### Author Response · Authors · 2024-12-02
> **Discussion Closing Soon: Your Feedback is Highly Valued**
>
> Dear reviewer nUpd,
>
> We hope this message finds you well. As the discussion period closes in less than 24 hours, we wish to sincerely thank you for acknowledging the "strong theoretical analysis, extensive empirical results, and increased clarity" of our work.
>
> We greatly value your recognition of the limitations of the popular Linear Scalarization (LS) approach in MORL and appreciate your acknowledgment that our work successfully overcomes these challenges.
>
> We respectfully ask you to consider raising your score or confidence level, as doing so would not only support our work but also benefit the MORL community. For example, even [the most recent MORL work that's published in Oct.3rd](https://arxiv.org/abs/2410.02236) continues to misinterpret linear scalarization’s ability to identify all Pareto-optimal policies. Our work addresses this critical issue, shedding light on key misconceptions and advancing future MORL research.
>
> Thank you for your time and thoughtful evaluation.
>
> Best regards,
>
> Author of submission 5131

---

### Official Review · Reviewer_NkSe · 2024-11-01

**Soundness:** 3
**Presentation:** 3
**Contribution:** 3
**Rating:** 6
**Confidence:** 2

**Summary:**

This paper presents Multi-objective Preference Control RL, an approach for multi-objective RL, which trains a meta-policy that takes user preference as input controlling the generated trajectories within the preference region on the Pareto frontier. They show that their meta-policy performs as
a multi-objective optimizer that can directly generate user-desired Pareto solutions. They theoretically analyze the convergence and controllability of the MORL objectives, and perform experiments on challenging robotics tasks.

**Strengths:**

The approach for employing preference control for MORL, by controlling generates trajectories is interesting and novel.

The paper performs detailed theoretical analysis and discusses convergence and controllability of the learnt policies, which is impressive.

The authors perform experiments and show impressive performance on challenging robotics tasks.

**Weaknesses:**

The paper method would benefit by adding a detailed algorithm explaining the method. The paper could benefit from the addition of pseudocode for the key steps of preference-conditioned policy training, and additional details on how the preference regularization is implemented algorithmically.

Extensive experimental and implementation details are missing, which would make the results hard to reproduce. Addition of information on hyperparameters, network architectures, training procedures, or environment specifications needed to reproduce the results will greatly help increase paper reproducibility.

I am currently recommending a weak reject, but will accordingly update the score based on the discussion and ratings by other reviewers.

**Questions:**

None.

---

> ### Author Response · Authors · 2024-11-18
> **A response**
>
> Thanks very much for the review and constructive feedback!
>
> We hope the pseudocode below offers better clarity for now, and we will include additional details in the revised version. For further intuition on how the proposed algorithm is implemented, you can also check and run the code for the toy example in Appendix F via this [anonymous link 1](https://anonymous.4open.science/r/ICLR255131/).
>
> ***
> # Pseudo Code
> ```python
> def train(N,E,agent):
>     # N: Number of preference samples
>     # E: Episodes per preference sample
>     # Agent should have preference-conditioned actor and critic
>     # The critic should output a vector of multi-objective values
>
>     for n in range(N):                     # Loop over preference samples
>         Sample preference p                # Sample a preference p
>         for e in range(E):                 # Loop over episodes for the this preference
>             Rollout with policy conditioned on p  # Rollout with the current preference
>             Store transitions (s,a,r_vec,p) in the buffer  # Store transitions
>
>         Sample (s,a,r_vec,p) from buffer       # Sample transitions from buffer
>         Update critic by the samples           # Approximate objective values
>         Calculate policy-level gradient        # Eq.20 for TD3, Eq.26/32 for PPO
>         Estimated values of the agent          # Eq.4 for TD3, Eq.5 for PPO
>         Calculate similarity gradient          # Eq.8
>         Solve Eq.6 using policy-level gradient   # Solve the problem for PreCo update
>         Update actor parameter by solution of Eq.6   # Eq.21/27 or Eq.29
> ```
> ***
> For the algorithmic implementation of the preference regulation, we estimate the values of the agent by Eq.(4) or Eq.(5), which is applicable to most mainstream RL algorithms.
>
> $$\hat{\mathbf{v}}^{\pi_\mathbf{p}} = E_{S_0 \sim p_0}\left[ \mathbf{q}_\theta(S_0,\pi(S_0,\mathbf{p}),{\mathbf{p}})  \right] \ \ \ (4)$$
>
> $$\hat{\mathbf{v}}^{\pi_\mathbf{p}} = E_{S_0 \sim p_0}\left[\sum_{t=0}^{T}\gamma^{t}\mathbf{r}(S_t,A_t) \right] \ \ \ (5)$$
>
>  then calculate the similarity defined by Eq.8. Take the gradient of the similarity w.r.t. the policy
>
> $$\nabla_{\pi_\mathbf{p}} \Psi(\mathbf{p},\hat{\mathbf{v}}^{\pi_\mathbf{p}}) = \nabla^T_{\pi_\mathbf{p}} \hat{\mathbf{v}}^{\pi_\mathbf{p}}\nabla_{\hat{\mathbf{v}}^{\pi_\mathbf{p}}} \Psi(\mathbf{p},\hat{\mathbf{v}}^{\pi_\mathbf{p}}),$$
>  we can get a similarity gradient that can be used for Eq.6.
> ***
> # Implementation details
>
> To address concerns about reproducibility, we have provided easy-to-run code for Fruit-Tree in this [anonymous link 2](https://anonymous.4open.science/r/PCRL-6E05). Additionally, we will include the following implementation details in the revised version:
> ***
> ## Fruit-Tree
>
> |Hyper-parameter |Value|
> |-----|--------|
> |RL backbone|PPO|
> |Discount ($\gamma$)|$0.99$       |
> |Optimizer  |Adam    |
> |Learning rate for networks |$3 × 10^{−4}$|
> |Number of hidden layers for all networks|$3$|
> |Number of hidden units per layer|$256$|
> |Activation function|ReLU|
> |Batch Size|$100$|
> |Gradient clipping |False|
> |Exploration method|Policy Entropy|
> |Entropy Coefficient|$0.001$|
> |epsilon-clip for PPO|$0.001$|
> |Epochs per PPO update|$3$|
> |Timesteps every update |$100$|
> |Maximum episode timesteps| $100$|
> |Number of episodes per preference sample |$20$|
> |Number of preference samples (for reward dimensions from 3D-6D)| $3000$ |
> |Evaluation episode| $10$|
> ***
> ## MO-Reacher
> |Hyper-parameter |Value|
> |-----|--------|
> |RL backbone|PPO|
> |Discount ($\gamma$)|$0.99$       |
> |Optimizer  |Adam    |
> |Learning rate for networks |$3 × 10^{−4}$|
> |Number of hidden layers for all networks|$3$|
> |Number of hidden units per layer|$256$|
> |Activation function|ReLU|
> |Batch Size|$250$|
> |Gradient clipping |False|
> |Exploration method|Policy Entropy|
> |Entropy Coefficient|$0.001$|
> |epsilon-clip for PPO|$0.001$|
> |Epochs per PPO update|$3$|
> |Timesteps every update |$100$|
> |Maximum episode timesteps| $250$|
> |Number of episodes per preference sample |$40$|
> |Number of preference samples (for 4D reward)| $600$ |
> |Evaluation episode| $10$|
>
> ***
> ## MO-Ant & MO-Hopper
> |Hyper-parameter |Value|
> |-----|--------|
> |RL backbone|TD3|
> |Discount ($\gamma$)|$0.99$       |
> |Optimizer  |Adam    |
> |Learning rate for networks |$3 × 10^{−4}$|
> |Number of hidden layers for all networks|$3$|
> |Number of hidden units per layer|$256$|
> |Activation function|ReLU|
> |Batch Size|$256$|
> |Buffer Size|$1\times 10^6$|
> |Starting timesteps|$2.5\times 10^3$|
> |Gradient clipping |False|
> |Exploration method|Noise|
> |Noise distribution|$\mathcal{N}(0,0.1^2)$|
> |Noise clipping limit|0.5|
> |Policy frequency (delay)|2|
> |Target network update rate ($\tau$) |$5\times 10^{−3}$|
> |Maximum episode timesteps | $500$|
> |Maximum total timesteps|$3\times10^6$|
> |Peference sampling|every new episode untill max total steps reached |
> |Evaluation episode| $10$|
> ***

---

> ### Author Response · Authors · 2024-11-23
> **Looking forward to hearing your feedback**
>
> Dear reviewer NkSe,
>
> Thanks again for your time, effort, and valuable suggestions to improve our work! We have addressed your initial concerns about clarity and reproducibility and have revised the manuscript accordingly. We are looking forward to hearing your feedback and will be happy to answer/discuss any further questions or concerns you may have.
>
> Best regards,

---

> > ### Comment · Reviewer_NkSe · 2024-11-25
> >
> > I appreciate the author's efforts for adding additional experimental details and pseudo-code which has improved paper clarity. However, I agree with reviewer tiB3 and caBw that the comparisons with state-of-the-art MORL approaches are missing, which are required to demonstrate the efficacy of this approach. Hence, I am keeping my original score.

---

> ### Author Response · Authors · 2024-11-25
> **key facts clarification**
>
> Thanks very much for the follow-up comment, and we are glad that your initial concerns about clarity and reproducibility are addressed. Just to clarify the **key facts** regarding the new concern of comparisons to ensure a fair and constructive final decision:
>
> ### **The opinion of reviewer caBw**
>
> In fact, reviewer caBw agrees that other methods [1][2][3] are not directly comparable or perform worse in harder environments (with larger objective number, not strictly convex Pareto front) with larger objective numbers. Their comment:
>
> > *"The additional experiments further demonstrate the effectiveness of the similarity function in improving the solution quality"*
>
> shows appreciation that our current experiments have fair and adequate comparisons, and are also sufficient to show the effectiveness of our proposed methods. The "comparison" in their final comment is only a suggestion for future work to adapt our method specifically for value-based off-policy RL and establish a fair apple-to-apple comparison.
>
> ### **Not directly comparable or significantly worse methods raised by reviewer tiB3:**
>
> Reviewer tiB3 listed [3][4][5], which involves significantly different settings (HER for off-policy RL [6], training curricula, and reward augmentation), making direct comparisons incompatible. Furthermore, even excluding HER, our baseline significantly outperforms [3] in harder environments, rendering comparison with [3] unnecessary. (Our implemented baseline has a significantly higher hypervolume ("$1.33e4$") than [3] ("$8427.51$") for 6-objective not strictly convex FTN environment)
>
> ### **Misleading and groundless accusations of reviewer tiB3**
> We kindly remind you that major accusations of reviewer tiB3 have been groundless or even proven to be false. For example, they use the following claim to attack our baselines:
> >*" (Yang et al. 2019)[3] is able to find most (if not all) the optimal solutions"*(while our baselines only learn limited Pareto optimal policies)
>
> However, the fact is that [3] only claimed to learn a convex coverage set, which is only a **subset** of Pareto optimal policies, and our implemented baseline has a significantly higher hypervolume ("$1.33e4$") than [3] ("$8427.51$"), learning even more Pareto optimal policies than [3].
>
> The core concerns of tiB3 have been based on the groundless accusation that "our LS baseline is poorly implemented". However, evidence has proven this accusation to be wrong. Therefore, tiB3's points appear biased and misleading.
>
> ***
> We kindly request you to reevaluate reviewer caBw's perspective and reassess the validity of reviewer tiB3's comments to ensure a fair and constructive final decision.
>
> Thanks in advance!
>
> Kind regards,
>
>
> ***
> [1] Basaklar et al.(2023). "PD-MORL: Preference-Driven Multi-Objective Reinforcement Learning Algorithm. In The Eleventh International Conference on Learning Representations."
>
> [2] Abels et al. "Dynamic weights in multi-objective deep reinforcement learning." International conference on machine learning. PMLR, 2019.
>
>  [3] Yang et al. (2019) "A generalized algorithm for multi-objective reinforcement learning and policy adaptation"
>
> [4] Alegre et al. (2023) "Sample-Efficient Multi-Objective Learning via Generalized Policy Improvement Prioritization"
>
> [5] Lu et al., (2023) "Multi-objective reinforcement learning: Convexity, stationarity and pareto optimality"
>
> [6]  Andrychowicz et al.(2018) "Hindsight Experience Replay. "

---

> ### Comment · Reviewer_NkSe · 2024-11-25
>
> The paper is still missing comparisons with other preference-based methods, and I strongly encourage the authors to include them to the final manuscript. However, in wake of reviewer caBw's comments, I am raising the current score.
>
> Further, I believe reviewer tiB3 is honestly and critically analyzing the paper, and the authors' strong statements in the response are highly unwarranted. I urge the author's to be more respectful to the reviewers and their service to the community.

---

> > ### Author Response · Authors · 2024-11-25
> >
> > Dear reviewer,
> >
> > Thank you very much, and apologize for the strong statements. We will be more respectful and work together to for a better community!
> >
> > Best regards,

---

> ### Author Response · Authors · 2024-11-28
> **Further response on baselines and our method's effectiveness**
>
> Dear reviewer,
>
> Thank you again for your valuable feedback and advice on fostering a respectful and positive community.
>
> We kindly remind you that we included a general response, "Final comment and additional SOTA baseline":
> 1. It clarifies why the listed methods [1][2][3][4] are not directly compatible.
>
> 2. Though CAPQL [4] was only implemented for SAC, we found a way to modify it into a general setting (without requiring Q-values), and we made a comparison with it to further demonstrate our method's effectiveness over SOTA methods.
>
> 3. We explained why the empirical results in the 6D fruit tree are sufficient to demonstrate our method's superiority over [1],[2], [3], and [4]. Its lower-dimensional discrete state and action spaces, combined with the challenge of 6 objectives and a non-strictly convex Pareto front, isolate MORL performance from lower-level RL factors, enabling a fair comparison independent of those modifications.
>
> 4. The result below confirms our LS baseline is properly implemented.
>
> ***
> Fruit-tree 6D
> |Method|HV(1e3)|
> |-|-|
> |GPI-LS[2]|$5.63\pm 0.1$|
> |Envelope[1]| $8.43$|
> |PD-MORL[3]|$9.30$|
> |CAPQL[4]|$5.95\pm 1.12$|
> |LS(our basline)|$5.74\pm 0.88$|
> |PreCo(our method)|$15.61\pm 0.75$|
>
> As shown, our LS baseline achieves comparable performance to the SOTA method GPI-LS [2], indicating it is not "poorly" implemented. Our proposed PreCo significantly outperforms the SOTA methods [1][2][3][4] in this environment that isolates MORL performance from lower-level RL factors.
>
> *Note: GPI-LS results are calculated using the "front" data from [link](https://wandb.ai/openrlbenchmark/MORL-Baselines/runs/wjclz5r4?nw=nwuseraraffin) provided by reviewer tiB3. Enelope and PD-MORL are reported in [3], and CAPQL represents our implementation of PPO with maximum entropy.*
> ***
> We sincerely hope this could further clarify your concerns and increase your confidence in your evaluation!
>
> ***
> [1] Yang et al. (2019) "A generalized algorithm for multi-objective reinforcement learning and policy adaptation"
>
> [2] Alegre et al. (2023) "Sample-Efficient Multi-Objective Learning via Generalized Policy Improvement Prioritization"
>
> [3] Basaklar et al.(2023). "PD-MORL: Preference-Driven Multi-Objective Reinforcement Learning Algorithm.
>
> [4] Lu et al., (2023) "Multi-objective reinforcement learning: Convexity, stationarity and pareto optimality"

---

> ### Author Response · Authors · 2024-12-02
> **Discussion Closing Soon: Your Feedback has been valuable**
>
> Dear Reviewer NkSe,
>
> We hope this message finds you well. As the discussion period closes in less than 24 hours, we wanted to kindly remind you of our earlier response addressing your comments.
>
> Before the Nov. 26 revision deadline, we added further results comparing state-of-the-art (SOTA) methods, demonstrating the empirical superiority of our approach despite unfair lower-level modifications like HER and specific training curriculum. Reviewer tiB3 acknowledged these improvements and raised their score, suggesting that we have adequately addressed some concerns.
>
> We kindly ask you to consider raising your score or confidence level, as this would not only support us but also contribute to the broader community. For example, even in [the most recent MORL work that's published in Oct.3rd](https://arxiv.org/abs/2410.02236), the misconception that linear scalarization can discover all Pareto optimal policies persists. Our work helps clarify this major issue, making a non-trivial contribution to the MORL community and advancing future research.
>
> Thank you for your time and thoughtful evaluation.
>
> Best regards,
>
> Author of submission 5131

---

### Official Review · Reviewer_caBw · 2024-11-01

**Soundness:** 3
**Presentation:** 3
**Contribution:** 2
**Rating:** 6
**Confidence:** 3

**Summary:**

This paper proposes a new multi-objective algorithm, PreCo, which aligns with preferences. Additionally, this work introduces a new similarity function as a regularizer for policy updates. Empirically, it demonstrates improved hypervolume (HV) scores across multiple multi-objective environments, showing the effectiveness of preference alignment.

**Strengths:**

+  This work provides comprehensive theoretical analysis for the convergence of Pareto stationary points.

+  Extensive experiments are conducted in both discrete and continuous state-action environments.

**Weaknesses:**

- There is a lack of a formal definition for the policy-level gradient, $\nabla_{\pi_p}v^{\pi_p}$, especially in continuous state-action spaces. Additionally, a major concern is the efficiency of computing the policy-level gradient in continuous state-action spaces. Recent RL works often use more expressive models as the policy network, such as diffusion models [1, 2] or normalizing flows [3]. It remains unclear how to compute the policy-level gradient for such generative models.

- The training time is not reported, which makes the claim of computational efficiency for solving the min-norm problem (Equation 6) in Section 3 unconvincing..

- The evaluation metrics are limited. The experiments mainly focus on hypervolume (HV) and cosine similarity between preferences and value functions. The similarity metric is designed to demonstrate the effectiveness of the proposed similarity function $\mathcal{\Psi}(\cdot, \cdot)$. Consequently, HV remains the sole evaluation metric for assessing the quality of the Pareto front. However, HV may increase due to improvement in only one of the objectives. In Figures 5(d) and 6(d), the authors provide visualizations of the Pareto front, which show only marginal improvements over EPO. Several other evaluation metrics could be used to assess the quality of results: Overall Non-dominated Vector Generation Ratio [4], Error Ratio [4], and Sparsity [5].

- Comparisons could be made more extensive. There exists a state-of-the-art preference-driven multi-objective RL algorithm [5].

- The authors may also wish to compare their approach with another baseline that incorporates preferences as input [6].

- Additionally, this work references SDMGrad [7] and aims to address a similar min-norm problem (Equation 6 in this work, Equation 8 in SDMGrad), yet experimental results from SDMGrad are missing. It would be valuable to observe the effectiveness of the similarity function based on a comparison between this work and SDMGrad.

[1] Michael Janner, Yilun Du, Joshua B. Tenenbaum, & Sergey Levine (2022). Planning with Diffusion for Flexible Behavior Synthesis. In International Conference on Machine Learning.
[2] Zhendong Wang, Jonathan J Hunt, & Mingyuan Zhou (2023). Diffusion Policies as an Expressive Policy Class for Offline Reinforcement Learning. In The Eleventh International Conference on Learning Representations.
[3] Brahmanage, Janaka, Jiajing Ling, and Akshat Kumar. "FlowPG: action-constrained policy gradient with normalizing flows." Advances in Neural Information Processing Systems 36 (2024).
[4] Van Veldhuizen, David Allen. Multiobjective evolutionary algorithms: classifications, analyses, and new innovations. Air Force Institute of Technology, 1999.
[5] Toygun Basaklar, Suat Gumussoy, & Umit Ogras (2023). PD-MORL: Preference-Driven Multi-Objective Reinforcement Learning Algorithm. In The Eleventh International Conference on Learning Representations.
[6] Abels, Axel, et al. "Dynamic weights in multi-objective deep reinforcement learning." International conference on machine learning. PMLR, 2019.
[7] Xiao, Peiyao, Hao Ban, and Kaiyi Ji. "Direction-oriented multi-objective learning: Simple and provable stochastic algorithms." Advances in Neural Information Processing Systems 36 (2024).

**Questions:**

- Are there other similarity functions that can be used in Equation (6), or what properties should these similarity functions have?

---

> ### Author Response · Authors · 2024-11-18
> **A response (1/2)**
>
> Thanks for the detailed review and constructive feedback! We aim to address your concerns below:
> ***
> > **Weakness 1.** *There is a lack of a formal definition for the policy-level gradient ...  generative models.*
> ***
> **Response 1:** Policy level gradient is defined for PPO/TD3 in Appendix.D. To provide more intuition, an illustrative diagram of the backward path for policy updates is provided in this [anonymous link](https://anonymous.4open.science/r/PCRL-6E05).
>
> For more expressive models in continuous space, we can replace $\nabla_{\pi_p}v^{\pi_p}$ in min-norm (eq.6) with the gradient w.r.t. $B$ samples of actions.
>
> $$\mathbf{g}_a=[\nabla{a_1}\mathbf{Q}({s_1},{a_1}),\nabla{a_2} \mathbf{Q}({s_2},{a_2}),...,\nabla{a_B} \mathbf{Q}({s_B},{a_B})]$$
>
> where $B$ denotes the batch size. Solving min-norm (Eq. 6) for $\mathbf{g}_a$ provides an update direction for the samples of action, enabling us to update the policy network accordingly.
>
>
> ***
> > **Weakness 2.** *The training time is not reported, which makes the claim of computational efficiency... unconvincing..*
> ***
> **Response 2:** The mainly claimed advantage is the **memory efficiency** of solving min-norm (eq.6) with $\nabla_{\pi_p}v^{\pi_p}$ rather than with $\nabla_{\theta}v^{\pi_p}$. The size of $\nabla_{\pi_p}v^{\pi_p}$ is typically $B$, the batch size (hundreds in most cases), whereas $\nabla_{\theta}v^{\pi_p}$'s size is the number of the neural network parameters, which can range from hundreds to billions.
>
> We conducted a simulated comparison of computational time on a problem with $6$ objectives, using a batch size of $128$ and a parameter size of $512$ to solve the min-norm problem in Eq. 6 for $100$ times. The results are summarized in the table below:
>
> |Gradient level|Gradient Size|Computation time(s)|
> |-----|---|-----|
> |Policy-level|128|4.33|
> |parameter-level|512|15.56|
>
> The parameter-level gradient was simulated using samples from "torch.rand(512)", while the policy-level gradient was simulated with "torch.rand(128)". These results already demonstrate a significant gap in computation time. In practice, this gap in gradient sizes can be more drastic—for example, 256 for policy-level versus vs. 7B for parameter-level gradients.
> ***
> > **Weakness 3.** The evaluation metrics are limited...
> ***
> **Response 3:** Thank you for the constructive suggestions on evaluation metrics! Below are the extended versions of Tables 2,3 and 4, now including Sparsity (SP) and Overall Non-dominated Vector Generation Ratio (NR) for Figures 5 and 6.
>
> **MO-Hopper(Fig.5)**
> |Method|HV(1e6)|HR|SP(1e3)|NR|
> |-----|---|-----|---|-----|
> |LS|2.52|0.929|10.73|0.35|
> |SDMgrad|2.24|0.904|4.03| 0.40|
> |EPO|2.06|0.985|35.04| 0.25|
> |CAGrad|0.94|0.902|2.74| 0.40|
> |PreCo(Ours)|2.76|0.957|22.27|0.60|
>
> **MO-Ant(Fig.6)**
> |Method|HV(1e6)|HR|SP(1e3)|NR|
> |-----|---|-----|---|-----|
> |LS|6.37|0.985|35.54|0.70|
> |SDMgrad|3.66|0.928|5.92| 0.60|
> |EPO|6.75|0.996|71.39| 0.55|
> |CAGrad|7.11|0.988|40.27| 0.65|
> |PreCo(Ours)|7.02|0.998|55.77|0.60|
>
> **MO-Reacher(Fig.7)**
> | method  | HV(1e8)    | HR    | SP                 | NR   |
> | ------- | ----- | ----- | ------------------ | ---- |
> | LS      | 20.55 | 0.762 | 30.73            | 0.75 |
> | SDMgrad | 20.67 | 0.763 |43.64              | 0.81|
> | EPO     | 11.40 | 0.906 | 703.78             | 1.0  |
> | CAgrad  | 2.82  | 0.761 | 3.87 | 1.0  |
> | PreCo(Ours)  | 33.78 | 0.908 | 6484.51  | 1.0  |
>
>
> We observe that the values produced by PreCo exhibit decent sparsity and a high non-dominated ratio. Full results for SP and NR will be included in the revisions.
> ***
> > **Weakness 4.** Comparisons could be made more extensive. ... multi-objective RL algorithm [5].
> ***
> **Response 4:** Thanks for the helpful suggestion and we will include a reference to PD-MORL in the related works section. However, PD-MORL [5] heavily relies on HER [1], a technique specifically designed for off-policy methods, which limits its general applicability to on-policy methods. A direct comparison between the full implementation of PD-MORL and PreCo is not entirely fair, as HER could also be incorporated into our method to achieve further improvements.
>
> We have tried to implement [5] in our setup without using HER, but the learning failed for mo-Hopper V4. Their paper[5] suggested that this may be due to certain preferences being underrepresented. We also believe a main reason is the conflicts among their deterministic policies' gradients, as their policies must optimize for both cosine similarity and Q-values. Instead, methods like MGDA[2], CAGrad[3], SDMGrad[4] and our PreCo find a common ascend direction (e.g., by solving min-norm problem in Eq. 6) to avoid gradient conflicts. In future work, we could draw inspiration from [5] to develop a better MORL algorithm specifically designed for off-policy, value-based methods.
>
> Additionally, their reported hypervolume for 6-D reward Fruit-Tree Navigation is $9299.15$, which  significantly lower than our PreCo's $1.56e4$.

---

> ### Author Response · Authors · 2024-11-18
> **A response (2/2)**
>
> ***
> > **Weakness 5.** The authors may also wish to compare their approach with another baseline that incorporates preferences as input [6].
> ***
> **Response 5:** Similar to [5], [6] is a MORL algorithm specifically designed for value-based methods, so it cannot be directly generalized to policy-based methods like PPO. In contrast, our proposed PreCo is a higher-level approach compatible with most mainstream RL algorithms. Additionally, [6] aims to maximize the linear scalarization (LS) objective, so it is an LS approach (a baseline covered in our experiments) tailored for Q-learning.
>
> ***
> > **Weakness 6.** Additionally, this work references SDMGrad ... SDMGrad.
> ***
>
> **Response 6:**
>
> **SDMGrad results are already included in all of our experiments reported in Section 5**.
>
>
> The key difference in formulation between PreCo and SDMGrad is that PreCo adds a similarity gradient (Eq.6) while SDMGrad SDMGrad incorporates a linear scalarization (LS) gradient (Eq.14 in Appendix B.1). For this reason, we categorize SDMGrad as an LS method and use it as an ablation case to compare the performance with and without our proposed similarity function.
>
>
> ## Answering questions
> ***
> > **Q1:** Are there other similarity functions that can be used in Equation (6), or what properties should these similarity functions have?
> ***
> **A1:** For Thereom 4.1 (Pareto optimal for constant $\lambda$) and 4.3 (Controllable for constant $\lambda$) to be valid:
> 1. It should be Lipschitz smooth.
> 2. Its gradient $g_s$ should be a positive linear combination of the objective gradients $g_i$. For example, the similarity gradient can be $g_s = 5g_1+0.5g_2$, where $5$ and $0.5$ are positive coefficients.
>
> For Thereom 4.2 (Pareto optimal for increasing $\lambda$) and 4.4 (Controllable for increasing $\lambda$) to be valid:
>
> 1. It should be Lipschitz smooth.
> 2. Its gradient $g_s$ should be convex combination of the objective gradients $g_i$. For example, the similarity gradient can be $g_s = 0.5g_1+0.5g_2$, where $0.5+0.5=1$ and they are convex coefficients.
>
> Appendix. E could offer more insights into our similarity function design.
> ***
>
> [1] Andrychowicz et al.(2018) "Hindsight Experience Replay. "
>
> [2] Désidéri(2009) "Multiple-Gradient Descent Algorithm"
>
> [3] Liu et al.(2021) "Conflict-Averse Gradient Descent for Multi-task Learning"
>
> [4] Xiao et al.(2023) "Direction-oriented Multi-objective Learning: Simple and Provable Stochastic Algorithms"
>
> [5] Basaklar et al.(2023). "PD-MORL: Preference-Driven Multi-Objective Reinforcement Learning Algorithm. In The Eleventh International Conference on Learning Representations."
>
> [6] Abels et al. "Dynamic weights in multi-objective deep reinforcement learning." International conference on machine learning. PMLR, 2019.

---

> > ### Comment · Reviewer_caBw · 2024-11-23
> >
> > I appreciate the authors' detailed response and their efforts to address my concerns. The additional experiments further demonstrate the effectiveness of the similarity function in improving the solution quality. Consequently, I have decided to raise my score to 6.
> > However, I strongly recommend that the authors include comparisons with other preference-based methods, such as [5, 6]. Currently, the authors only provide results from the 6D reward Fruit-Tree environment, which is quite limited. Akin readers may conclude that the performance of PreCo is inferior to PD-MORL in the Mo-Ant environment. A full implementation of both PD-MORL and PreCo with HER would provide a proper apple-to-apple comparison. Moreover, since PreCo is an off-policy method suitable for continuous action spaces, the lack of comparisons with other off-policy preference-based methods is unsatisfactory.

---

> ### Author Response · Authors · 2024-11-28
>
> Thanks once again for your time and effort in evaluating this work. Your constructive suggestions have greatly helped us improve the manuscript. We fully agree that a modified PreCo in an off-policy setting with HER could make the results more convincing. Due to computational and time limitations, we might only be able to explore this in the future.
>
> However, we would like to clarify one point regarding the environment:
>
> Although the Ant environment is indeed challenging for RL due to its high dimensionality and sample complexity, its Pareto front landscape is actually quite simple. The multiple objectives in Ant are quite symmetric (e.g., go east or go north), and its Pareto front is strictly convex. As a result, performance in Ant can be highly dependent on the underlying RL implementation (such as HER[1][3] and training curriculum[2]) rather than the MORL algorithm itself.
>
> In contrast, the fruit-tree environment features lower-dimensional discrete state and action spaces, combined with the challenge of 6 objectives and a non-strictly convex Pareto front. This design isolates MORL performance from lower-level RL factors, enabling a fairer comparison that is independent of those modifications.
>
> Below is the table showing the superior performance of our methods over the mentioned methods [1][2][3][4] during the discussions, including their full implementations (such as HER[1][3] and curriculum training[2]), highlighting our advantages over SOTA MORL methods.
>
> **Fruit-tree 6D**
> ***
> |Method|HV(1e3)|
> |-|-|
> |GPI-LS[2]|$5.63\pm 0.1$|
> |Envelope[1]| $8.43$|
> |PD-MORL[3]|$9.30$|
> |CAPQL[4]|$5.95\pm 1.12$|
> |LS (our baseline)|$5.74\pm 0.88$|
> |EPO (our proposed, used as baseline)|$14.97\pm 2.29$|
> |PreCo (our proposed, used as main method)|$15.61\pm 0.75$|
>
> *Note: All results are calculated from objective returns, using the origin as the reference point. GPI-LS results are calculated using the "front" data from [link](https://wandb.ai/openrlbenchmark/MORL-Baselines/runs/wjclz5r4?nw=nwuseraraffin) provided by reviewer tiB3. Envelope and PD-MORL are reported in [3], and CAPQL represents our implementation of PPO with maximum entropy explained in Appendix G.*
>
> ***
>
> If this further resolves your concerns, we would be truly grateful if you could consider increasing your ratings or confidence.
>
> ***
> [1] Yang et al. (2019) "A generalized algorithm for multi-objective reinforcement learning and policy adaptation"
>
> [2] Alegre et al. (2023) "Sample-Efficient Multi-Objective Learning via Generalized Policy Improvement Prioritization"
>
> [3] Basaklar et al.(2023). "PD-MORL: Preference-Driven Multi-Objective Reinforcement Learning Algorithm.
>
> [4] Lu et al., (2023) "Multi-objective reinforcement learning: Convexity, stationarity and pareto optimality"

---

> ### Author Response · Authors · 2024-12-02
> **Discussion Closing Soon: Your Feedback is greatly Valued**
>
> Dear Reviewer caBw,
>
> We hope this message finds you well. With the discussion period closing in less than 24 hours, we kindly remind you of our earlier response addressing your comments.
>
> Before the Nov. 26 revision deadline, we added new results comparing state-of-the-art methods, PD-MORL and CAPQL, with their lower-level modifications (HER and preference space segmentation) preserved. They demonstrate the empirical superiority and theoretical advantages of our approach. Unlike CAPQL, our method avoids information loss by preserving the original problem structure. Compared to PD-MORL, our approach is more general and avoids potential conflicts between cosine similarity and objective gradients.
>
> We respectfully ask you to consider raising your score or confidence level, as doing so would not only support our work but also contribute significantly to the MORL community. For example, even [the most recent MORL work that's published in Oct.3rd](https://arxiv.org/abs/2410.02236) continues to misinterpret the ability of linear scalarization to discover all Pareto-optimal policies. Our work addresses this critical issue, offering a valuable contribution to the MORL community and advancing future research.
>
> Thank you for your time and thoughtful evaluation.
>
> Best regards,
> Author of submission 5131

---

### Author Response · Authors · 2024-11-22
**Summary of revision**

We sincerely thank all reviewers [R1 (caBw), R2 (NkSe), R3 (nUpd), R4 (tiB3)] for their thoughtful feedback. We are glad that the reviewers recognized the novelty and importance of the problem we address [R2, R3, R4], appreciated the depth and comprehensiveness of our theoretical analysis [R1,R2], and found our choice of evaluation environments extensive and appropriate [R1,R2,R3,R4].

Based on the feedback, we have revised the manuscript with the following adjustments to enhance clarity and reproducibility:

* Added a more detailed pseudocode for improved clarity(Algo.2).
* Included an illustrative diagram to enhance intuition of the proposed method(Fig.9).
* Provided code and figures for a [toy example](https://anonymous.4open.science/r/ICLR255131) to improve understanding(App.F).
* Included implementation details and [easy-to-run code](https://anonymous.4open.science/r/PCRL-6E05) to ensure reproducibility(App.C).
* Reran experiments with multiple seeds and updated results with statistical dispersion metrics(Sec.5, App.C).
* Added additional comparisons with related works(App.A).
* Refined the text for better clarity, improved intuition, and consistent terminology(Sec.2-5).

Our contributions include three key aspects:
* **Conceptual**: Introduced similarity-based preference control to overcome limitations of linear scalarization, and incorporated gradient manipulation from optimization literature for MORL.
* **Theoretical:** Proved that our proposed method has the desired properties for MORL, specifically that it converges to preference-aligned Pareto-optimal policies under stochastic gradients.
* **Algorithimic:** Developed a practical algorithm and conducted experiments that demonstrate its effectiveness.

We hope the revision clarifies our theoretical contributions and further justifies the effectiveness of our algorithm.

---

### Author Response · Authors · 2024-11-26
**Final comment and addtional SOTA baseline**

We sincerely thank all reviewers for their time and valuable feedback. We appreciate your careful examination of our rebuttal and the subsequent score adjustments. Here, we provide a final comment with additional results in our 2nd revision to further strengthen your confidence in the evaluation.
***
Regarding the comparisons raised by reviewers tiB3 ([1][2][4]) and reviewer caBw ([3]):

- **[1][2][3]:** These are methods limited to off-policy RL algorithms with Q-functions (e.g., HER[5], GPI[6]). They are incompatible with our general setting, which also applies to A3C/PPO, methods that do not require Q-functions.

- **[1][2]:** Exclusively optimizing the Linear Scalarization (LS) objective, their solutions are restricted to the Convex Coverage Set (CCS)[7], only a subset of the Pareto front, whereas our proposed PreCo and EPO are similarity-based methods that are not restricted by CCS solutions.

- **CAPQL[4]**, Though implemented for SAC, it is the only SOTA method overcoming LS limitations and is potentially generalizable to policy-based methods without Q-functions. However, as noted in their Remark 5, its reward augmentation can cause information loss in the original problem, with excessive augmentation potentially leading to performance degradation. In contrast, our proposed method has the theoretical advantage of preserving the original problem for robust performance.

For a convincing demonstration of our method's superiority over SOTA methods in a fair comparison, we adapted CAPQL for PPO and tested it independently of HER-like techniques, highlighting the advantages of our approach.
***

We have implemented a new SOTA baseline incorporating the reward augmentation term from CAPQL[1]. CAPQL was originally designed for SAC, as the entropy maximization in SAC serves as the reward augmentation focus[1]. To ensure a fair comparison independent of settings, code-level implementations, and algorithmic techniques (such as HER[2]), we modified our original LS with PPO into a "maximum entropy PPO", where the critic values $v^\pi(s,p)$ learn the weighted sums of returns and entropies of policy distribution $\pi(\cdot|s,p)$. The advantages are then calculated by weighted sums of returns and entropies minus values $v^\pi(s,p)$. Details are in the Appendix G of 2nd revision of our manuscript.

The results for the 6D Fruit Tree environment are shown below:
|$\alpha$|Hyper volume(1e3)|Cosine Similarity|
|-|-|-|
|$0 $ (LS) |$5.74±0.88$|$0.718±0.040$|
|$0.01$|$5.95±1.12$|$0.722±0.006$|
|$0.05$|$5.18±0.36$|$0.718±0.040$|
|$0.1$|$1.75±1.35$|$0.633±0.141$|
|EPO|$14.97±2.29$|$0.77±0.03$|
|PreCo(ours)|$15.61±0.75$|$0.78±0.03$|

The Fruit-Tree environment is simple for RL due to small discrete state and action spaces but challenging for MORL with 6 objectives and a non-strictly convex Pareto front. This comparison isolates MORL performance from lower-level RL factors, directly highlighting our method's strengths. The results show when the augmentation strength $\alpha=0.01$, the performance of CAPQL-modified PPO is marginally better than the original LS ($\alpha=0$), but still significantly worse than similarity-based methods (EPO, PreCo(ours)). Larger $\alpha$ values lead to performance drops. This result aligns with Remark 5 and Figure 9 in [1], which highlights that such augmentation can cause information loss in the original problem, and excessive augmentation results in performance degradation. In contrast, our method has the theoretical advantage of overcoming the LS limitation without any reward augmentation, thus avoiding information loss from the original problem.

The results in this environment, with a high objective number of 6 and a non-strictly convex Pareto front, effectively demonstrate the advantages of our proposed methods over CAPQL. We have included this result and analysis in the Appendix G of the 2nd revision of our manuscript.

***
[1] Yang et al. (2019) "A generalized algorithm for multi-objective reinforcement learning and policy adaptation"

[2] Alegre et al. (2023) "Sample-Efficient Multi-Objective Learning via Generalized Policy Improvement Prioritization"

[3] Basaklar et al.(2023). "PD-MORL: Preference-Driven Multi-Objective Reinforcement Learning Algorithm.

[4] Lu et al., (2023) "Multi-objective reinforcement learning: Convexity, stationarity and pareto optimality"

[5] Andrychowicz et al.(2018) "Hindsight Experience Replay. "

[6] Barreto et al.(2020) "Fast reinforcement learning with generalized policy updates"

[7] Roijers et al.(2013) "A Survey of Multi-Objective Sequential Decision-Making"

---

### Author Response · Authors · 2024-11-28
**Evidence showing our original implementation of Linear Scalarization (LS) is fair and adequate**

Thanks again for your attention! This is further evidence that our LS baseline has been properly implemented. If this resolves your concerns, we would be truly grateful if you could consider increasing your ratings or confidence.

Using the [link](https://wandb.ai/openrlbenchmark/MORL-Baselines/runs/wjclz5r4?nw=nwuseraraffin) provided by reviewer tiB3, we retrieved results for the state-of-the-art method GPI-LS [2]. We extracted the objective data and calculated the hypervolume using the origin as the reference point, finding it to be $\mathbf{5.63e3}$. This value is very close to the $\mathbf{5.74e3}$ achieved by our LS baseline, confirming that our implementation is properly executed and achieves comparable performance to SOTA methods.

A complete comparison of the discussed SOTA methods[1][2][3][4] is presented in the table below:

**Fruit-tree 6D**
***
|Method|HV(1e3)|
|-|-|
|GPI-LS[2]|$\mathbf{5.63\pm 0.1}$|
|Envelope[1]| $8.43$|
|PD-MORL[3]|$9.30$|
|CAPQL[4]|$5.95\pm 1.12$|
|LS (our baseline)|$\mathbf{5.74\pm 0.88}$|
|EPO (our proposed, used as baseline)|$14.97\pm 2.29$|
|PreCo (our proposed, used as main method)|$15.61\pm 0.75$|

The fruit-tree environment features lower-dimensional discrete state and action spaces, combined with the challenge of 6 objectives and a non-strictly convex Pareto front. **This design isolates MORL performance from lower-level RL factors, enabling a fair comparison independent of such modifications**. As shown, our LS baseline achieves performance comparable to the SOTA method GPI-LS [2], confirming it is not "poorly" implemented. Additionally, the results further validate our claimed advantages of similarity-based methods (EPO/PreCo), as they significantly outperform the SOTA methods, even without the benefits of a specific training curriculum or HER.

*Note: GPI-LS results are calculated using the "front" data from [link](https://wandb.ai/openrlbenchmark/MORL-Baselines/runs/wjclz5r4?nw=nwuseraraffin) provided by reviewer tiB3. Enelope and PD-MORL are reported in [3], and CAPQL represents our implementation of PPO with maximum entropy explained in Appendix G. All results are calculated from objective returns, using the origin as the reference point.*

***
[1] Yang et al. (2019) "A generalized algorithm for multi-objective reinforcement learning and policy adaptation"

[2] Alegre et al. (2023) "Sample-Efficient Multi-Objective Learning via Generalized Policy Improvement Prioritization"

[3] Basaklar et al.(2023). "PD-MORL: Preference-Driven Multi-Objective Reinforcement Learning Algorithm.

[4] Lu et al., (2023) "Multi-objective reinforcement learning: Convexity, stationarity and pareto optimality"

---

### Author Response · Authors · 2024-12-02
**Summary of the Discussion**

## Summarised contribution
We propose a novel general framework, **Preference Control (PC) RL**, for Multi-Objective Reinforcement Learning (MORL) with preference control. Our main contribution can be summarized as:

* PCRL is the first preference conditioned MORL framework to integrate recent advances in Multi-Objective Optimization (MOO) [1][2][3].
* It supports both policy-based and value-based RL algorithms.

Our proposed algoirthm PREference COntrol (**PreCo**) unifies the following key advantages of several recent MOO algorithms and is specifically designed for RL.

1. Exact preference alignment [1].
2. Conflict resolution for multi-objective gradients [2].
3. Robustness to gradient stochasticity [3].

Rigorous theoretical analysis, illustrative toy example and extensive MORL experiments are conducted to justify the advantages of PreCo

***
## Before and after discusion
We provide a summary of the reviewer-author discussion:

**Main concerns of reviewers before discusion:**
1. Clarity of definitions, math notations and algorithm details.
2. Reproducibility and seed-sensitivity of experiments.
3. Evaluation metrics can be more extensive.
4. Not sure if our baselines represent SOTA performance of MORL methods

**During the discussion:**
1. We improved the manuscript with additional diagrams, pseudocode, source code, and illustrative toy examples.
2. We reported the variance and confidence interval as requested. For reproducibility, we provided source code, seeds, and implementation details (hyperparameters, network architectures, training procedures, and environment specifications).
3. We reported additional evaluation metrics, such as sparsity and overall non-dominated vector generation ratio.
4. We provided evidence confirming that our linear scalarization (LS) baseline's performance is SOTA among LS-typed methods by comparing it with a SOTA method of the type [4]. We also compared our proposed PreCo against all discussed SOTA methods in a challenging MORL environment, demonstrating its superiority over [4][5][6][7], even though [4][5][6] rely on unfair lower-level modifications such as HER [8] and specific training curriculums.

**After the discussion:**
**All reviewers increased their scores**, acknowledging that we adequately addressed their concerns. While all other reviewers provided positive scores, reviewer tiB3 only raised their score to 5. We reached out for clarification on their remaining concerns, but unfortunately, we have not receive a response so far.


***
[1] Mahapatra et al.2020 "Multi-Task Learning with User Preferences:..."

[2] Liu et al 2021 "Conflict-averse gradient descent for multi-task learning."

[3] Xiao et al 2023 "Direction-oriented multi-objective learning:..."

[4] Alegre et al. (2023) "Sample-Efficient Multi-Objective Learning via Generalized Policy Improvement Prioritization"

[5] Yang et al. (2019) "A generalized algorithm for multi-objective reinforcement learning and policy adaptation"

[6] Basaklar et al.(2023). "PD-MORL: Preference-Driven Multi-Objective Reinforcement Learning Algorithm.

[7] Lu et al., (2023) "Multi-objective reinforcement learning: Convexity, stationarity and pareto optimality"

[8] Andrychowicz et al.(2018) "Hindsight Experience Replay. "

---

### Author Response · Authors · 2024-12-03
**Final Hour Reminder: Your feedbacks are invaluable**

Dear Reviewers

Thank you once again for all your invaluable feedback! As we enter the final hour of discussion, we kindly ask you to consider providing greater support for our work by increasing your scores and confidence levels.

We believe this is justified for the following reasons:

1. **Solid Approach:** We have introduced a systematic and general methodology to address the limitations of most MORL methods based on Linear Scalarization (LS). This is supported by both rigorous theoretical analysis and extensive experimental evidence.
1. **Novelty:** To our knowledge, we were among the first to design a MORL algorithm inspired by recent advancements in multi-objective optimization. Before our work, existing MORL methods mentioned during the discussion had not explicitly addressed the challenges of gradient conflicts and common improving directions.
2. **Advancing the Field:** Our work highlights overlooked limitations of LS, which even the most recent MORL publications fail to address[1], claiming full Pareto front discovery of LS method. Recognizing our contributions could help clarify this misconception and guide future research in the MORL community.
3. **Addressing Concerns:** We have thoroughly addressed the initial concerns through improved clarity, additional experimental results, and by providing source code and implementation details.
4. **Resolved Misunderstandings:** Many critiques stemmed from misunderstandings, which we have since clarified. After these clarifications, we believe no doubts remain regarding the validity of our contributions.

Thank you for your consideration and support!


***
[1] Liu et al. (2024 Oct.3rd)"C-MORL: Multi-Objective Reinforcement Learning through Efficient Discovery of Pareto Front"

https://arxiv.org/abs/2410.02236

---

### Meta-Review · Area_Chair_gGve · 2024-12-26

**Metareview:**

The authors kindly provided a summary of contributions and reviewers' concerns. As strongly suggested by the authors, the AC reconsidered the evaluation of a specific reviewer.

This paper investigate the multi-objective reinforcement learning (MORL) problem with user preferences. It "takes user preference as input
controlling the generated trajectories within the preference region on the Pareto frontier." The authors provided theoretical analysis and simple empirical evaluations. There is a lack of reference on Pareto optimization and preference-based RL. On the analytical side, the theoretical results are as expected. It would be better to have sufficient empirical validation across a broader range of environments.

It is a nice incremental work for MORL, but hasn't reached the bar of ICLR 2025.

**Additional Comments On Reviewer Discussion:**

One reviewer raised their score from 3 to 5. The authors strongly suggested the AC to reconsider the evaluation of this specific reviewer.

---

### Decision · Program_Chairs · 2025-01-22

Reject